# ML Estimation from Bits

## Abstract

Estimating statistical parameters from quantized signals has received significant attention in recent years, as recovering information from quantized measurements has numerous applications across signal processing, communications, and data analysis. In this work, we focus on maximum likelihood (ML) estimation of statistical parameters from quantized samples. Directly solving the ML problem is challenging, as the likelihood function involves multiple integrals that are difficult to evaluate. To address this challenge, we propose an expectation-conditional-maximization (ECM) algorithm under a general distributional framework. Our approach generalizes the quantization model to multi-bit settings and allows the underlying signal to follow any distribution within the normal mean-variance mixture family. By designing suitable surrogate functions, the ECM algorithm ensures that all model parameters can be updated in closed form at each iteration. Leveraging the ECM framework, we provide convergence guarantees, and under specific distributional assumptions, we further derive bounds on the convergence rate and the statistical error. Extensive experiments demonstrate the effectiveness of our method in recovering statistical parameters from quantized data.

## 1 Introduction

Quantization, which represents signals using a finite number of bits, offers a hardware-efficient approach for data storage and transmission, reducing power consumption while maintaining acceptable accuracy (Roberts, 1962; Widrow & Kollár, 2008). These properties have led to the widespread adoption of quantization in applications that require efficient processing of large-scale data.

In recommendation systems, for example, user ratings are often quantized (e.g., binary preferences or 1-5 star ratings), motivating extensive research on recovering the underlying rating matrix (Davenport et al., 2014; Cai & Zhou, 2013; Bhaskar, 2016; Bottegal & Suykens, 2017; Gao et al., 2018). In compressed sensing, quantization arises when estimating sparse signals from limited measurements, with a growing focus on one-bit and multi-bit quantized sensing (Boufounos & Baraniuk, 2008a; Zymnis et al., 2009; Plan & Vershynin, 2013; Ai et al., 2014; Shao et al., 2024). Quantization is also widely used in wireless communication and sensing, including channel estimation, array detection, and radar signal processing (Choi et al., 2016; Stöckle et al., 2016; Plabst et al., 2018; Ren & Li, 2017; Ameri et al., 2019; Jin et al., 2020; Bar-Shalom & Weiss, 2002b; Lu et al., 2024). Beyond signal recovery, quantized data naturally appear in regression and subspace learning, giving rise to quantized regression and subspace estimation problems (Mayne, 1967; Gyorfi & Wegkamp, 2008; Chen et al., 2023; Chi & Fu, 2017; Dirksen et al., 2025).

In many applications, the primary interest is not the signal itself, but its underlying statistical properties. Researchers accomplish downstream tasks such as classification and anomaly detection by estimating the mean and covariance matrix of the signal. In many applications, the primary interest is not the signal itself, but its underlying statistical properties, such as the mean (Papadopoulos et al., 2001; Dabeer & Masry, 2008) for distributed detection (Ribeiro & Giannakis, 2006a;b; Fang & Li, 2008), and the covariance matrix (Van Vleck & Middleton, 1966; Gray & Stockham, 1993; Liu & Lin, 2021; Dirksen et al., 2022; Xiao et al., 2023) for direction-of-arrival estimation (Eamaz et al., 2022; 2023; Bar-Shalom & Weiss, 2002a), power estimation in wireless sensor networks (Mo et al., 2017), spectrum sensing (Yang et al., 2025), and networked sensing (Chi & Fu, 2017).

However, quantization inevitably discards part of the original information, making the recovery of signals or their statistical properties from quantized measurements a challenging and practically

important problem. In this paper, we study parameter estimation of $\boldsymbol{x}$ from quantized measurements. Given a random signal $\boldsymbol{x}$, the quantized measurement $\boldsymbol{y}$ is obtained in the following way

$$\boldsymbol{y} = \mathcal{Q}(\boldsymbol{x}), \tag{1}$$

where $\mathcal{Q} : \mathbb{R}^d \to \mathcal{K}^d$ is a quantization function, or quantizer, with the set $\mathcal{K}$ containing finite $e$ elements, i.e., $\mathcal{K} = \{k_1, \ldots, k_e\}$. The definition of the $i$-th element $[\mathcal{Q}(\boldsymbol{a})]_i$ is given by

$$[\mathcal{Q}(\boldsymbol{a})]_i = k_l, \quad \text{if } a_i \in [\tau_{l-1}, \tau_l), \tag{2}$$

where $\bigcup_{l=1}^{e}[\tau_{l-1}, \tau_l) = \mathbb{R}$ with $\tau_0 = -\infty$ and $\tau_e = \infty$. When the number of bits used to represent the quantized value, i.e., $\log_2 e$, is small, we call it a coarse quantization. The coarsest quantization method, which is also the most commonly adopted one in practice, is the one-bit quantization, i.e., $e = 2$. With the additional conditions $k_1 = -1$, $k_2 = 1$, and $\tau_1 = 0$, the quantization function $\mathcal{Q}$ becomes the element-wise signum function.

## 1.1 RELATED WORKS

Parameter estimation from quantized signals has been extensively studied. Among the many directions, two fundamental lines of inquiry concern the estimation of the mean and the covariance of $\boldsymbol{x}$, which we take as our starting point. Furthermore, we will demonstrate that our proposed framework is versatile and can be extended to various related tasks, including quantized regression, quantized matrix completion, and quantized compressed sensing.

**Quantized mean estimation** In Papadopoulos et al. (2001), in order to estimate signals from wireless sensor networks, an optimization problem is formulated for estimating the mean under a one-dimensional Gaussian distribution assumption, based on multi-bit quantized data. To address this optimization problem, the authors propose an ECM-based algorithm. Subsequently, Ribeiro & Giannakis (2006b) focuses on the specific scenario of one-bit quantization under a Gaussian distribution and derives a closed-form expression for mean estimation in this case. Further, Fang & Li (2008) extends this line of research by introducing an adaptive threshold into the one-bit quantization function, thereby improving mean estimation accuracy. In addition, Dabeer & Masry (2008) generalizes the problem to the multidimensional setting. Ribeiro & Giannakis (2006a) extends the model distribution to the generalized Gaussian distribution under the one-dimensional assumption.

**Quantized covariance/correlation estimation** In Van Vleck & Middleton (1966), the authors investigate the estimation of a correlation matrix from one-bit quantized zero-mean Gaussian measurements, based on the arcsine law (Lévy, 1940). However, in such one-bit zero-mean settings, the individual variance of each dimension cannot be determined, which prevents recovery of the full covariance matrix. To estimate the variance, the "dithering technique" is introduced, which is to set a threshold in the signum function (Liu & Lin, 2021). Based on the dithering technique, the variance can be obtained in closed form (Fang & Li, 2008), but the correlation matrix should be solved analytically. Consequently, subsequent studies (Dirksen et al., 2022; Eamaz et al., 2023; Xiao et al., 2023; Liu & Chou, 2025) have proposed various strategies for correlation estimation. The estimation approaches for correlation can be classified into two categories. The first category relies on the correlation coefficient function of the one-bit Gaussian distribution, with analyses usually restricted to pairwise interactions between dimensions in the multivariate setting. Since this function is generally computationally intractable, different approximate functions have been proposed to compute the correlation coefficient (Eamaz et al., 2023; Xiao et al., 2023; Liu & Chou, 2025). In Eamaz et al. (2022; 2023), the authors introduce a known, time-varying threshold for the signum function and propose a modified arcsine law to express correlation coefficients as integrals, which are then approximated using Gauss–Legendre quadratures. The method in (Xiao et al., 2023) applies maximum likelihood (ML) estimation to compute correlation coefficients, which is equivalent to iteratively evaluating their approximate functions. The work (Liu & Chou, 2025) represents the correlation coefficient function as an infinite series via the one-bit Hermite law (Liu & Lin, 2021) and then approximates it using harmonic approximation. The second category (Dirksen et al., 2022; Chen et al., 2024) directly employs the sample covariance matrix computed using multiple thresholds. These methods are not restricted to one-bit quantization and can be generalized to multi-bit cases, though typically at the expense of reduced estimation accuracy.

## 1.2 CONTRIBUTION

In this paper, we study the maximum likelihood (ML) estimation of the parameters of a random signal $\boldsymbol{x}$ with quantization. We assume that $\boldsymbol{x}$ follows a general normal mean–variance mixture model (Barndorff-Nielsen et al., 1982), which is a flexible family encompassing several well-known distributions, including Gaussian, $t$, generalized hyperbolic skew-$t$ (GHST), hyperbolic, and generalized hyperbolic (GH) distributions. Unlike previous approaches that are restricted to one-bit or Gaussian settings, our framework accommodates arbitrary bit quantization functions and any distribution within the normal mean–variance mixture family. To solve the ML estimation problem, we develop an expectation conditional maximization (ECM) algorithm. The method alternates between two steps: (i) an expectation step (E-step), where we construct a surrogate of the likelihood via Jensen's inequality, and (ii) a conditional maximization step (CM-step), where the surrogate is maximized with respect to a subset of parameters while holding the others fixed. This procedure yields closed-form updates for the location, skewness, and scatter parameters. The proposed ECM algorithm inherits the interpretability and stability of the ECM framework, while we further establish that it converges globally at a linear rate.

## 2 PROBLEM FORMULATION

A random variable $\boldsymbol{x} \in \mathbb{R}^d$ following a normal mean-variance mixture model is represented by

$$\boldsymbol{x} = \boldsymbol{\mu} + z\boldsymbol{\xi} + (z\boldsymbol{\Sigma})^{\frac{1}{2}} \boldsymbol{\epsilon}, \tag{3}$$

where $\boldsymbol{\mu}$ is the location parameter, $\boldsymbol{\xi}$ is the skewness parameter, $\boldsymbol{\Sigma}$ is the scatter parameter, $z$ is a nonnegative random variable with density function $p(z)$, and $\boldsymbol{\epsilon}$ is a standard normal random variable with mean zero and covariance matrix identity. $z$ and $\boldsymbol{\epsilon}$ are independent from each other. The mean and the covariance matrix of $\boldsymbol{x}$ are computed as $\boldsymbol{\mu} + \mathsf{E}[z]\boldsymbol{\xi}$ and $\left(\mathsf{E}[z^2] - \mathsf{E}[z]^2\right)\boldsymbol{\xi}\boldsymbol{\xi}^\top + \mathsf{E}[z]\boldsymbol{\Sigma}$, respectively. The conditional density function of $\boldsymbol{x}$ given $z$ is

$$p(\boldsymbol{x} \mid z; \boldsymbol{\theta}) = \frac{1}{(2\pi)^{\frac{d}{2}} \det(z\boldsymbol{\Sigma})^{\frac{1}{2}}} \exp\left(-\frac{1}{2z} \|\boldsymbol{x} - \boldsymbol{\mu} - z\boldsymbol{\xi}\|_{\boldsymbol{\Sigma}^{-1}}^2\right), \tag{4}$$

where $\boldsymbol{\theta} = \left[\boldsymbol{\mu}^\top, \boldsymbol{\xi}^\top, \mathrm{vec}\left(\boldsymbol{\Sigma}\right)^\top\right]^\top$ and $\|\boldsymbol{x}\|_{\boldsymbol{A}} = \boldsymbol{x}^\top \boldsymbol{A}\boldsymbol{x}$. Following the definitions of $\boldsymbol{y}$ in (1), the density function of $\boldsymbol{y}$ is given as follows:

$$p(\boldsymbol{y}; \boldsymbol{\theta}) = \int_{\mathcal{Q}^{-1}(\boldsymbol{y})} p(\boldsymbol{x}; \boldsymbol{\theta})\mathrm{d}\boldsymbol{x} = \int_{\mathcal{Q}^{-1}(\boldsymbol{y})} \int_0^\infty p(\boldsymbol{x} \mid z; \boldsymbol{\theta})p(z)\mathrm{d}z\mathrm{d}\boldsymbol{x}, \tag{5}$$

where $\mathcal{Q}^{-1}(\boldsymbol{y})$ maps the quantized data $\boldsymbol{y}$ into a hyper-rectangle whose projection on each dimension $i$ is an interval corresponding to $[\mathcal{Q}(\boldsymbol{y})]_i$. Given $n$ independent and identically distributed samples $\boldsymbol{y}_1, \ldots, \boldsymbol{y}_n$, the ML estimation problem is given by

$$\max_{\boldsymbol{\theta}} \quad L(\boldsymbol{\theta}) = \sum_{t=1}^n \log p(\boldsymbol{y}_t; \boldsymbol{\theta}). \tag{6}$$

When $d = 1$ and $z$ is deterministic, i.e., $\boldsymbol{x}$ is a univariate normal random variable, and one-bit quantization is applied, the estimation problem admits a closed-form solution (Ribeiro & Giannakis, 2006b). In contrast, as long as $d \geq 2$, the density function in (5) involves multiple integrals, making the optimization problem substantially more challenging.

## 3 THE ECM ALGORITHM

The optimization problem in (6) is challenging due to the multiple integrals in the density function (5). Nevertheless, the ECM framework can be applied by leveraging the models in (1) and (3), where $\boldsymbol{x}$ and $z$ can be naturally treated as latent variables. In the following, we introduce an ECM procedure for (6).

### 3.1 E-STEP

In the E-step, we derive a surrogate function for the log-likelihood function $L(\boldsymbol{\theta})$. Based on (1), the observed signal $\boldsymbol{y}$ is conditioned on the hidden variable $\boldsymbol{x}$. Hence, given $[\boldsymbol{y}_1 \ldots, \boldsymbol{y}_n]^\top$ and the corresponding hidden variables $[\boldsymbol{x}_1 \ldots, \boldsymbol{x}_n]^\top$, we have[1]

$$L(\boldsymbol{\theta}) = \sum_{t=1}^{n} \log \int_{\mathbb{R}} p(\boldsymbol{y}_t \mid \boldsymbol{x}_t; \boldsymbol{\theta}) p(\boldsymbol{x}_t; \boldsymbol{\theta}) \mathrm{d}\boldsymbol{x}_t \geq \sum_{t=1}^{n} \mathsf{E}_{\boldsymbol{x}_t|\boldsymbol{y}_t;\underline{\boldsymbol{\theta}}} \log p(\boldsymbol{y}_t, \boldsymbol{x}_t; \boldsymbol{\theta}) + \mathrm{const.}, \quad (7)$$

where the inequality is based on Jensen's inequality and $\mathrm{const.}$ is some constant term independent of $\boldsymbol{\theta}$. Under the models specified in (1), the joint density function is given by

$$p(\boldsymbol{x}_t, \boldsymbol{y}_t; \boldsymbol{\theta}) = p(\boldsymbol{x}_t; \boldsymbol{\theta}) \delta \left( \mathcal{Q}\left(\boldsymbol{x}_t\right) - \boldsymbol{y}_t \right), \quad (8)$$

where $\delta(\cdot)$ denotes the multivariate Dirac delta function, defined as $\delta\left(\boldsymbol{x}_t\right) = \begin{cases} 0, & \text{if } \boldsymbol{x}_t \neq \boldsymbol{0} \\ +\infty, & \text{if } \boldsymbol{x}_t = \boldsymbol{0} \end{cases}$.

Since the term $\delta\left(\mathcal{Q}\left(\boldsymbol{x}_t\right) - \boldsymbol{y}_t\right)$ is independent of $\boldsymbol{\theta}$, we further obtain

$$L(\boldsymbol{\theta}) \geq \sum_{t=1}^{n} \mathsf{E}_{\boldsymbol{x}_t|\boldsymbol{y}_t;\underline{\boldsymbol{\theta}}} \log p(\boldsymbol{x}_t; \boldsymbol{\theta}) + \mathrm{const.} = \sum_{t=1}^{n} \mathsf{E}_{\boldsymbol{x}_t|\boldsymbol{y}_t;\underline{\boldsymbol{\theta}}} \log \int_0^\infty p(\boldsymbol{x}_t, z_t; \boldsymbol{\theta}) \mathrm{d}z_t + \mathrm{const.}$$

Regarding $z$ as a hidden variable, applying Jensen's inequality leads to

$$L(\boldsymbol{\theta}) \geq \sum_{t=1}^{n} \mathsf{E}_{\boldsymbol{x}_t|\boldsymbol{y}_t;\underline{\boldsymbol{\theta}}} \left[ \mathsf{E}_{z_t|\boldsymbol{x}_t;\underline{\boldsymbol{\theta}}} \log p(\boldsymbol{x}_t, z_t; \boldsymbol{\theta}) \right] + \mathrm{const.} \triangleq S(\boldsymbol{\theta}; \underline{\boldsymbol{\theta}}). \quad (9)$$

Since $z \mid \boldsymbol{x}$ is independent of $\boldsymbol{x} \mid \boldsymbol{y}$, we have $p(\boldsymbol{x} \mid \boldsymbol{y}; \boldsymbol{\theta}) p(z \mid \boldsymbol{x}; \boldsymbol{\theta}) = p(\boldsymbol{x}, z \mid \boldsymbol{y}; \boldsymbol{\theta})$. Given the $p(\boldsymbol{x} \mid z; \boldsymbol{\theta})$ in (4), the surrogate function $S(\boldsymbol{\theta}; \underline{\boldsymbol{\theta}})$ can be further expressed as follows:

$$S(\boldsymbol{\theta}; \underline{\boldsymbol{\theta}}) = \sum_{t=1}^{n} \left[ \mathsf{E}_{z_t|\boldsymbol{y}_t;\underline{\boldsymbol{\theta}}} \left[ \log p(z_t) \right] - \frac{1}{2} \log \det \boldsymbol{\Sigma} \right. \quad (10)$$

$$\left. - \frac{1}{2} \mathrm{Tr} \left( \left(\boldsymbol{U}_t - 2\boldsymbol{v}_t \boldsymbol{\mu}^\top + \iota_t \boldsymbol{\mu}\boldsymbol{\mu}^\top \right) - 2(\boldsymbol{w}_t - \boldsymbol{\mu})\boldsymbol{\xi}^\top + \zeta_t \boldsymbol{\xi}\boldsymbol{\xi}^\top \right) \boldsymbol{\Sigma}^{-1} \right] + \mathrm{const.},$$

where

$$\boldsymbol{U}_t = \mathsf{E}_{\boldsymbol{x}_t, z_t|\boldsymbol{y}_t;\underline{\boldsymbol{\theta}}}[z^{-1}\boldsymbol{x}\boldsymbol{x}^\top], \quad \boldsymbol{v}_t = \mathsf{E}_{\boldsymbol{x}_t, z_t|\boldsymbol{y}_t;\underline{\boldsymbol{\theta}}}[z_t^{-1}\boldsymbol{x}_t], \quad \boldsymbol{w}_t = \mathsf{E}_{\boldsymbol{x}_t|\boldsymbol{y}_t;\underline{\boldsymbol{\theta}}}[\boldsymbol{x}_t],$$

$$\iota_t = \mathsf{E}_{z_t|\boldsymbol{y}_t;\underline{\boldsymbol{\theta}}}[z_t^{-1}], \quad\quad\quad \zeta_t = \mathsf{E}_{z_t|\boldsymbol{y}_t;\underline{\boldsymbol{\theta}}}[z_t].$$

The details for computing these expectations are given in Appendix A. In the term $\mathsf{E}_{z_t|\boldsymbol{y}_t;\underline{\boldsymbol{\theta}}} \left[ \log p(z_t) \right]$, some scalar parameters (such as the shape parameter $\nu$ in Student's $t$ distribution) are contained. In the practical implementation (Galarza et al., 2021), they are typically treated as given or estimated through the one-dimensional search.

### 3.2 CM-STEP

Based on the surrogate function (10), in the CM-step, we solve the following optimization problem:

$$\max_{\boldsymbol{\theta}} \ S(\boldsymbol{\theta}; \underline{\boldsymbol{\theta}}), \quad (11)$$

where parameters $\boldsymbol{\mu}$, $\boldsymbol{\xi}$, and $\boldsymbol{\Sigma}$ can be solved with closed-form solutions:

$$\boldsymbol{\mu} = \frac{\sum_{t=1}^{n}(\boldsymbol{v}_t - \boldsymbol{\xi})}{\sum_{t=1}^{n} \iota_t}, \quad\quad \boldsymbol{\xi} = \frac{\sum_{t=1}^{n}(\boldsymbol{w}_t - \boldsymbol{\mu})}{\sum_{t=1}^{n} \zeta_t},$$

$$\boldsymbol{\Sigma} = \frac{1}{n} \sum_{t=1}^{n} \left( \left(\boldsymbol{U}_t - 2\boldsymbol{v}_t\boldsymbol{\mu}^\top + \iota_t \boldsymbol{\mu}\boldsymbol{\mu}^\top \right) - 2(\boldsymbol{w}_t - \boldsymbol{\mu})\boldsymbol{\xi}^\top + \zeta_t \boldsymbol{\xi}\boldsymbol{\xi}^\top \right). \quad (12)$$

In the context of quantization model estimation, a prototypical scenario is the one-bit Gaussian case (i.e., $e = 1$ and $z$ is a constant). However, this model suffers from an inherent identifiability issue: for each dimension $i = 1, \ldots, d$, only the ratio $\frac{\mu_i}{\sigma_i^2}$ can be determined, rather than the individual

---

[1]Throughout this paper, underlined variables denote those whose values are given as constants.

parameters. Hence, the estimated values of $\mu_i$ and $\sigma_i^2$ differ from their desirable values by an arbitrary scaling factor. To resolve this, existing estimation methods in the one-bit Gaussian setting typically fix one parameter (either the location vector or the scatter matrix) to identify the other. Moreover, the same ambiguity persists under one-bit quantization whenever $x$ follows any elliptical distribution.

## 4 CONVERGENCE ANALYSIS

In this section, we analyze the convergence properties of the proposed ECM algorithm. The ECM algorithm is guaranteed to converge to a specific stationary point given an initial point (McLachlan & Krishnan, 2008). Define $\boldsymbol{\theta}^{\star} = \left[ \boldsymbol{\mu}^{\star\top}, \boldsymbol{\xi}^{\star\top}, \operatorname{vec}\left(\boldsymbol{\Sigma}^{\star}\right)^{\top} \right]^{\top}$ as a stationary point of the optimization problem in (6). We first establish the convergence of each parameter individually, while keeping the other two fixed, i.e., the convergence property of the conditional maximization step.

**Proposition 1.** *Denote the sequence generated by the ECM algorithm as $\left\{\boldsymbol{\theta}^{(k)}\right\}$. For any $k \geq 0$, we have*

$$\|\boldsymbol{\mu}^{(k+1)} - \boldsymbol{\mu}^{\star}\|_2 \leq c_\mu \|\boldsymbol{\mu}^{(k)} - \boldsymbol{\mu}^{\star}\|_2,$$

$$\|\boldsymbol{\xi}^{(k+1)} - \boldsymbol{\xi}^{\star}\|_2 \leq c_\xi \|\boldsymbol{\xi}^{(k)} - \boldsymbol{\xi}^{\star}\|_2,$$

$$\|\boldsymbol{\Sigma}^{(k+1)} - \boldsymbol{\Sigma}^{\star}\|_{\mathsf{F}} \leq c_\Sigma \|\boldsymbol{\Sigma}^{(k)} - \boldsymbol{\Sigma}^{\star}\|_{\mathsf{F}},$$

*where*

$$c_\mu = \max_{\boldsymbol{\theta}} \left\| \frac{\sum_{t=1}^n \boldsymbol{\Sigma}^{-1} \mathsf{Cov}_{\boldsymbol{x}_t, z_t | \boldsymbol{y}_t; \boldsymbol{\theta}} \left[ z_t^{-1} \boldsymbol{x}_t \right]}{\sum_{t=1}^n \mathsf{E}_{z | \boldsymbol{y}_t; \boldsymbol{\theta}} [z_t^{-1}]} \right\|_2 \in (0, 1),$$

$$c_\xi = \max_{\boldsymbol{\theta}} \left\| \frac{\sum_{t=1}^n \boldsymbol{\Sigma}^{-1} \mathsf{Cov}_{\boldsymbol{x}_t, z_t | \boldsymbol{y}_t; \boldsymbol{\theta}} \left[ \boldsymbol{x}_t - z_t \boldsymbol{\xi} \right]}{\sum_{t=1}^n \mathsf{E}_{z_t | \boldsymbol{y}_t; \boldsymbol{\theta}} [z_t]} \right\|_2 \in (0, 1),$$

$$c_\Sigma = \max_{\boldsymbol{\theta}} \left\| \frac{1}{2n} \sum_{t=1}^n \left( \boldsymbol{\Sigma}^{-1} \otimes \boldsymbol{\Sigma}^{-1} \right) \mathsf{Cov}_{\boldsymbol{x}_t, z_t | \boldsymbol{y}_t; \boldsymbol{\theta}} \left[ z_t^{-1} \operatorname{vec}\left( \boldsymbol{x}_t \boldsymbol{x}_t^{\top} \right) \right] \right\|_2 \in (0, 1).$$

Based on the result in Proposition 1 and Meng & Rubin (1994), we establish that the proposed ECM algorithm converges globally at a linear rate, as detailed in the following result.

**Theorem 2.** *Denote the sequence generated by the ECM algorithm as $\left\{\boldsymbol{\theta}^{(k)}\right\}$. For any $k \geq 0$, we have*

$$\|\boldsymbol{\theta}^{(k)} - \boldsymbol{\theta}^{\star}\|_2 \leq c^k \|\boldsymbol{\theta}^{(0)} - \boldsymbol{\theta}^{\star}\|_2, \tag{13}$$

*where $c = \max\{c_\mu, c_\xi, c_\Sigma\} \in (0, 1)$.*

## 5 EXTENDED APPLICATIONS

In the previous sections, an algorithm to the ML estimation problem was proposed for modeling quantized data by a normal mean-variance mixture model. Modeling data with a probabilistic model and then performing ML estimation is a common practice in the field of machine learning. Hence, the proposed algorithm can be readily extended to typical machine learning applications from quantized data. In this section, two applications of the proposed model and estimation methods to quantized matrix completion and quantized compressive sensing will be introduced, and experimental validations will be presented in a subsequent section.

### 5.1 QUANTIZED MATRIX COMPLETION

The goal of the low-rank matrix completion problem (Candes & Recht, 2012) is to recover an unknown low-rank matrix $\boldsymbol{M} \in \mathbb{R}^{d_1 \times d_2}$ from an observed, yet incomplete, matrix. Let $\boldsymbol{X}$ denote a matrix whose entries are drawn from a normal mean–variance mixture distribution. We use $\mu_{ij}$ and $x_{ij}$ to represent the $(i, j)$-th entries of $\boldsymbol{M}$ and $\boldsymbol{X}$, respectively. Based on the normal mean-variance model, the relationship between $\mu_{ij}$ and $x_{ij}$ is given by

$$x_{ij} = \mu_{ij} + z\xi + z^{1/2}\sigma\epsilon, \tag{14}$$

where $\mu_{ij}$ serves as the location parameter of $x_{ij}$. In the quantization scenario (Davenport et al., 2014), however, the matrix $\boldsymbol{X}$ is not directly accessible; instead, we observe its quantized counterpart $\boldsymbol{Y} = \mathcal{Q}(\boldsymbol{X})$. Moreover, the entire matrix $\boldsymbol{Y}$ is not available for observation. Let $\boldsymbol{\Omega}$ denote the index set of the observed entries; specifically, if $(i, j) \in \boldsymbol{\Omega}$, then $Y_{ij}$ is observed, otherwise $Y_{ij}$ is missing. Our goal is to recover $\boldsymbol{M}$ from incomplete $\boldsymbol{Y}$. The study of matrix completion was popularized following the Netflix Million Dollar Challenge, which posed the task: accurately predicting the values of those entries with a user–item matrix in which entries represent item ratings.

To recover $\boldsymbol{M}$ from quantized and corrupted observations, a seminal study introduced an ML framework for one-bit low-rank matrix completion (Davenport et al., 2014). Building on this work, random dithering was incorporated into the quantization function to improve recovery performance (Eamaz et al., 2024). Both approaches employed the nuclear norm to relax the low-rank constraint and used projected gradient descent for optimization. An alternative strategy reformulated the low-rank constraint via matrix factorization, followed by projected gradient descent (Bhaskar & Javanmard, 2015). Similarly, factorization combined with a majorization–minimization method was used to derive a surrogate objective, which was solved via the Gauss–Newton method (Liu et al., 2025). The framework was further extended from one-bit to multi-bit quantization using low-rank factorization together with projected gradient descent (Bhaskar, 2016).

## 5.2 Quantized Compressive Sensing

The one-bit compressive sensing (Boufounos & Baraniuk (2008b)) aims to estimate a sparse signal lying in a known measurement subspace based on observed quantized data (Jacques et al., 2013; Chen et al., 2024). Existing methods for this problem generally fall into three categories. The first two primarily focus on 1-bit quantization without incorporating additive noise into the original measurements (Li et al., 2018). The first category, termed regularizer-class algorithms (Laska et al., 2011), introduces additional regularization terms to the classical compressive sensing recovery problem to enforce consistency between the sparse signal and the measurements. The second category, known as penalty-class algorithms (Yan et al., 2012), models sign flips between quantized and recovered measurements as penalty terms. The third category assumes the presence of additive noise in the original measurements (Zymnis et al., 2009; Knudson et al., 2016; Shao et al., 2024). Our proposed model is developed as a further extension of the third approach. To address potential outliers in the additive noise and enhance the robustness of the model, we employ a normal mean–variance mixture model for noise modeling. Denote $\boldsymbol{\Phi} \in \mathbb{R}^{d_1 \times d_2}$ as the known measurement matrix, and $\boldsymbol{\vartheta} \in \mathbb{R}^{d_2}$ as the sparse signal to be estimated. Based on the normal mean-variance model, the quantized compressed sensing model is given by

$$\boldsymbol{y} = \mathcal{Q}(\boldsymbol{x}), \quad \boldsymbol{x} = \boldsymbol{\Phi}\boldsymbol{\vartheta} + z\boldsymbol{\xi} + z^{1/2}\sigma\boldsymbol{\epsilon}, \tag{15}$$

where $\boldsymbol{\Phi}\boldsymbol{\vartheta}$ serves as the location parameter of $\boldsymbol{x}$.

# 6 Experiments

## 6.1 Quantized Covariance/Correlation Estimation

In this section, we address the problem of one-bit quantized covariance estimation. Existing approaches can be broadly categorized as follows: non-dithered methods, which include the arcsine law method with a zero threshold (Van Vleck & Middleton, 1966) (Zero Threshold), and methods utilizing a non-zero threshold, such as one-bit autocorrelation estimation (Liu & Lin, 2021) (One-bit Autocorrelation), the one-bit Hermite law (Liu & Chou, 2025) (One-bit Hermite Law), and the covariance matrix recovery method (Xiao et al., 2023) (One-bit MLE). Another category is dithered methods, where the dithering signal follows either a uniform distribution (Dirksen et al., 2022) (Dithering Threshold) or a Gaussian distribution (Eamaz et al., 2022) (One-bit Time-varying); the dithering threshold can also be adaptive based on the data (Dirksen & Maly, 2024) (Adaptive Dithering). For the multi-bit problem, there are the multi-bit covariance estimator (Multi-bit Estimator) and the multi-bit parameter-free estimator (Parameter-free Estimator) (Chen & Ng, 2025).

We begin by comparing the estimation accuracy of our method against existing approaches since the previous work could only estimate the correlation matrix of one-bit data with zero mean. We generated samples from a Student's $t$ distribution with $\nu = 3$ degrees of freedom (Figure 2). We

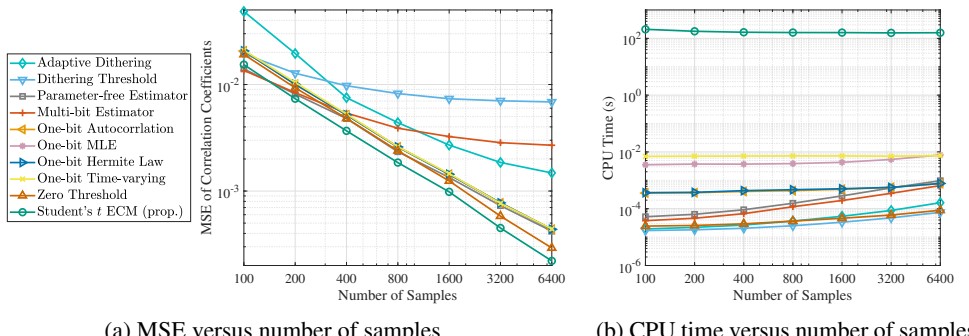

(a) MSE versus number of samples        (b) CPU time versus number of samples

Figure 1: Algorithm performance comparison for the Student's $t$ distribution

estimated the correlation matrix using the following methods: the Student's $t$ ECM, the Gaussian ECM, the Parameter-free Estimator, and the Multi-bit Estimator. We also compared the results against those obtained by applying the EM algorithm to perform MLE on the unquantized data (Unquantized MLE). The proposed Student's $t$ MLE and Gaussian MLE show that as the number of bits increases, the MSE of estimating correlation will decrease and approach the result of the MLE without quantization. However, as the number of bits increases, the time required by the proposed algorithms will also increase.

Figures 3 and 4 present a comparative analysis of the convergence rates of the algorithm across four statistical distributions: Gaussian, Student's $t$, GHST, and GH. The results are organized into four subfigures (a–d), each corresponding to one distribution. The convergence is measured by the distance to the optimum, defined as $\|x^{(t)} - x^\star\|_2$ for vectors $x$ and $x^\star$, and as $\|X^{(k)} - X^\star\|_{\mathsf{F}}$ for matrices $X$ and $X^\star$. All proposed methods achieve linear convergence rates. The numerical values of these rates, which vary per iteration, are detailed in Figure 4. Finally, Table 1 reports experimental results where samples are randomly generated from the GH distribution with parameters $\lambda = -0.5$, $\delta = 2$, and $\gamma = 2$, with $e = 1$ and threshold $\tau = 2$. The MSE of the correlation matrix estimated via the GH method is consistently lower than that obtained by alternative approaches.

Table 1: Correlation MSE comparison for different methods under the synthetic data

| Method | $10^2$ samples | $10^3$ samples | $10^4$ samples |
|---|---|---|---|
| Gaussian ECM | $4.7960 \times 10^{-2}$ | $7.5326 \times 10^{-3}$ | $4.9231 \times 10^{-3}$ |
| Student's $t$ ECM | $5.8602 \times 10^{-2}$ | $9.5297 \times 10^{-3}$ | $6.6826 \times 10^{-3}$ |
| GHST ECM | $1.5503 \times 10^{-1}$ | $1.0318 \times 10^{-1}$ | $9.5506 \times 10^{-2}$ |
| GH ECM | $\mathbf{4.5377 \times 10^{-2}}$ | $\mathbf{7.3532 \times 10^{-4}}$ | $\mathbf{9.0011 \times 10^{-4}}$ |

## 6.2 QUANTIZED MATRIX COMPLETION

Given (14) and $Y = \mathcal{Q}(X)$, the ML estimation problem for $M$ is given by

$$\max_{M} \quad \sum_{(i,j)\in\Omega} \log p\left(y_{ij} \mid \mu_{ij}\right) \tag{16}$$
$$\text{s.t.} \quad \text{rank}(M) \leq r.$$

By applying a low-rank factorization to the matrix $M$, we express it as $M = AB^\top$, where $A \in \mathbb{R}^{d_1 \times r}$ and $B \in \mathbb{R}^{d_2 \times r}$. Through the E-step in our proposed ECM algorithm, we have the surrogate function of (16) as

$$\sum_{(i,j)\in\Omega} \left(a_i b_j^\top - e_{ij}\right)^2, \tag{17}$$

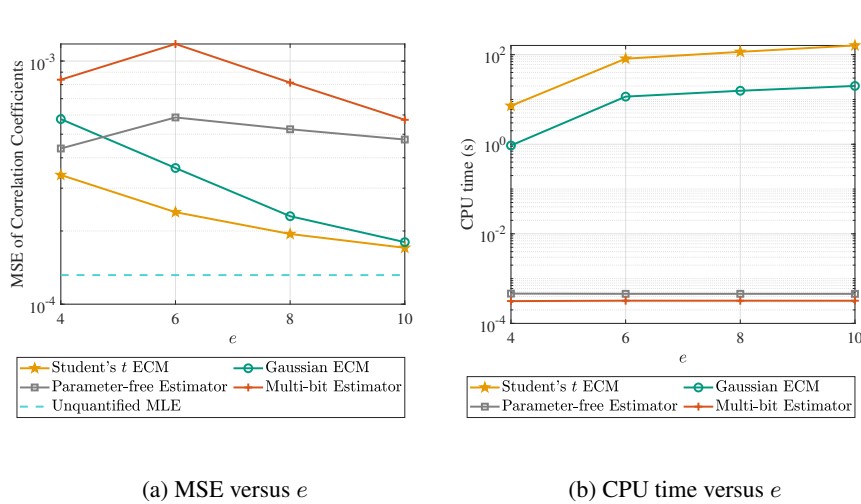

(a) MSE versus $e$

(b) CPU time versus $e$

Figure 2: Algorithms performance comparison versus the number of bits

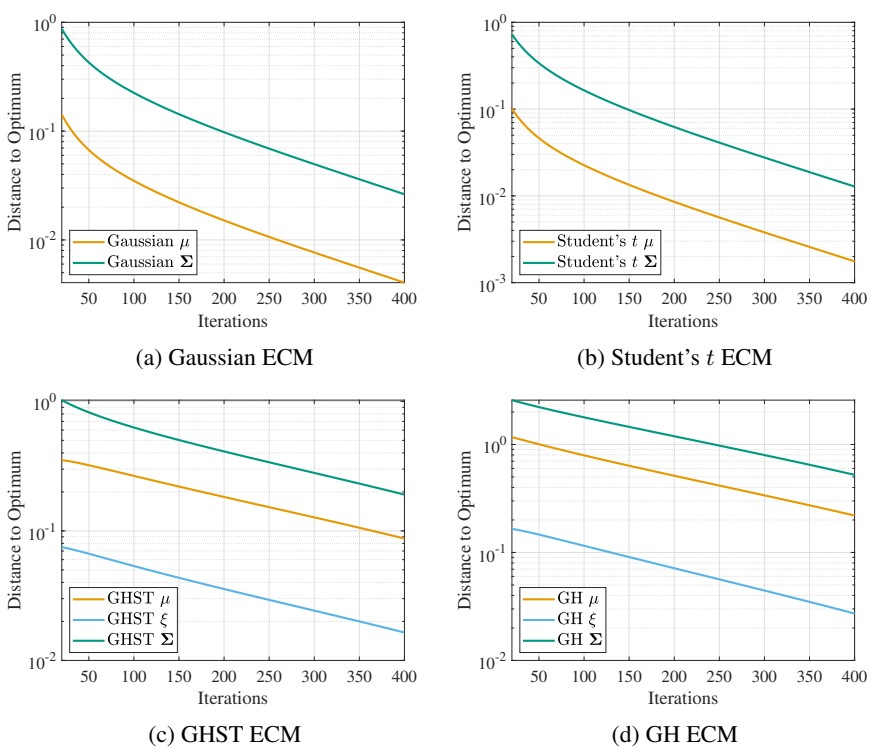

(a) Gaussian ECM

(b) Student's $t$ ECM

(c) GHST ECM

(d) GH ECM

Figure 3: Convergence rate versus iterations

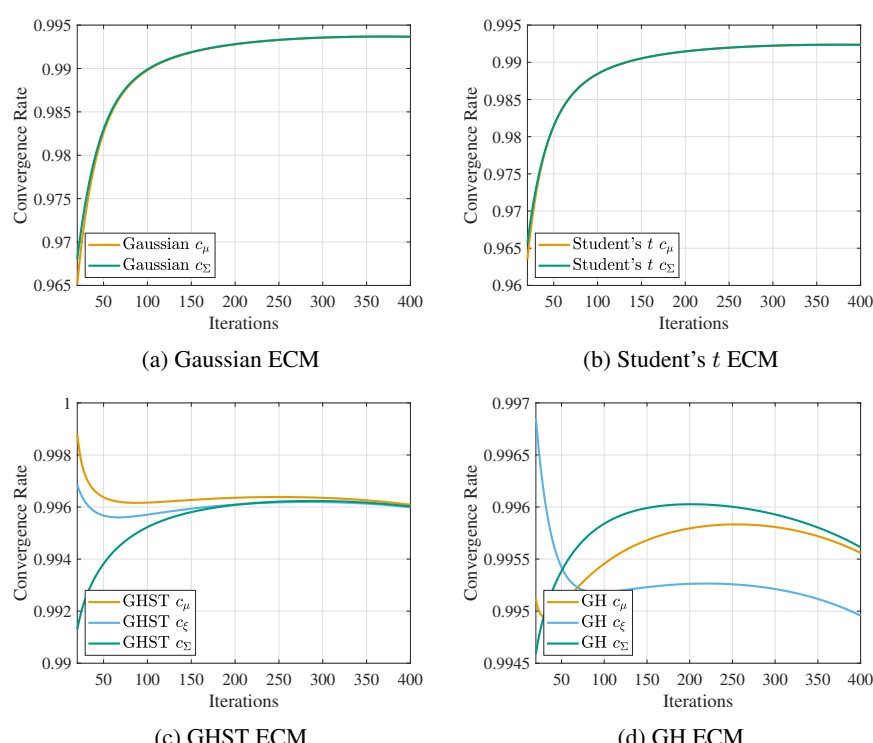

Figure 4: Convergence rate versus iterations

where $\boldsymbol{a}_i$ and $\boldsymbol{b}_i$ are $i$-th row of the matrix $\boldsymbol{A}$ and $\boldsymbol{B}$, respectively, and $e_{ij} = \frac{v_{ij}-\xi}{\iota_{ij}}$. The details of obtaining the surrogate (17) are given in the Appendix D. Then we can use the ECM algorithm to alternatively update $\boldsymbol{A}$ and $\boldsymbol{B}$.

For the experimental design, the MovieLens 1M dataset (Harper & Konstan, 2015) is employed, which contains 1000000 movie ratings provided by 6040 users for 3952 movies, with each rating ranging from 1 to 5. All ratings are randomly partitioned into training and test sets, accounting for 40% and 60% of the data, respectively. The training set is used to predict the completed rating matrix, and the entries overlapping between the completed rating matrix and the test set are then compared to evaluate prediction accuracy and root mean square error (RMSE), which are defined in Appendix E.

Existing methods encompass classic machine learning approaches for matrix completion, including the singular value decomposition (SVD (Sarwar et al., 2000)) model, the $\ell_2$-regularized matrix factorization ($\ell_2$-regularized (Paterek, 2007)) model, and the nuclear norm minimization (nuclear norm (Cai et al., 2010)) model. Furthermore, several approaches are established upon random variable model assumptions. These include: directly modeling the Gaussian without quantization (Gaussian (Candes & Plan, 2010)); 1-bit quantization with a Gaussian distribution (1-bit Gaussian (Davenport et al., 2014)); and multi-bit quantization under a Gaussian assumption (multi-bit Gaussian (Bhaskar, 2016)). Our proposed method is regarded as an extension of the multi-bit Gaussian approach by extending the Gaussian assumption to a normal mean-variance mixture model, comprising Student's $t$, GHST, and GH distributions.

The results are demonstrated in Table 2. As shown in the table, the proposed method consistently achieves superior performance in terms of both accuracy and RMSE. Among the evaluated models, the approach based on the GH distribution attains the best results, attributable to its flexibility and robustness in modeling skewness and heavy tails. Details regarding the experimental parameter settings are provided in the Appendix E.

Table 2: Accuracy and RMSE comparisons of matrix completion on MovieLens 1M

| Method | Accuracy | RMSE | Time (s) |
|---|---|---|---|
| SVD | 0.4261 | 0.9148 | 364.36 |
| L2-regularization | 0.4388 | 0.9355 | 165.08 |
| Nuclear Norm | 0.3802 | 1.1662 | 867.39 |
| Gaussian | 0.4363 | 0.9486 | 25.26 |
| 1-bit Gaussian | 0.4217 | 0.9659 | 110.44 |
| Multi-bit Gaussian | 0.4453 | 0.9151 | 90.84 |
| Multi-bit Student's $t$ ECM (prop.) | 0.4464 | 0.9091 | 232.16 |
| Multi-bit GHST ECM (prop.) | 0.4495 | 0.9076 | 252.23 |
| Multi-bit GH ECM (prop.) | **0.4510** | **0.8892** | 312.32 |

## 6.3 QUANTIZED COMPRESSIVE SENSING

Given (15) and the $\ell_1$ regularization model from Zymnis et al. (2009), we have the optimization problem

$$\max_{\boldsymbol{\vartheta}} \quad \log p\left(\boldsymbol{y} \mid \boldsymbol{\vartheta}\right) + \eta \|\boldsymbol{\vartheta}\|_1. \tag{18}$$

With the E-step in our proposed ECM algorithm, we have the surrogate function of (18) as

$$\frac{1}{2} \left\|\boldsymbol{\Phi}\boldsymbol{\vartheta} - \boldsymbol{e}\right\|_2^2 + \eta \|\boldsymbol{\vartheta}\|_1,$$

where the details of deriving the surrogate function are given in the Appendix D. We can use the FISTA algorithm (Beck & Teboulle, 2009) to solve the surrogate.

Performance comparisons are conducted using synthetic data, where the measurement dimension is denoted as $d_1 = 3000$ and the 30-sparse signal dimension as $d_2 = 1000$. To evaluate algorithmic robustness, an additive noise term exhibiting both skewed and heavy-tailed characteristics is introduced to the original measurements. The existing methods included in our comparison consist of restricted step shrinkage (RSS) (Laska et al., 2011), adaptive outlier pursuit (AOP) (Yan et al., 2012), and 1-bit Gaussian MLE (Zymnis et al., 2009). For each signal-to-noise ratio (SNR) level, experiments are repeated 10 times, and the average cosine similarity (Cos Sim) and computational time are reported. The results are summarized in the Table 3, which demonstrates that the 1-bit GH ECM algorithm achieves the largest average cosine similarity. The definitions of SNR and cosine similarity are given in Appendix E.

Table 3: Cosine similarity comparisons of 1-bit compressive sensing under different SNR levels.

| Method | SNR = 0 dB | | SNR = -5 dB | | SNR = -10 dB | |
|---|---|---|---|---|---|---|
| | Cos Sim | Time (s) | Cos Sim | Time (s) | Cos Sim | Time (s) |
| RSS | 0.7872 | 0.2639 | 0.6598 | 0.2651 | 0.3027 | 0.2546 |
| AOP | 0.7052 | 0.2689 | 0.4041 | 0.2951 | 0.2209 | 0.2738 |
| 1-bit Gaussian | 0.8220 | 2.7704 | 0.5487 | 3.0875 | 0.2896 | 2.7202 |
| 1-bit Student's $t$ ECM (prop.) | 0.8396 | 2.8573 | 0.5678 | 3.1643 | 0.3031 | 3.3174 |
| 1-bit GHST ECM (prop.) | 0.8893 | 2.7435 | 0.7701 | 3.4204 | 0.5237 | 4.0281 |
| 1-bit GH ECM (prop.) | **0.8915** | 2.7982 | **0.7717** | 3.3562 | **0.5407** | 3.9535 |

## 7 CONCLUSION AND DISCUSSION

In this paper, we propose an ECM-based algorithm for parameter estimation in quantized models. The method exhibits excellent scalability, handling problems from one-bit to multi-bit quantization and accommodating distributions ranging from Gaussian to the broader class of normal mean–variance mixtures. Experiments show that our approach yields more accurate estimates than existing methods in certain cases (e.g., the one-bit Gaussian setting), while maintaining high accuracy across a wide range of extended scenarios. Furthermore, it demonstrates strong potential in the typical machine learning tasks including quantized matrix completion and quantized compressive sensing.

## ETHICS STATEMENT

All authors of this paper have read and agree to comply with the ICLR Code of Ethics (`https://iclr.cc/public/CodeOfEthics`). This work does not involve human subjects, sensitive data, or applications that could directly cause harm. The datasets employed in our experiments are publicly available and do not contain personally identifiable information. We have taken measures to ensure that our methodology does not introduce or exacerbate unfair bias, and potential limitations as well as societal impacts are discussed in the main text. There are no conflicts of interest or external sponsorships that could influence the results or their interpretation.

## REPRODUCIBILITY STATEMENT

We are committed to ensuring the reproducibility of our results. All experimental details, including model architectures, hyperparameters, training procedures, and evaluation metrics, are described in Section 6 and Appendix E. The datasets used are publicly accessible, and we include the version information in Section 6.

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

## LLM USAGE STATEMENT

During the preparation of this manuscript, we employed large language models (LLMs) as general-purpose writing and editing tools. LLMs were utilized to enhance the clarity, grammar, and structure of the text, but they did not contribute to research ideation, experimental design, or scientific content. No LLM was used to generate new ideas, mathematical proofs, or experimental results. All content, including any text suggested by LLMs, was thoroughly reviewed and verified by the authors, who assume full responsibility for the final manuscript.

## A   EXPECTATION EVALUATIONS ON SPECIFIC DISTRIBUTION ASSUMPTIONS

From Section 3, we have established that when the random variable $x$ in model (3) belongs to the normal mean–variance mixture family, the surrogate function (10) can be derived, along with its closed-form solutions with respect to $\mu$, $\xi$, and $\Sigma$. Nevertheless, these closed-form solutions (12) depend on the expected values $U_t$, $v_t$, $w_t$, $\iota_t$, and $\zeta_t$, which are determined by the distribution of the random variable $x$. Therefore, in the following, we analyze the integrals corresponding to these expected values under various distributions of $x$, and examine the convergence properties and statistical characteristics of the proposed algorithm under specific distributional assumptions. We first consider the fundamental case of the Gaussian distribution (i.e., $z = 1$ and $\xi$). Let $x \sim \mathcal{N}(\mu, \Sigma)$. In this case, the first- and second-order moments are given by Ho et al. (2012) as

$$\mathsf{E}_{x|y;\theta}[x] = \mu + p^{-1}(y;\theta)\,\Sigma q, \tag{19}$$

$$\mathsf{E}_{x|y;\theta}[xx^\top] = \mu\mu^\top + \Sigma + p^{-1}(y;\theta)\left(2\left(\Sigma q\right)\mu^\top + \Sigma\left(H + D\right)\Sigma\right). \tag{20}$$

Let the interval $\mathcal{Q}^{-1}(y_i)$ be $[l_i, u_i]$. The $i$-th elements of vector $q$ is given by

$$q_i = p(x_i = l_i, y_{\backslash i};\theta) - p(x_i = u_i, y_{\backslash i};\theta), \tag{21}$$

The matrix $H$ is a matrix with all diagonal entries being zero and the $(i, j)$-th off-diagonal element being

$$H_{ij} = h(l_i, l_j) - h(l_i, u_j) - h(u_i, l_j) + h(u_i, u_j), \tag{22}$$

with

$$h(c_i, c_j) = p(x_i = c_i, x_j = c_j, y_{\backslash i,j};\theta)).$$

The matrix $D$ is a diagonal matrix with the diagonal entries:

$$D_{ii} = \frac{l_i - \mu_i}{\sigma_i^2} p(x_i = l_i, y_{\backslash i};\theta) - \frac{u_i - \mu_i}{\sigma_i^2} p(x_i = u_i, y_{\backslash i};\theta) - \frac{[\Sigma H]_{ii}}{\sigma_i^2}. \tag{23}$$

Now we consider the general normal mean-variance mixture with $x = \mu + z\xi + (z\Sigma)^{\frac{1}{2}}\epsilon$. In this case, the first and second moments satisfy

$$\mathsf{E}_{x,z|y;\theta}[x] = \mathsf{E}_{z|y;\theta}\left[\mathsf{E}_{x|z,y;\theta}[x]\right] \text{ and } \mathsf{E}_{x,z|y;\theta}[xx^\top] = \mathsf{E}_{z|y;\theta}\left[\mathsf{E}_{x|z,y;\theta}[xx^\top]\right].$$

Based on $x \mid z \sim \mathcal{N}(\mu + z\xi, z\Sigma)$ and the moments in (19) and (20), we have

$$\mathsf{E}_{x|z,y;\theta}[x] = \mu + z\xi + p^{-1}(y \mid z;\theta)z\Sigma q_z, \tag{24}$$

$$\begin{aligned}\mathsf{E}_{x|z,y;\theta}[xx^\top] = {} & \mu\mu^\top + z^2\xi\xi^\top + 2z\mu\xi^\top + z\Sigma \\ & + p^{-1}(y \mid z;\theta)\left(2z\left(\Sigma q_z\right)\mu^\top + 2z^2\left(\Sigma q_z\right)\xi^\top + z^2\Sigma\left(H_z + D_z\right)\Sigma\right),\end{aligned} \tag{25}$$

where the $i$-th elements of vector $q_z$ is denoted as

$$q_{z,i} = p(x_i = l_i, y_{\backslash i} \mid z;\theta) - p(x_i = u_i, y_{\backslash i} \mid z;\theta), \tag{26}$$

the matrix $H_z$ is a matrix with all diagonal entries being zero and the $(i, j)$-th off-diagonal element being

$$H_{z,ij} = h_z(l_i, l_j) - h_z(l_i, u_j) - h_z(u_i, l_j) + h_z(u_i, u_j), \tag{27}$$

with

$$h_z(c_i, c_j) = p(x_i = c_i, x_j = c_j, y_{\backslash i,j} \mid z;\theta)),$$

and the matrix $\boldsymbol{D}_z$ is a diagonal matrix with the diagonal entries:

$$\boldsymbol{D}_{z,ii} = \frac{l_i - \mu_i - z\xi_i}{\sigma_i^2} p(x_i = l_i, \boldsymbol{y}_{\setminus i} \mid z; \boldsymbol{\theta}) - \frac{u_i - \mu_i - z\xi_i}{\sigma_i^2} p(x_i = u_i, \boldsymbol{y}_{\setminus i} \mid z; \boldsymbol{\theta}) - \frac{[\boldsymbol{\Sigma}\boldsymbol{H}_z]_{ii}}{\sigma_i^2}. \tag{28}$$

Based on (24) and (25), we can obtain

$$\mathsf{E}_{\boldsymbol{x},z|\boldsymbol{y};\boldsymbol{\theta}}[\boldsymbol{x}] = \boldsymbol{\mu} + \mathsf{E}_{z|\boldsymbol{y};\boldsymbol{\theta}}[z]\boldsymbol{\xi} + \mathsf{E}_{z|\boldsymbol{y};\boldsymbol{\theta}}\left[zp^{-1}(\boldsymbol{y} \mid z; \boldsymbol{\theta})\boldsymbol{\Sigma}\boldsymbol{q}_z\right], \tag{29}$$

$$\begin{aligned}
\mathsf{E}_{\boldsymbol{x},z|\boldsymbol{y};\boldsymbol{\theta}}\left[\boldsymbol{x}\boldsymbol{x}^\top\right] &= \boldsymbol{\mu}\boldsymbol{\mu}^\top + \mathsf{E}_{z|\boldsymbol{y};\boldsymbol{\theta}}\left[z^2\right]\boldsymbol{\xi}\boldsymbol{\xi}^\top + 2\,\mathsf{E}_{z|\boldsymbol{y};\boldsymbol{\theta}}[z]\boldsymbol{\mu}\boldsymbol{\xi}^\top + \mathsf{E}_{z|\boldsymbol{y};\boldsymbol{\theta}}[z]\boldsymbol{\Sigma} \\
&\quad + \mathsf{E}_{z|\boldsymbol{y};\boldsymbol{\theta}}\left[p^{-1}(\boldsymbol{y} \mid z; \boldsymbol{\theta})\left(2z\left(\boldsymbol{\Sigma}\boldsymbol{q}_z\right)\boldsymbol{\mu}^\top + 2z^2\left(\boldsymbol{\Sigma}\boldsymbol{q}_z\right)\boldsymbol{\xi}^\top + z^2\boldsymbol{\Sigma}\left(\boldsymbol{H}_z + \boldsymbol{D}_z\right)\boldsymbol{\Sigma}\right)\right].
\end{aligned} \tag{30}$$

We first compute $\mathsf{E}_{z|\boldsymbol{y};\boldsymbol{\theta}}\left[z^k\right]$ as follows:

$$\mathsf{E}_{z|\boldsymbol{y};\boldsymbol{\theta}}\left[z^k\right] = \int_0^{+\infty} p(z \mid \boldsymbol{y}; \boldsymbol{\theta})z^k \mathrm{d}z = \int_0^{+\infty} \frac{p(\boldsymbol{y} \mid z; \boldsymbol{\theta})p(z)}{p(\boldsymbol{y}; \boldsymbol{\theta})} z^k \mathrm{d}z.$$

Here, we introduce the size-biased distribution of order $k$ of the positive random variable $z$, which has the density function $p_k(z) = \frac{z^k p(z)}{\mathsf{E}_z[z^k]}$. Hence, we have

$$\mathsf{E}_{z|\boldsymbol{y};\boldsymbol{\theta}}\left[z^k\right] = \mathsf{E}_z[z^k]\frac{\int_0^{+\infty} p(\boldsymbol{y} \mid z; \boldsymbol{\theta})p_k(z)\mathrm{d}z}{p(\boldsymbol{y}; \boldsymbol{\theta})} = \mathsf{E}_z[z^k]\frac{p_k(\boldsymbol{y}; \boldsymbol{\theta})}{p(\boldsymbol{y}; \boldsymbol{\theta})}, \tag{31}$$

where we denote $p_k(\boldsymbol{y}; \boldsymbol{\theta}) = \int_0^{+\infty} p(\boldsymbol{y} \mid z; \boldsymbol{\theta})p_k(z)\mathrm{d}z$.

Similarly, we can obtain

$$\mathsf{E}_{z|\boldsymbol{y};\boldsymbol{\theta}}\left[z^k p^{-1}(\boldsymbol{y} \mid z; \boldsymbol{\theta})\boldsymbol{q}_z\right] = \mathsf{E}_z[z^k]p^{-1}(\boldsymbol{y}; \boldsymbol{\theta})\boldsymbol{q}_k, \tag{32}$$

where $\boldsymbol{q}_k = \int_0^{+\infty} \boldsymbol{q}_z p_k(z)\mathrm{d}z$ with the $i$-th element

$$q_{k,i} = p_k(x_i = l_i, \boldsymbol{y}_{\setminus i}; \boldsymbol{\theta}) - p_k(x_i = u_i, \boldsymbol{y}_{\setminus i}; \boldsymbol{\theta}). \tag{33}$$

For the expectation of $z^k \boldsymbol{H}_z$ we have

$$\mathsf{E}_{z|\boldsymbol{y};\boldsymbol{\theta}}\left[z^k p^{-1}(\boldsymbol{y} \mid z; \boldsymbol{\theta})\left(\boldsymbol{H}_z + \boldsymbol{D}_z\right)\right] = p^{-1}(\boldsymbol{y}; \boldsymbol{\theta})\left(\boldsymbol{H}_k + \boldsymbol{D}_k\right), \tag{34}$$

with the $(i, j)$ entry of

$$\boldsymbol{H}_{k,ij} = \mathsf{E}_z[z^k]\left(h_k(l_i, l_j) - h_k(l_i, u_j) - h_k(u_i, l_j) + h_k(u_i, u_j)\right), \tag{35}$$

$$h_k(c_i, c_j) = p_k(x_i = c_i, x_j = c_j, \boldsymbol{y}_{\setminus i,j}; \boldsymbol{\theta})), \tag{36}$$

and the matrix $\boldsymbol{D}_k$ is a diagonal matrix with the diagonal entries:

$$\boldsymbol{D}_{k,ii} = \mathsf{E}_z[z^{k-1}]\frac{l_i - \mu_i}{\sigma_i^2}p_{k-1}(x_i = l_i, \boldsymbol{y}_{\setminus i} \mid z; \boldsymbol{\theta}) - \mathsf{E}_z[z^{k-1}]\frac{u_i - \mu_i}{\sigma_i^2}p_{k-1}(x_i = u_i, \boldsymbol{y}_{\setminus i} \mid z; \boldsymbol{\theta})$$

$$- \mathsf{E}_z[z^k]\frac{\xi_i}{\sigma_i^2}q_{k,i} - \frac{[\boldsymbol{\Sigma}\boldsymbol{H}_k]_{ii}}{\sigma_i^2}. \tag{37}$$

Therefore, the expectations in (29) and (30) become

$$\mathsf{E}_{\boldsymbol{x},z|\boldsymbol{y};\boldsymbol{\theta}}[\boldsymbol{x}] = \boldsymbol{\mu} + \mathsf{E}_{z|\boldsymbol{y};\boldsymbol{\theta}}[z]\boldsymbol{\xi} + \mathsf{E}_z[z]p^{-1}(\boldsymbol{y}; \boldsymbol{\theta})\boldsymbol{\Sigma}\boldsymbol{q}_1, \tag{38}$$

$$\begin{aligned}
\mathsf{E}_{\boldsymbol{x},z|\boldsymbol{y};\boldsymbol{\theta}}\left[\boldsymbol{x}\boldsymbol{x}^\top\right] &= \boldsymbol{\mu}\boldsymbol{\mu}^\top + \mathsf{E}_{z|\boldsymbol{y};\boldsymbol{\theta}}\left[z^2\right]\boldsymbol{\xi}\boldsymbol{\xi}^\top + 2\mathsf{E}_{z|\boldsymbol{y};\boldsymbol{\theta}}[z]\boldsymbol{\mu}\boldsymbol{\xi}^\top + \mathsf{E}_{z|\boldsymbol{y};\boldsymbol{\theta}}[z]\boldsymbol{\Sigma} \\
&\quad + p^{-1}(\boldsymbol{y}; \boldsymbol{\theta})\left(2\mathsf{E}_z[z]\left(\boldsymbol{\Sigma}\boldsymbol{q}_1\right)\boldsymbol{\mu}^\top + 2\mathsf{E}_z\left[z^2\right]\left(\boldsymbol{\Sigma}\boldsymbol{q}_2\right)\boldsymbol{\xi}^\top + \boldsymbol{\Sigma}\left(\boldsymbol{H}_2 + \boldsymbol{D}_2\right)\boldsymbol{\Sigma}\right).
\end{aligned} \tag{39}$$

Meanwhile, we can also obtain

$$\mathsf{E}_{\boldsymbol{x},z|\boldsymbol{y};\boldsymbol{\theta}}\left[z^{-1}\boldsymbol{x}\right] = \mathsf{E}_{z|\boldsymbol{y};\boldsymbol{\theta}}\left[\mathsf{E}_{\boldsymbol{x}|z,\boldsymbol{y};\boldsymbol{\theta}}[z^{-1}\boldsymbol{x}]\right] = \mathsf{E}_{z|\boldsymbol{y};\boldsymbol{\theta}}\left[z^{-1}\right]\boldsymbol{\mu} + \boldsymbol{\xi} + p^{-1}(\boldsymbol{y};\boldsymbol{\theta})\boldsymbol{\Sigma q}, \tag{40}$$

$$\mathsf{E}_{\boldsymbol{x},z|\boldsymbol{y};\boldsymbol{\theta}}\left[z^{-1}\boldsymbol{x}\boldsymbol{x}^\top\right] = \mathsf{E}_{z|\boldsymbol{y};\boldsymbol{\theta}}\left[\mathsf{E}_{\boldsymbol{x}|z,\boldsymbol{y};\boldsymbol{\theta}}[z^{-1}\boldsymbol{x}\boldsymbol{x}^\top]\right]$$

$$= \mathsf{E}_{z|\boldsymbol{y};\boldsymbol{\theta}}\left[z^{-1}\right]\boldsymbol{\mu}\boldsymbol{\mu}^\top + \mathsf{E}_{z|\boldsymbol{y};\boldsymbol{\theta}}\left[z\right]\boldsymbol{\xi}\boldsymbol{\xi}^\top + 2\boldsymbol{\mu}\boldsymbol{\xi}^\top + \boldsymbol{\Sigma} \tag{41}$$

$$+ p^{-1}(\boldsymbol{y};\boldsymbol{\theta})\left(2\left(\boldsymbol{\Sigma q}\right)\boldsymbol{\mu}^\top + 2\mathsf{E}_z\left[z\right]\left(\boldsymbol{\Sigma q}_1\right)\boldsymbol{\xi}^\top + \boldsymbol{\Sigma}\left(\boldsymbol{H}_1 + \boldsymbol{D}_1\right)\boldsymbol{\Sigma}\right).$$

# B THE PROOF OF PROPOSITION 1

In this section, we prove the global linear convergence of our proposed ECM algorithm with respect to the three parameters: $\boldsymbol{\mu}$, $\boldsymbol{\Sigma}$, and $\boldsymbol{\xi}$. Since the ECM algorithm can always converge to the stationary points (McLachlan & Krishnan, 2008), we define $\boldsymbol{\mu}^\star$, $\boldsymbol{\Sigma}^\star$, and $\boldsymbol{\xi}^\star$ as the stationary points of the optimization problem (6). In the following, we give the convergence analysis of these three parameters with the convergence rate, respectively.

## B.1 CONVERGENCE RATE OF LOCATION PARAMETER

Given the update rule of $\boldsymbol{\mu}$ [2] in (12), we have

$$\boldsymbol{\mu} = \frac{\sum_{t=1}^n(\boldsymbol{v}_t - \underline{\boldsymbol{\xi}})}{\sum_{t=1}^n \iota_t} = \frac{\sum_{t=1}^n(\mathsf{E}_{\boldsymbol{x}_t,z_t|\boldsymbol{y}_t;\underline{\boldsymbol{\theta}}}[z^{-1}\boldsymbol{x}] - \underline{\boldsymbol{\xi}})}{\sum_{t=1}^n \mathsf{E}_{z_t|\boldsymbol{y}_t;\underline{\boldsymbol{\theta}}}[z^{-1}]}. \tag{42}$$

Based on (31) and (40), the update rule of $\boldsymbol{\mu}$ in (42) becomes

$$\boldsymbol{\mu} = \underline{\boldsymbol{\mu}} + \left(\sum_{i=1}^n \frac{p_{-1}(\boldsymbol{y}_i;\underline{\boldsymbol{\theta}})}{p(\boldsymbol{y}_i;\underline{\boldsymbol{\theta}})}\right)^{-1}\sum_{t=1}^n p^{-1}(\boldsymbol{y}_t;\underline{\boldsymbol{\theta}})\underline{\boldsymbol{\Sigma}}\boldsymbol{q}_t \tag{43}$$

Hence, the distance between $\boldsymbol{\mu}$ at the current iteration and $\boldsymbol{\mu}^\star$ is given by

$$\|\boldsymbol{\mu} - \boldsymbol{\mu}^\star\|_2 = \left\|\underline{\boldsymbol{\mu}} - \boldsymbol{\mu}^\star + \left(\sum_{i=1}^n \frac{p_{-1}(\boldsymbol{y}_i;\underline{\boldsymbol{\theta}})}{p(\boldsymbol{y}_i;\underline{\boldsymbol{\theta}})}\right)^{-1}\sum_{t=1}^n p^{-1}(\boldsymbol{y}_t;\underline{\boldsymbol{\theta}})\underline{\boldsymbol{\Sigma}}\boldsymbol{q}_t\right\|_2. \tag{44}$$

To further analyze the term on the right side in (44), we first give a result of $\boldsymbol{q}$.

**Lemma 3.** *When $\boldsymbol{x}$ follows a normal mean-variance mixture and $\boldsymbol{q}$ is defined in (21), we have*

$$\boldsymbol{q} = \nabla_{\boldsymbol{\mu}}p(\boldsymbol{y};\boldsymbol{\theta}).$$

*Proof.* Consider the partial derivative of $p(\boldsymbol{y};\boldsymbol{\theta})$ with respect to the $i$-th element of $\boldsymbol{\mu}$. We have that

$$\nabla_{\mu_i}p(\boldsymbol{y};\boldsymbol{\theta}) = \nabla_{\mu_i}\int_{\mathcal{Q}^{-1}(\boldsymbol{y})} p(\boldsymbol{x};\boldsymbol{\theta})\mathrm{d}\boldsymbol{x} = \int_{\mathcal{Q}^{-1}(\boldsymbol{y})}\nabla_{\mu_i}p(\boldsymbol{x};\boldsymbol{\theta})\mathrm{d}\boldsymbol{x},$$

where the interchange of the integration and differentiation operations in the second equal sign is valid as the Leibniz integral rule (Casella & Berger, 2024). Since the partial derivatives of $p(\boldsymbol{x};\boldsymbol{\theta})$

$$\nabla_{\mu_i}p(\boldsymbol{x}\mid z;\boldsymbol{\theta}) = -\nabla_{x_i}p(\boldsymbol{x}\mid z;\boldsymbol{\theta}),$$

we can obtain that

$$\nabla_{\mu_i}p(\boldsymbol{y};\boldsymbol{\theta}) = -\int_0^{+\infty}\int_{\mathcal{Q}^{-1}(\boldsymbol{y}_{\backslash i})}\int_{l_i}^{u_i}\nabla_{x_i}p(\boldsymbol{x}\mid z;\boldsymbol{\theta})p(z)\mathrm{d}x_i\mathrm{d}\boldsymbol{x}_{\backslash i}\mathrm{d}z$$

$$= \int_0^{+\infty}\int_{\mathcal{Q}^{-1}(\boldsymbol{y}_{\backslash i})}p(z)\left(p(\boldsymbol{x}_{\backslash i}, x_i = l_i\mid z;\boldsymbol{\theta}) - p(\boldsymbol{x}_{\backslash i}, x_i = u_i\mid z;\boldsymbol{\theta})\right)\mathrm{d}\boldsymbol{x}_{\backslash i}\mathrm{d}z$$

$$= p(x_i = l_i, \boldsymbol{y}_{\backslash i}, \boldsymbol{\theta}) - p(x_i = u_i, \boldsymbol{y}_{\backslash i}, \boldsymbol{\theta}),$$

which is equivalent to the definition of the $i$-th element of $\boldsymbol{q}$ in (21).

---

[2]For simplicity, we use $\underline{\boldsymbol{\mu}}$, $\underline{\boldsymbol{\xi}}$, and $\underline{\boldsymbol{\Sigma}}$ to denote the parameters $\boldsymbol{\mu}^{(k)}$, $\boldsymbol{\xi}^{(k)}$, and $\boldsymbol{\Sigma}^{(k)}$ before the $k$-th update, and use $\boldsymbol{\mu}$, $\boldsymbol{\xi}$, and $\boldsymbol{\Sigma}$ to denote the updated parameters $\boldsymbol{\mu}^{(k+1)}$, $\boldsymbol{\xi}^{(k+1)}$, and $\boldsymbol{\Sigma}^{(k+1)}$.

Based on Lemma 3, we have $p^{-1}(\boldsymbol{y};\boldsymbol{\theta})\boldsymbol{q} = \nabla_{\boldsymbol{\mu}}\log p(\boldsymbol{y};\boldsymbol{\theta})$ and $\boldsymbol{q}^{\star} = \mathbf{0}$. Then the distance (44) becomes

$$
\begin{aligned}
\|\boldsymbol{\mu} - \boldsymbol{\mu}^{\star}\|_2 &= \left\| \underline{\boldsymbol{\mu}} - \boldsymbol{\mu}^{\star} + \left( \sum_{i=1}^{n} \frac{p_{-1}(\boldsymbol{y}_i;\underline{\boldsymbol{\theta}})}{p(\boldsymbol{y}_i;\underline{\boldsymbol{\theta}})} \right)^{-1} \sum_{t=1}^{n} \underline{\boldsymbol{\Sigma}}\nabla_{\underline{\boldsymbol{\mu}}}\log p(\boldsymbol{y}_t;\underline{\boldsymbol{\theta}}) \right. \\
&\quad \left. - \left( \sum_{i=1}^{n} \frac{p_{-1}(\boldsymbol{y}_i;\boldsymbol{\theta}^{\star})}{p(\boldsymbol{y}_i;\boldsymbol{\theta}^{\star})} \right)^{-1} \sum_{t=1}^{n} \underline{\boldsymbol{\Sigma}}\nabla_{\boldsymbol{\mu}^{\star}}\log p(\boldsymbol{y}_t;\boldsymbol{\theta}^{\star}) \right\|_2 \\
&= \left\| \underline{\boldsymbol{\mu}} - \boldsymbol{\mu}^{\star} + \int_0^1 \left( \sum_{i=1}^{n} \frac{p_{-1}(\boldsymbol{y}_i;\tilde{\boldsymbol{\theta}}_\mu)}{p(\boldsymbol{y}_i;\tilde{\boldsymbol{\theta}}_\mu)} \right)^{-1} \sum_{t=1}^{n} \underline{\boldsymbol{\Sigma}}\nabla_{\tilde{\boldsymbol{\mu}}}^2 \log p(\boldsymbol{y}_t;\tilde{\boldsymbol{\theta}}_\mu)(\tilde{\boldsymbol{\mu}} - \boldsymbol{\mu}^{\star})\mathrm{d}\beta \right\|_2 \\
&\leq \sup_{\beta\in(0,1]} \left\| \boldsymbol{I} + \left( \sum_{i=1}^{n} \frac{p_{-1}(\boldsymbol{y}_i;\tilde{\boldsymbol{\theta}}_\mu)}{p(\boldsymbol{y}_i;\tilde{\boldsymbol{\theta}}_\mu)} \right)^{-1} \sum_{t=1}^{n} \underline{\boldsymbol{\Sigma}}\nabla_{\tilde{\boldsymbol{\mu}}}^2 \log p(\boldsymbol{y}_t;\tilde{\boldsymbol{\theta}}_\mu) \right\|_2 \|\underline{\boldsymbol{\mu}} - \boldsymbol{\mu}^{\star}\|_2,
\end{aligned}
$$

where the second equation is given by the mean value theorem of integrals and $\tilde{\boldsymbol{\theta}}_\mu = \{\tilde{\boldsymbol{\mu}}, \underline{\boldsymbol{\Sigma}}, \underline{\boldsymbol{\xi}}\}$ with $\tilde{\boldsymbol{\mu}} = \beta\underline{\boldsymbol{\mu}} + (1-\beta)\boldsymbol{\mu}^{\star}$ and $\beta \in (0,1]$.

To bound the term $\sup_{\beta\in(0,1]} \left\| \boldsymbol{I} + \left( \sum_{i=1}^{n} \frac{p_{-1}(\boldsymbol{y}_i;\tilde{\boldsymbol{\theta}}_\mu)}{p(\boldsymbol{y}_i;\tilde{\boldsymbol{\theta}}_\mu)} \right)^{-1} \sum_{t=1}^{n} \underline{\boldsymbol{\Sigma}}\nabla_{\tilde{\boldsymbol{\mu}}}^2 \log p(\boldsymbol{y}_t;\tilde{\boldsymbol{\theta}}_\mu) \right\|_2$, we need some results for the second order derivative of $\log p(\boldsymbol{y}_t;\boldsymbol{\theta})$ at first.

**Lemma 4.** *The second order derivative of $\log p(\boldsymbol{y};\boldsymbol{\theta})$ with respect to $\boldsymbol{\mu}$, i.e., $\nabla_{\boldsymbol{\mu}}^2 \log p(\boldsymbol{y};\boldsymbol{\theta})$, satisfy*

*1. $\nabla_{\boldsymbol{\mu}}^2 \log p(\boldsymbol{y};\boldsymbol{\theta})$ is a negative definite matrix;*

*2. $\boldsymbol{I} + \left( \sum_{i=1}^{n} \frac{p_{-1}(\boldsymbol{y}_i;\boldsymbol{\theta})}{p(\boldsymbol{y}_i;\boldsymbol{\theta})} \right)^{-1} \sum_{t=1}^{n} \boldsymbol{\Sigma}\nabla_{\boldsymbol{\mu}}^2 \log p(\boldsymbol{y}_t;\boldsymbol{\theta})$ is a positive definite matrix,*

*for all $\boldsymbol{\theta}$ in the feasible set.*

*Proof.* The second order derivative of $\log p(\boldsymbol{y};\boldsymbol{\theta})$ with respect to $\boldsymbol{\mu}$ is given as

$$
\nabla_{\boldsymbol{\mu}}^2 \log p(\boldsymbol{y};\boldsymbol{\theta}) = \nabla_{\boldsymbol{\mu}}\left( p^{-1}(\boldsymbol{y};\boldsymbol{\theta})\nabla_{\boldsymbol{\mu}}p(\boldsymbol{y};\boldsymbol{\theta}) \right) = \frac{\nabla_{\boldsymbol{\mu}}^2 p(\boldsymbol{y};\boldsymbol{\theta})\cdot p(\boldsymbol{y};\boldsymbol{\theta}) - \nabla_{\boldsymbol{\mu}}p(\boldsymbol{y};\boldsymbol{\theta})\nabla_{\boldsymbol{\mu}}^{\top}p(\boldsymbol{y};\boldsymbol{\theta})}{p^2(\boldsymbol{y};\boldsymbol{\theta})}.
\tag{45}
$$

To analyze the above expression, we should compute the first and second order derivatives of $p(\boldsymbol{y};\boldsymbol{\theta})$ with respect to $\boldsymbol{\mu}$ at first. The first order one is given as follows:

$$
\nabla_{\boldsymbol{\mu}}p(\boldsymbol{y};\boldsymbol{\theta}) = \nabla_{\boldsymbol{\mu}}\int_{\mathcal{Q}^{-1}(\boldsymbol{y})} p(\boldsymbol{x};\boldsymbol{\theta})\mathrm{d}\boldsymbol{x} = \int_{\mathcal{Q}^{-1}(\boldsymbol{y})} \nabla_{\boldsymbol{\mu}}p(\boldsymbol{x};\boldsymbol{\theta})\mathrm{d}\boldsymbol{x}.
\tag{46}
$$

Since $p(\boldsymbol{x};\boldsymbol{\theta}) = \int_0^{+\infty} p(z)p(\boldsymbol{x}\mid z;\boldsymbol{\theta})\mathrm{d}z$, (46) becomes

$$
\begin{aligned}
\nabla_{\boldsymbol{\mu}}p(\boldsymbol{y};\boldsymbol{\theta}) &= \int_{\mathcal{Q}^{-1}(\boldsymbol{y})}\int_0^{+\infty} p(z)\nabla_{\boldsymbol{\mu}}p(\boldsymbol{x}\mid z;\boldsymbol{\theta})\mathrm{d}z\mathrm{d}\boldsymbol{x} \\
&= \int_{\mathcal{Q}^{-1}(\boldsymbol{y})}\int_0^{+\infty} \frac{p(z)}{z}p(\boldsymbol{x}\mid z;\boldsymbol{\theta})\boldsymbol{\Sigma}^{-1}(\boldsymbol{x}-\boldsymbol{\mu}-z\boldsymbol{\xi})\mathrm{d}z\mathrm{d}\boldsymbol{x}.
\end{aligned}
$$

Then the second order one can be further computed as

$$
\begin{aligned}
\nabla_{\boldsymbol{\mu}}^2 p(\boldsymbol{y};\boldsymbol{\theta}) &= \int_{\mathcal{Q}^{-1}(\boldsymbol{y})}\int_0^{+\infty} \frac{p(z)}{z}\nabla_{\boldsymbol{\mu}}\left( p(\boldsymbol{x}\mid z;\boldsymbol{\theta})\boldsymbol{\Sigma}^{-1}(\boldsymbol{x}-\boldsymbol{\mu}-z\boldsymbol{\xi}) \right)\mathrm{d}z\mathrm{d}\boldsymbol{x} \\
&= \int_{\mathcal{Q}^{-1}(\boldsymbol{y})}\int_0^{+\infty} \boldsymbol{\Sigma}^{-1}p(\boldsymbol{x}\mid z;\boldsymbol{\theta})p(z)\left( -z^{-1}\boldsymbol{I} + z^{-2}(\boldsymbol{x}-\boldsymbol{\mu}-z\boldsymbol{\xi})(\boldsymbol{x}-\boldsymbol{\mu}-z\boldsymbol{\xi})^{\top}\boldsymbol{\Sigma}^{-1} \right)\mathrm{d}z\mathrm{d}\boldsymbol{x}.
\end{aligned}
\tag{47}
$$

**Lemma 5.** *For a function $f(\boldsymbol{x}, z)$, if $\int_{\mathcal{Q}^{-1}(\boldsymbol{y})} p(\boldsymbol{x} \mid z; \boldsymbol{\theta}) f(\boldsymbol{x}, z) \mathrm{d}\boldsymbol{x}$ is integrable, we have*

$$\int_{\mathbb{R}^d} \int_0^{+\infty} p(z) p(\boldsymbol{x}, \boldsymbol{y} \mid z; \boldsymbol{\theta}) f(\boldsymbol{x}, z) \mathrm{d}z \mathrm{d}\boldsymbol{x} = p(\boldsymbol{y}; \boldsymbol{\theta}) \mathsf{E}_{\boldsymbol{x}, z \mid \boldsymbol{y}; \boldsymbol{\theta}} \left[ f(\boldsymbol{x}, z) \right].$$

*Proof.* Given that

$$\int_{\mathcal{Q}^{-1}(\boldsymbol{y})} p(\boldsymbol{x} \mid z; \boldsymbol{\theta}) f(\boldsymbol{x}, z) \mathrm{d}\boldsymbol{x} = \int_{\mathbb{R}^d} p(\boldsymbol{x}, \boldsymbol{y} \mid z; \boldsymbol{\theta}) f(\boldsymbol{x}, z) \mathrm{d}\boldsymbol{x},$$

we can obtain that

$$\int_{\mathbb{R}^d} \int_0^{+\infty} p(z) p(\boldsymbol{x}, \boldsymbol{y} \mid z; \boldsymbol{\theta}) f(\boldsymbol{x}, z) \mathrm{d}z \mathrm{d}\boldsymbol{x} = p(\boldsymbol{y}; \boldsymbol{\theta}) \int_{\mathbb{R}^d} \int_0^{+\infty} p(\boldsymbol{x}, z \mid \boldsymbol{y}; \boldsymbol{\theta}) f(\boldsymbol{x}, z) \mathrm{d}z \mathrm{d}\boldsymbol{x}$$
$$= p(\boldsymbol{y}; \boldsymbol{\theta}) \mathsf{E}_{\boldsymbol{x}, z \mid \boldsymbol{y}; \boldsymbol{\theta}} \left[ f(\boldsymbol{x}, z) \right].$$

$\square$

Based on Lemma 5, the first and second order derivatives of $p(\boldsymbol{y}; \boldsymbol{\theta})$ becomes

$$\nabla_{\boldsymbol{\mu}} p(\boldsymbol{y}; \boldsymbol{\theta}) = p(\boldsymbol{y}; \boldsymbol{\theta}) \boldsymbol{\Sigma}^{-1} \mathsf{E}_{\boldsymbol{x}, z \mid \boldsymbol{y}; \boldsymbol{\theta}} \left[ z^{-1} (\boldsymbol{x} - \boldsymbol{\mu} - z\boldsymbol{\xi}) \right],$$

and

$$\nabla_{\boldsymbol{\mu}}^2 p(\boldsymbol{y}; \boldsymbol{\theta}) = p(\boldsymbol{y}; \boldsymbol{\theta}) \boldsymbol{\Sigma}^{-1} \left( -\mathsf{E}_{z \mid \boldsymbol{y}; \boldsymbol{\theta}} \left[ z^{-1} \right] \boldsymbol{I} + \mathsf{E}_{\boldsymbol{x}, z \mid \boldsymbol{y}; \boldsymbol{\theta}} \left[ z^{-2} (\boldsymbol{x} - \boldsymbol{\mu} - z\boldsymbol{\xi}) (\boldsymbol{x} - \boldsymbol{\mu} - z\boldsymbol{\xi})^\top \right] \boldsymbol{\Sigma}^{-1} \right).$$

Then the second order derivative of $\log p(\boldsymbol{y}; \boldsymbol{\theta})$ in (45) becomes

$$\nabla_{\boldsymbol{\mu}}^2 \log p(\boldsymbol{y}; \boldsymbol{\theta}) = -\boldsymbol{\Sigma}^{-1} + \boldsymbol{\Sigma}^{-1} \mathsf{E}_{\boldsymbol{x} \mid \boldsymbol{y}; \boldsymbol{\theta}} [(\boldsymbol{x} - \boldsymbol{\mu})(\boldsymbol{x} - \boldsymbol{\mu})^\top] \boldsymbol{\Sigma}^{-1} - \boldsymbol{\Sigma}^{-1} \mathsf{E}_{\boldsymbol{x} \mid \boldsymbol{y}; \boldsymbol{\theta}} [\boldsymbol{x} - \boldsymbol{\mu}] \mathsf{E}_{\boldsymbol{x} \mid \boldsymbol{y}; \boldsymbol{\theta}} [\boldsymbol{x} - \boldsymbol{\mu}]^\top \boldsymbol{\Sigma}^{-1}$$
$$= \boldsymbol{\Sigma}^{-1} \left( -\mathsf{E}_{z \mid \boldsymbol{y}; \boldsymbol{\theta}} \left[ z^{-1} \right] \boldsymbol{\Sigma} + \mathsf{Cov}_{\boldsymbol{x}, z \mid \boldsymbol{y}; \boldsymbol{\theta}} \left[ z^{-1} \boldsymbol{x} \right] \right) \boldsymbol{\Sigma}^{-1}. \tag{48}$$

To prove the negative definite of $\nabla_{\boldsymbol{\mu}}^2 \log p(\boldsymbol{y}; \boldsymbol{\theta})$, we introduce the Brascamp–Lieb inequality.

**Lemma 6** (Brascamp & Lieb (1976))**.** *Consider a probability density function $p(\boldsymbol{x})$ which is log-concave to $\boldsymbol{x}$. The Brascamp–Lieb inequality is given by*

$$\mathsf{Cov}_{\tilde{\boldsymbol{x}}}(f(\boldsymbol{x})) \preceq \mathsf{E}_{\tilde{\boldsymbol{x}}} [\nabla_{\boldsymbol{x}}^\top f(\boldsymbol{x}) [\nabla_{\boldsymbol{x}}^2 (-\log p(\tilde{\boldsymbol{x}}))]^{-1} \nabla_{\boldsymbol{x}} f(\boldsymbol{x})],$$

*where the equality is obtained when $\log p(\tilde{\boldsymbol{x}})$ is linear with respect to $\boldsymbol{x}$.*

Let $\tilde{\boldsymbol{x}} = \boldsymbol{x}, z \mid \boldsymbol{y}$ and $f(\boldsymbol{x}) = z^{-1} \boldsymbol{x}$. Since $\nabla_{\boldsymbol{x}}^2 \log p(\boldsymbol{x}, z \mid \boldsymbol{y}; \boldsymbol{\theta}) = -\frac{\boldsymbol{\Sigma}^{-1}}{z}$, $p(\boldsymbol{x}, z \mid \boldsymbol{y}; \boldsymbol{\theta})$ is a log-concave function with respect to $\boldsymbol{x}$. Based on the Brascamp–Lieb inequality in Lemma 6, since $\log p(\boldsymbol{x}, z \mid \boldsymbol{y}; \boldsymbol{\theta})$ is non-linear to $\boldsymbol{x}$, we have

$$\mathsf{Cov}_{\boldsymbol{x}, z \mid \boldsymbol{y}; \boldsymbol{\theta}} \left[ z^{-1} \boldsymbol{x} \right] \prec \mathsf{E}_{\boldsymbol{x}, z \mid \boldsymbol{y}; \boldsymbol{\theta}} \left[ z^{-1} z \boldsymbol{\Sigma} z^{-1} \right] = \mathsf{E}_{\boldsymbol{x}, z \mid \boldsymbol{y}; \boldsymbol{\theta}} \left[ z^{-1} \right] \boldsymbol{\Sigma},$$

and hence $\nabla_{\boldsymbol{\mu}}^2 \log p(\boldsymbol{y}; \boldsymbol{\theta})$ is negative definite. Substituting $\nabla_{\boldsymbol{\mu}}^2 \log p(\boldsymbol{y}; \boldsymbol{\theta})$ in (48) into $\boldsymbol{I} + \left( \sum_{i=1}^n \frac{p_{-1}(\boldsymbol{y}_i; \boldsymbol{\theta})}{p(\boldsymbol{y}_i; \boldsymbol{\theta})} \right)^{-1} \sum_{t=1}^n \boldsymbol{\Sigma} \nabla_{\boldsymbol{\mu}}^2 \log p(\boldsymbol{y}_t; \boldsymbol{\theta})$, we have

$$\boldsymbol{I} + \left( \sum_{i=1}^n \frac{p_{-1}(\boldsymbol{y}_i; \boldsymbol{\theta})}{p(\boldsymbol{y}_i; \boldsymbol{\theta})} \right)^{-1} \sum_{t=1}^n \boldsymbol{\Sigma} \nabla_{\boldsymbol{\mu}}^2 \log p(\boldsymbol{y}_t; \boldsymbol{\theta}) = \frac{\sum_{t=1}^n \boldsymbol{\Sigma}^{-1} \mathsf{Cov}_{\boldsymbol{x}, z \mid \boldsymbol{y}_t; \boldsymbol{\theta}} \left[ z^{-1} \boldsymbol{x} \right]}{\sum_{t=1}^n \mathsf{E}_{z \mid \boldsymbol{y}_t; \boldsymbol{\theta}} [z^{-1}]}.$$

Since both $\boldsymbol{\Sigma}$ and $\mathsf{Cov}_{\boldsymbol{x}, z \mid \boldsymbol{y}; \boldsymbol{\theta}} \left[ z^{-1} \boldsymbol{x}_t \right]$ are positive definite matrices, $\boldsymbol{I} + \left( \sum_{i=1}^n \frac{p_{-1}(\boldsymbol{y}_i; \boldsymbol{\theta})}{p(\boldsymbol{y}_i; \boldsymbol{\theta})} \right)^{-1} \sum_{t=1}^n \boldsymbol{\Sigma} \nabla_{\boldsymbol{\mu}}^2 \log p(\boldsymbol{y}_t; \boldsymbol{\theta})$ is positive definite. $\square$

Based on Lemma 4, we have that all the eigenvalues of matrix $\boldsymbol{I} + \boldsymbol{\Sigma}\nabla_{\boldsymbol{\mu}}^2 \log p(\boldsymbol{y};\boldsymbol{\theta})$ for any $\boldsymbol{\theta}$ are belongs to $(0,1)$, hence we have

$$
\begin{aligned}
&\sup_{\beta \in (0,1]} \left\| \boldsymbol{I} + \left( \sum_{i=1}^{n} \frac{p_{-1}(\boldsymbol{y}_i; \tilde{\boldsymbol{\theta}}_{\mu})}{p(\boldsymbol{y}_i; \tilde{\boldsymbol{\theta}}_{\mu})} \right)^{-1} \sum_{t=1}^{n} \underline{\boldsymbol{\Sigma}} \nabla_{\tilde{\boldsymbol{\mu}}}^2 \log p(\boldsymbol{y}_t; \tilde{\boldsymbol{\theta}}_{\mu}) \right\|_2 \\
&= \sup_{\beta \in (0,1]} \left\| \frac{\sum_{t=1}^{n} \underline{\boldsymbol{\Sigma}}^{-1} \mathsf{Cov}_{\boldsymbol{x},z|\boldsymbol{y}_t;\tilde{\boldsymbol{\theta}}_{\mu}} \left[ z^{-1}\boldsymbol{x} \right]}{\sum_{t=1}^{n} \mathsf{E}_{z|\boldsymbol{y}_t;\tilde{\boldsymbol{\theta}}_{\mu}}[z^{-1}]} \right\|_2 \\
&\leq \max_{\boldsymbol{\theta}} \left\| \frac{\sum_{t=1}^{n} \boldsymbol{\Sigma}^{-1} \mathsf{Cov}_{\boldsymbol{x},z|\boldsymbol{y}_t;\boldsymbol{\theta}} \left[ z^{-1}\boldsymbol{x} \right]}{\sum_{t=1}^{n} \mathsf{E}_{z|\boldsymbol{y}_t;\boldsymbol{\theta}}[z^{-1}]} \right\|_2 \\
&\triangleq c_{\mu} \in (0,1).
\end{aligned}
$$

$\square$

### B.2 Convergence Rate of Skewness Parameter

Given the update rule of $\boldsymbol{\xi}$ in (12), we have

$$
\boldsymbol{\xi} = \frac{\sum_{t=1}^{n}(\boldsymbol{w}_t - \boldsymbol{\mu})}{\sum_{t=1}^{n} \zeta_t} = \frac{\sum_{t=1}^{n}(\mathsf{E}_{\boldsymbol{x}|\boldsymbol{y}_t;\underline{\boldsymbol{\theta}}}[\boldsymbol{x}] - \boldsymbol{\mu})}{\sum_{t=1}^{n} \mathsf{E}_{z_t|\boldsymbol{y}_t;\underline{\boldsymbol{\theta}}}[z]}. \tag{49}
$$

Based on (38), we have

$$
\mathsf{E}_{\boldsymbol{x}|\boldsymbol{y};\boldsymbol{\theta}}[\boldsymbol{x}] - \boldsymbol{\mu} = \mathsf{E}_{z|\boldsymbol{y};\boldsymbol{\theta}}[z]\boldsymbol{\xi} + p^{-1}(\boldsymbol{y};\boldsymbol{\theta})\boldsymbol{\Sigma}\boldsymbol{q}_1.
$$

Hence, the update rule of $\boldsymbol{\xi}$ in (49) becomes

$$
\boldsymbol{\xi} = \underline{\boldsymbol{\xi}} + \left( \sum_{i=1}^{n} \frac{p_1(\boldsymbol{y}_i;\boldsymbol{\theta})}{p(\boldsymbol{y}_i;\underline{\boldsymbol{\theta}})} \right)^{-1} \sum_{t=1}^{n} p^{-1}(\boldsymbol{y}_t;\underline{\boldsymbol{\theta}})\underline{\boldsymbol{\Sigma}}\boldsymbol{q}_{1,t} \tag{50}
$$

Hence, the distance between $\boldsymbol{\xi}$ at the current iteration and $\boldsymbol{\xi}^{\star}$ is given by

$$
\|\boldsymbol{\xi} - \boldsymbol{\xi}^{\star}\|_2 = \left\| \underline{\boldsymbol{\xi}} - \boldsymbol{\xi}^{\star} + \left( \sum_{i=1}^{n} \frac{p_1(\boldsymbol{y}_i;\boldsymbol{\theta})}{p(\boldsymbol{y}_i;\underline{\boldsymbol{\theta}})} \right)^{-1} \sum_{t=1}^{n} p^{-1}(\boldsymbol{y}_t;\underline{\boldsymbol{\theta}})\underline{\boldsymbol{\Sigma}}\boldsymbol{q}_{1,t} \right\|_2. \tag{51}
$$

To further analyze the term on the right side in (51), we first give a result of $\boldsymbol{q}_1$.

**Lemma 7.** *When $\boldsymbol{x}$ follows a normal mean-variance mixture and $\boldsymbol{q}_1$ is defined in (21), we have*

$$
\boldsymbol{q}_1 = \nabla_{\boldsymbol{\xi}} p(\boldsymbol{y};\boldsymbol{\theta}).
$$

*Proof.* Consider the partial derivative of $p(\boldsymbol{y};\boldsymbol{\theta})$ with respect to the $i$-th element of $\boldsymbol{\xi}$. We have that

$$
\nabla_{\xi_i} p(\boldsymbol{y};\boldsymbol{\theta}) = \nabla_{\xi_i} \int_{\mathcal{Q}^{-1}(\boldsymbol{y})} p(\boldsymbol{x};\boldsymbol{\theta})\mathrm{d}\boldsymbol{x} = \int_{\mathcal{Q}^{-1}(\boldsymbol{y})} \nabla_{\xi_i} p(\boldsymbol{x};\boldsymbol{\theta})\mathrm{d}\boldsymbol{x}.
$$

Since the partial derivatives of $p(\boldsymbol{x};\boldsymbol{\theta})$

$$
\nabla_{\xi_i} p(\boldsymbol{x} \mid z;\boldsymbol{\theta}) = -z\nabla_{x_i} p(\boldsymbol{x} \mid z;\boldsymbol{\theta}),
$$

we can obtain that

$$
\begin{aligned}
\nabla_{\xi_i} p(\boldsymbol{y};\boldsymbol{\theta}) &= -\int_{0}^{+\infty} \int_{\mathcal{Q}^{-1}(\boldsymbol{y}_{\setminus i})} \int_{l_i}^{u_i} z\nabla_{x_i} p(\boldsymbol{x} \mid z;\boldsymbol{\theta})p(z)\mathrm{d}x_i\mathrm{d}\boldsymbol{x}_{\setminus i}\mathrm{d}z \\
&= \int_{0}^{+\infty} \int_{\mathcal{Q}^{-1}(\boldsymbol{y}_{\setminus i})} \frac{p(z)}{z} z \left( p(\boldsymbol{x}_{\setminus i}, x_i = l_i \mid z;\boldsymbol{\theta}) - p(\boldsymbol{x}_{\setminus i}, x_i = u_i \mid z;\boldsymbol{\theta}) \right) \mathrm{d}\boldsymbol{x}_{\setminus i}\mathrm{d}z \\
&= p(x_i = l_i, \boldsymbol{y}_{\setminus i} \mid z,\boldsymbol{\theta}) - p(x_i = u_i, \boldsymbol{y}_{\setminus i} \mid z,\boldsymbol{\theta}).
\end{aligned}
$$

which is equivalent to the definition of the $i$-th element of $\boldsymbol{q}_1$ from (33).

Based on Lemma 7, we have $p^{-1}(\boldsymbol{y}; \boldsymbol{\theta})\boldsymbol{q}_1 = \nabla_{\boldsymbol{\xi}} \log p(\boldsymbol{y}; \boldsymbol{\theta})$ and $\boldsymbol{q}_1^\star = \boldsymbol{0}$. Then the distance (44) becomes

$$
\begin{aligned}
\|\boldsymbol{\xi} - \boldsymbol{\xi}^\star\|_2 &= \left\| \underline{\boldsymbol{\xi}} - \boldsymbol{\xi}^\star + \left( \sum_{i=1}^n \frac{p_1(\boldsymbol{y}_i; \underline{\boldsymbol{\theta}})}{p(\boldsymbol{y}_i; \underline{\boldsymbol{\theta}})} \right)^{-1} \sum_{t=1}^n \underline{\boldsymbol{\Sigma}} \nabla_{\underline{\boldsymbol{\xi}}} \log p(\boldsymbol{y}; \underline{\boldsymbol{\theta}}) \right. \\
&\qquad \left. - \left( \sum_{i=1}^n \frac{p_1(\boldsymbol{y}_i; \boldsymbol{\theta}^\star)}{p(\boldsymbol{y}_i; \boldsymbol{\theta}^\star)} \right)^{-1} \sum_{t=1}^n \boldsymbol{\Sigma}^\star \nabla_{\boldsymbol{\xi}^\star} \log p(\boldsymbol{y}; \boldsymbol{\theta}^\star) \right\|_2 \\
&= \left\| \underline{\boldsymbol{\xi}} - \boldsymbol{\xi}^\star + \int_0^1 \left( \sum_{i=1}^n \frac{p_1(\boldsymbol{y}_i; \tilde{\boldsymbol{\theta}}_\xi)}{p(\boldsymbol{y}_i; \tilde{\boldsymbol{\theta}}_\xi)} \right)^{-1} \sum_{t=1}^n \tilde{\boldsymbol{\Sigma}} \nabla_{\tilde{\boldsymbol{\xi}}}^2 \log p(\boldsymbol{y}_t; \tilde{\boldsymbol{\theta}}_\xi)(\tilde{\boldsymbol{\xi}} - \boldsymbol{\xi}^\star) \mathrm{d}\beta \right\|_2 \\
&\leq \sup_{\beta \in (0,1]} \left\| \boldsymbol{I} + \left( \sum_{i=1}^n \frac{p_1(\boldsymbol{y}_i; \tilde{\boldsymbol{\theta}}_\xi)}{p(\boldsymbol{y}_i; \tilde{\boldsymbol{\theta}}_\xi)} \right)^{-1} \sum_{t=1}^n \tilde{\boldsymbol{\Sigma}} \nabla_{\tilde{\boldsymbol{\xi}}}^2 \log p(\boldsymbol{y}_t; \tilde{\boldsymbol{\theta}}_\xi) \right\|_2 \left\| \underline{\boldsymbol{\xi}} - \boldsymbol{\xi}^\star \right\|_2,
\end{aligned}
$$

where the second equation is given by the mean value theorem of integrals and $\tilde{\theta}_\xi = \left\{ \underline{\boldsymbol{\mu}}, \underline{\boldsymbol{\Sigma}}, \tilde{\boldsymbol{\xi}} \right\}$ with $\tilde{\boldsymbol{\xi}} = \beta \underline{\boldsymbol{\xi}} + (1-\beta)\boldsymbol{\xi}^\star$, and $\beta \in (0, 1]$.

To bound the term $\sup_{\beta \in (0,1]} \left\| \boldsymbol{I} + \left( \sum_{i=1}^n \frac{p_{-1}(\boldsymbol{y}_i; \tilde{\boldsymbol{\theta}}_\xi)}{p(\boldsymbol{y}_i; \tilde{\boldsymbol{\theta}}_\xi)} \right)^{-1} \sum_{t=1}^n \tilde{\boldsymbol{\Sigma}} \nabla_{\tilde{\boldsymbol{\xi}}}^2 \log p(\boldsymbol{y}_t; \tilde{\boldsymbol{\theta}}_\xi) \right\|_2$, we need some results for the second order derivative of $\log p(\boldsymbol{y}_t; \boldsymbol{\theta})$ at first.

**Lemma 8.** *The second order derivative of $\log p(\boldsymbol{y}; \boldsymbol{\theta})$ with respect to $\boldsymbol{\xi}$, i.e., $\nabla_{\boldsymbol{\xi}}^2 \log p(\boldsymbol{y}; \boldsymbol{\theta})$, satisfy*

    *1. $\nabla_{\boldsymbol{\xi}}^2 \log p(\boldsymbol{y}; \boldsymbol{\theta})$ is a negative definite matrix;*

    *2. $\boldsymbol{I} + \left( \sum_{i=1}^n \frac{p_{-1}(\boldsymbol{y}_i; \boldsymbol{\theta})}{p(\boldsymbol{y}_i; \boldsymbol{\theta})} \right)^{-1} \sum_{t=1}^n \boldsymbol{\Sigma} \nabla_{\boldsymbol{\xi}}^2 \log p(\boldsymbol{y}_t; \boldsymbol{\theta})$ is a positive definite matrix.*

*Proof.* The second order derivative of $\log p(\boldsymbol{y}; \boldsymbol{\theta})$ with respect to $\boldsymbol{\xi}$ is given as

$$
\nabla_{\boldsymbol{\xi}}^2 \log p(\boldsymbol{y}; \boldsymbol{\theta}) = \nabla_{\boldsymbol{\xi}} \left( p^{-1}(\boldsymbol{y}; \boldsymbol{\theta}) \nabla_{\boldsymbol{\xi}} p(\boldsymbol{y}; \boldsymbol{\theta}) \right) = \frac{\nabla_{\boldsymbol{\xi}}^2 p(\boldsymbol{y}; \boldsymbol{\theta}) \cdot p(\boldsymbol{y}; \boldsymbol{\theta}) - \nabla_{\boldsymbol{\xi}} p(\boldsymbol{y}; \boldsymbol{\theta}) \nabla_{\boldsymbol{\xi}}^\top p(\boldsymbol{y}; \boldsymbol{\theta})}{p^2(\boldsymbol{y}; \boldsymbol{\theta})}.
$$
(52)

To analyze the above expression, we should compute the first and second order derivatives of $p(\boldsymbol{y}; \boldsymbol{\theta})$ with respect to $\boldsymbol{\xi}$ at first. The first order one is given as follows:

$$
\nabla_{\boldsymbol{\xi}} p(\boldsymbol{y}; \boldsymbol{\theta}) = \nabla_{\boldsymbol{\xi}} \int_{\mathcal{Q}^{-1}(\boldsymbol{y})} p(\boldsymbol{x}; \boldsymbol{\theta}) \mathrm{d}\boldsymbol{x} = \int_{\mathcal{Q}^{-1}(\boldsymbol{y})} \nabla_{\boldsymbol{\xi}} p(\boldsymbol{x}; \boldsymbol{\theta}) \mathrm{d}\boldsymbol{x}.
$$
(53)

Since $p(\boldsymbol{x}; \boldsymbol{\theta}) = \int_0^{+\infty} p(z) p(\boldsymbol{x} \mid z; \boldsymbol{\theta}) \mathrm{d}z$, (53) becomes

$$
\begin{aligned}
\nabla_{\boldsymbol{\xi}} p(\boldsymbol{y}; \boldsymbol{\theta}) &= \int_{\mathcal{Q}^{-1}(\boldsymbol{y})} \int_0^{+\infty} p(z) \nabla_{\boldsymbol{\xi}} p(\boldsymbol{x} \mid z; \boldsymbol{\theta}) \mathrm{d}z \mathrm{d}\boldsymbol{x} \\
&= \int_{\mathcal{Q}^{-1}(\boldsymbol{y})} \int_0^{+\infty} p(z) p(\boldsymbol{x} \mid z; \boldsymbol{\theta}) \boldsymbol{\Sigma}^{-1}(\boldsymbol{x} - \boldsymbol{\mu} - z\boldsymbol{\xi}) \mathrm{d}z \mathrm{d}\boldsymbol{x}.
\end{aligned}
$$

Then the second order derivative can be further computed as

$$
\begin{aligned}
\nabla_{\boldsymbol{\xi}}^2 p(\boldsymbol{y}; \boldsymbol{\theta}) &= \int_{\mathcal{Q}^{-1}(\boldsymbol{y})} \int_0^{+\infty} p(z) \nabla_{\boldsymbol{\xi}} \left( p(\boldsymbol{x} \mid z; \boldsymbol{\theta}) \boldsymbol{\Sigma}^{-1}(\boldsymbol{x} - \boldsymbol{\mu} - z\boldsymbol{\xi}) \right) \mathrm{d}z \mathrm{d}\boldsymbol{x} \\
&= \int_{\mathcal{Q}^{-1}(\boldsymbol{y})} \int_0^{+\infty} \boldsymbol{\Sigma}^{-1} p(\boldsymbol{x} \mid z; \boldsymbol{\theta}) p(z) \left( -z\boldsymbol{I} + (\boldsymbol{x} - \boldsymbol{\mu} - z\boldsymbol{\xi})(\boldsymbol{x} - \boldsymbol{\mu} - z\boldsymbol{\xi})^\top \boldsymbol{\Sigma}^{-1} \right) \mathrm{d}z \mathrm{d}\boldsymbol{x}.
\end{aligned}
$$
(54)

Based on Lemma 5, the derivatives of $p(\boldsymbol{y}; \boldsymbol{\theta})$ becomes

$$\nabla_{\boldsymbol{\xi}} p(\boldsymbol{y}; \boldsymbol{\theta}) = p(\boldsymbol{y}; \boldsymbol{\theta}) \boldsymbol{\Sigma}^{-1} \mathsf{E}_{\boldsymbol{x}, z | \boldsymbol{y}; \boldsymbol{\theta}} \left[ \boldsymbol{x} - \boldsymbol{\mu} - z\boldsymbol{\xi} \right],$$

and

$$\nabla_{\boldsymbol{\xi}}^2 p(\boldsymbol{y}; \boldsymbol{\theta}) = p(\boldsymbol{y}; \boldsymbol{\theta}) \boldsymbol{\Sigma}^{-1} \left( -\mathsf{E}_{z | \boldsymbol{y}; \boldsymbol{\theta}} \left[ z \right] \boldsymbol{I} + \mathsf{E}_{\boldsymbol{x}, z | \boldsymbol{y}; \boldsymbol{\theta}} \left[ (\boldsymbol{x} - \boldsymbol{\mu} - z\boldsymbol{\xi})(\boldsymbol{x} - \boldsymbol{\mu} - z\boldsymbol{\xi})^\top \right] \boldsymbol{\Sigma}^{-1} \right).$$

Then the second order derivative of $\log p(\boldsymbol{y}; \boldsymbol{\theta})$ in (52) becomes

$$\nabla_{\boldsymbol{\xi}}^2 \log p(\boldsymbol{y}; \boldsymbol{\theta}) = \boldsymbol{\Sigma}^{-1} \left( -\mathsf{E}_{z | \boldsymbol{y}; \boldsymbol{\theta}} \left[ z \right] \boldsymbol{\Sigma} + \mathsf{Cov}_{\boldsymbol{x}, z | \boldsymbol{y}; \boldsymbol{\theta}} \left[ \boldsymbol{x} - z\boldsymbol{\xi} \right] \right) \boldsymbol{\Sigma}^{-1}. \tag{55}$$

Setting $\tilde{\boldsymbol{x}} = \boldsymbol{x}, z \mid \boldsymbol{y}$ and $f(\boldsymbol{x}) = \boldsymbol{x} - z\boldsymbol{\xi}$, since $\nabla_{\boldsymbol{x}}^2 \log p(\boldsymbol{x}, z \mid \boldsymbol{y}; \boldsymbol{\theta}) = -\frac{\boldsymbol{\Sigma}^{-1}}{z}$, $p(\boldsymbol{x}, z \mid \boldsymbol{y}; \boldsymbol{\theta})$ is a log-concave function with respect to $\boldsymbol{x}$. Based on the Brascamp–Lieb inequality in Lemma 6, we have

$$\mathsf{Cov}_{\boldsymbol{x}, z | \boldsymbol{y}; \boldsymbol{\theta}} \left[ \boldsymbol{x} - z\boldsymbol{\xi} \right] \prec \mathsf{E}_{\boldsymbol{x}, z | \boldsymbol{y}; \boldsymbol{\theta}} \left[ z\boldsymbol{\Sigma} \right] = \mathsf{E}_{\boldsymbol{x}, z | \boldsymbol{y}; \boldsymbol{\theta}} \left[ z \right] \boldsymbol{\Sigma},$$

and hence $\nabla_{\boldsymbol{\xi}}^2 p(\boldsymbol{y}; \boldsymbol{\theta})$ is negative definite.

Substituting $\nabla_{\boldsymbol{\xi}}^2 \log p(\boldsymbol{y}; \boldsymbol{\theta})$ in (55) into $\boldsymbol{I} + \left( \sum_{i=1}^n \frac{p_{-1}(\boldsymbol{y}_i; \boldsymbol{\theta})}{p(\boldsymbol{y}_i; \boldsymbol{\theta})} \right)^{-1} \sum_{t=1}^n \boldsymbol{\Sigma} \nabla_{\boldsymbol{\xi}}^2 \log p(\boldsymbol{y}_t; \boldsymbol{\theta})$, we have

$$\boldsymbol{I} + \left( \sum_{i=1}^n \frac{p_1(\boldsymbol{y}_i; \boldsymbol{\theta})}{p(\boldsymbol{y}_i; \boldsymbol{\theta})} \right)^{-1} \sum_{t=1}^n \boldsymbol{\Sigma} \nabla_{\boldsymbol{\xi}}^2 \log p(\boldsymbol{y}_t; \boldsymbol{\theta}) = \frac{\sum_{t=1}^n \boldsymbol{\Sigma}^{-1} \mathsf{Cov}_{\boldsymbol{x}, z | \boldsymbol{y}_t; \boldsymbol{\theta}} \left[ \boldsymbol{x} - z\boldsymbol{\xi} \right]}{\sum_{t=1}^n \mathsf{E}_{z | \boldsymbol{y}_t; \boldsymbol{\theta}}[z]}.$$

Since both $\boldsymbol{\Sigma}$ and $\mathsf{Cov}_{\boldsymbol{x}, z | \boldsymbol{y}; \boldsymbol{\theta}} \left[ \boldsymbol{x}_t - z\boldsymbol{\xi} \right]$ are positive definite matrices, $\boldsymbol{I} + \left( \sum_{i=1}^n \frac{p_{-1}(\boldsymbol{y}_i; \boldsymbol{\theta})}{p(\boldsymbol{y}_i; \boldsymbol{\theta})} \right)^{-1} \sum_{t=1}^n \boldsymbol{\Sigma} \nabla_{\boldsymbol{\xi}}^2 \log p(\boldsymbol{y}_t; \boldsymbol{\theta})$ is positive definite. $\square$

Based on Lemma 8, we have that all the eigenvalues of matrix $\boldsymbol{I} + \boldsymbol{\Sigma} \nabla_{\boldsymbol{\xi}}^2 \log p(\boldsymbol{y}; \boldsymbol{\theta})$ for any $\boldsymbol{\theta}$ are belong to $(0, 1)$, hence we have

$$\sup_{\beta \in (0, 1]} \left\| \boldsymbol{I} + \left( \sum_{i=1}^n \frac{p_1(\boldsymbol{y}_i; \tilde{\boldsymbol{\theta}}_\xi)}{p(\boldsymbol{y}_i; \tilde{\boldsymbol{\theta}}_\xi)} \right)^{-1} \sum_{t=1}^n \underline{\boldsymbol{\Sigma}} \nabla_{\tilde{\boldsymbol{\xi}}}^2 \log p(\boldsymbol{y}_t; \tilde{\boldsymbol{\theta}}_\xi) \right\|_2$$

$$= \sup_{\beta \in (0, 1]} \left\| \frac{\sum_{t=1}^n \boldsymbol{\Sigma}^{-1} \mathsf{Cov}_{\boldsymbol{x}, z | \boldsymbol{y}_t; \tilde{\boldsymbol{\theta}}_\xi} \left[ \boldsymbol{x} - z\boldsymbol{\xi} \right]}{\sum_{t=1}^n \mathsf{E}_{z | \boldsymbol{y}_t; \tilde{\boldsymbol{\theta}}_\xi}[z]} \right\|_2$$

$$\leq \max_{\boldsymbol{\theta}} \left\| \frac{\sum_{t=1}^n \boldsymbol{\Sigma}^{-1} \mathsf{Cov}_{\boldsymbol{x}, z | \boldsymbol{y}_t; \boldsymbol{\theta}} \left[ \boldsymbol{x} - z\boldsymbol{\xi} \right]}{\sum_{t=1}^n \mathsf{E}_{z | \boldsymbol{y}_t; \boldsymbol{\theta}}[z]} \right\|_2$$

$$\triangleq c_\xi \in (0, 1).$$

$\square$

## B.3 CONVERGENCE RATE OF SCATTER MATRIX

Given the update rule of $\boldsymbol{\Sigma}$ in (12), we have

$$\boldsymbol{\Sigma} = \frac{1}{n} \sum_{t=1}^n \left( \left( \boldsymbol{U}_t - 2\boldsymbol{v}_t \underline{\boldsymbol{\mu}}^\top + \iota_t \underline{\boldsymbol{\mu}} \underline{\boldsymbol{\mu}}^\top \right) - 2(\boldsymbol{w}_t - \underline{\boldsymbol{\mu}}) \underline{\boldsymbol{\xi}}^\top + \zeta_t \underline{\boldsymbol{\xi}} \underline{\boldsymbol{\xi}}^\top \right)$$

$$= \frac{1}{n} \sum_{t=1}^n \left( \mathsf{E}_{\boldsymbol{x}_t, z_t | \boldsymbol{y}_t; \underline{\boldsymbol{\theta}}}[z^{-1} \boldsymbol{x} \boldsymbol{x}^\top] - 2\mathsf{E}_{\boldsymbol{x}_t, z_t | \boldsymbol{y}_t; \underline{\boldsymbol{\theta}}}[z^{-1} \boldsymbol{x}_t] \underline{\boldsymbol{\mu}}^\top + \mathsf{E}_{z_t | \boldsymbol{y}_t; \underline{\boldsymbol{\theta}}}[z^{-1}] \underline{\boldsymbol{\mu}} \underline{\boldsymbol{\mu}}^\top \right. \tag{56}$$

$$\left. - 2(\mathsf{E}_{\boldsymbol{x} | \boldsymbol{y}_t; \underline{\boldsymbol{\theta}}}[\boldsymbol{x}] - \underline{\boldsymbol{\mu}}) \underline{\boldsymbol{\xi}}^\top + \mathsf{E}_{z_t | \boldsymbol{y}_t; \underline{\boldsymbol{\theta}}}[z] \underline{\boldsymbol{\xi}} \underline{\boldsymbol{\xi}}^\top \right).$$

In the following, we derive all five terms in the summation in (56).

**Term 1**: Based on (41), we have

$$\mathsf{E}_{\boldsymbol{x},z|\boldsymbol{y};\underline{\boldsymbol{\theta}}}\left[z^{-1}\boldsymbol{x}\boldsymbol{x}^\top\right] = \mathsf{E}_{z|\boldsymbol{y};\underline{\boldsymbol{\theta}}}\left[z^{-1}\right]\underline{\boldsymbol{\mu}}\underline{\boldsymbol{\mu}}^\top + \mathsf{E}_{z|\boldsymbol{y};\underline{\boldsymbol{\theta}}}\left[z\right]\underline{\boldsymbol{\xi}}\underline{\boldsymbol{\xi}}^\top + 2\underline{\boldsymbol{\mu}}\underline{\boldsymbol{\xi}}^\top + \underline{\boldsymbol{\Sigma}}$$
$$+ p^{-1}(\boldsymbol{y};\underline{\boldsymbol{\theta}})\left(2\left(\underline{\boldsymbol{\Sigma}}\boldsymbol{q}\right)\underline{\boldsymbol{\mu}}^\top + 2\mathsf{E}_z\left[z\right]\left(\underline{\boldsymbol{\Sigma}}\boldsymbol{q}_1\right)\underline{\boldsymbol{\xi}}^\top + \underline{\boldsymbol{\Sigma}}\left(\boldsymbol{H}_1 + \boldsymbol{D}_1\right)\underline{\boldsymbol{\Sigma}}\right),$$

where $\boldsymbol{q}$, $\boldsymbol{q}_1$, $\boldsymbol{H}_1$ and $\boldsymbol{D}_1$ are defined in (21), (32), (34) and (37), respectively.

**Term 2**: Based on (40), we can obtain

$$-2\mathsf{E}_{\boldsymbol{x}_t,z_t|\boldsymbol{y}_t;\underline{\boldsymbol{\theta}}}[z_t^{-1}\boldsymbol{x}_t]\underline{\boldsymbol{\mu}}^\top = -2\left(\mathsf{E}_{z|\boldsymbol{y};\underline{\boldsymbol{\theta}}}\left[z^{-1}\right]\underline{\boldsymbol{\mu}}\underline{\boldsymbol{\mu}}^\top + \underline{\boldsymbol{\xi}} + p^{-1}(\boldsymbol{y};\underline{\boldsymbol{\theta}})\underline{\boldsymbol{\Sigma}}\boldsymbol{q}\right)\underline{\boldsymbol{\mu}}^\top.$$

**Term 4**: Based on (38), we have

$$-2(\mathsf{E}_{\boldsymbol{x}_t|\boldsymbol{y}_t;\underline{\boldsymbol{\theta}}}[\boldsymbol{x}_t] - \underline{\boldsymbol{\mu}})\underline{\boldsymbol{\xi}}^\top = -2\mathsf{E}_{z|\boldsymbol{y};\underline{\boldsymbol{\theta}}}\left[z\right]\underline{\boldsymbol{\xi}}\underline{\boldsymbol{\xi}}^\top - 2p^{-1}(\boldsymbol{y};\boldsymbol{\theta})\left(\underline{\boldsymbol{\Sigma}}\boldsymbol{q}_1\right)\underline{\boldsymbol{\xi}}^\top$$

Upon cancellation of the opposing terms, the update rule in (56) reduces to:

$$\boldsymbol{\Sigma} = \underline{\boldsymbol{\Sigma}} + \frac{1}{n}\sum_{t=1}^{n}p^{-1}(\boldsymbol{y}_t;\underline{\boldsymbol{\theta}})\underline{\boldsymbol{\Sigma}}(\boldsymbol{H}_{1,t} + \boldsymbol{D}_{1,t})\underline{\boldsymbol{\Sigma}}, \tag{57}$$

Hence, the distance between $\boldsymbol{\Sigma}$ at the current iteration and $\boldsymbol{\Sigma}^\star$ is given by

$$\|\boldsymbol{\Sigma} - \boldsymbol{\Sigma}^\star\|_{\mathsf{F}} = \left\|\underline{\boldsymbol{\Sigma}} - \boldsymbol{\Sigma}^\star + \frac{1}{n}\sum_{t=1}^{n}p^{-1}(\boldsymbol{y}_t;\underline{\boldsymbol{\theta}})\underline{\boldsymbol{\Sigma}}(\boldsymbol{H}_{1,t} + \boldsymbol{D}_{1,t})\underline{\boldsymbol{\Sigma}}\right\|_{\mathsf{F}}. \tag{58}$$

To further analyze the term on the right side, we first give a result of $\boldsymbol{H}_{1,t} + \boldsymbol{D}_{1,t}$.

**Lemma 9.** *When $\boldsymbol{x}$ follows a normal mean variance mixture and $\boldsymbol{H}_1$ and $\boldsymbol{D}_1$ are defined in (34) and (37), respectively, we have*

$$\frac{1}{2}(\boldsymbol{H}_1 + \boldsymbol{D}_1) = \nabla_{\boldsymbol{\Sigma}}p(\boldsymbol{y};\boldsymbol{\theta}).$$

*Proof.* Considering the partial derivative of $p(\boldsymbol{y};\boldsymbol{\theta})$ with respect to the $(i,j)$-th element of $\boldsymbol{\Sigma}$, we have that

$$\nabla_{\boldsymbol{\Sigma}_{ij}}p(\boldsymbol{y};\boldsymbol{\theta}) = \nabla_{\boldsymbol{\Sigma}_{ij}}\int_{\mathcal{Q}^{-1}(\boldsymbol{y})}p(\boldsymbol{x};\boldsymbol{\theta})\mathrm{d}\boldsymbol{x} = \int_{\mathcal{Q}^{-1}(\boldsymbol{y})}\nabla_{\boldsymbol{\Sigma}_{ij}}p(\boldsymbol{x};\boldsymbol{\theta})\mathrm{d}\boldsymbol{x}.$$

The partial derivatives of $p(\boldsymbol{x};\boldsymbol{\theta})$ satisfy the following equation:

$$\nabla_{\boldsymbol{\Sigma}_{ij}}p(\boldsymbol{x}\mid z;\boldsymbol{\theta}) = \frac{z}{2}\nabla_{x_i x_j}^2 p(\boldsymbol{x}\mid z;\boldsymbol{\theta}), \tag{59}$$

Hence, for $i \neq j$, we can obtain that

$$\nabla_{\boldsymbol{\Sigma}_{ij}}p(\boldsymbol{y};\boldsymbol{\theta}) = \int_{\mathcal{Q}^{-1}(\boldsymbol{y}_{\backslash i,j})}\int_{l_i}^{u_i}\int_{l_j}^{u_j}\int_{0}^{+\infty}\frac{z}{2}\nabla_{x_i x_j}^2 p(\boldsymbol{x}\mid z;\boldsymbol{\theta})p(z)\mathrm{d}z\mathrm{d}x_i\mathrm{d}x_j\mathrm{d}\boldsymbol{x}_{\backslash i,j}$$

$$= \int_{\mathcal{Q}^{-1}(\boldsymbol{y}_{\backslash i,j})}\int_{0}^{+\infty}\frac{zp(z)}{2}\left(p(\boldsymbol{x}_{\backslash i,j}, x_i = l_i, x_j = l_j \mid z;\boldsymbol{\theta}) - p(\boldsymbol{x}_{\backslash i,j}, x_i = l_i, x_j = u_j \mid z;\boldsymbol{\theta})\right.$$

$$\left.-p(\boldsymbol{x}_{\backslash i,j}, x_i = u_i, x_j = l_j \mid z;\boldsymbol{\theta}) + p(\boldsymbol{x}_{\backslash i,j}, x_i = u_i, x_j = u_j \mid z;\boldsymbol{\theta})\right)\mathrm{d}z\mathrm{d}\boldsymbol{x}_{\backslash i,j}$$

$$= \frac{1}{2}\mathsf{E}_z(z)\left(p_1(\boldsymbol{y}_{\backslash i,j}, x_i = l_i, x_j = l_j;\boldsymbol{\theta}) - p_1(\boldsymbol{y}_{\backslash i,j}, x_i = l_i, x_j = u_j;\boldsymbol{\theta})\right.$$

$$\left.-p_1(\boldsymbol{y}_{\backslash i,j}, x_i = u_i, x_j = l_j;\boldsymbol{\theta}) + p_1(\boldsymbol{y}_{\backslash i,j}, x_i = u_i, x_j = u_j;\boldsymbol{\theta})\right)$$

$$= \frac{1}{2}\boldsymbol{H}_{1,ij}.$$

For the case $i = j$, we first introduce a result.

**Lemma 10.** *When $\boldsymbol{x}$ follows a normal mean-variance mixture, we have the following equation:*

$$\sum_{k=1}^{d}\boldsymbol{\Sigma}_{ik}\nabla_{x_k x_i}^2 p_1(\boldsymbol{x}\mid z;\boldsymbol{\theta}) = \nabla_{x_i}\left(-\frac{p_1(\boldsymbol{x}\mid z;\boldsymbol{\theta})}{z}(x_i - \mu_i - z\xi_i)\right). \tag{60}$$

*Proof.* We begin with

$$\sum_{k=1}^{d} \boldsymbol{\Sigma}_{ik} \nabla_{x_k x_i}^2 p_1(\boldsymbol{x} \mid z; \boldsymbol{\theta}) = \nabla_{x_i} \left( \sum_{k=1}^{d} \boldsymbol{\Sigma}_{ik} \nabla_{x_k} p_1(\boldsymbol{x} \mid z; \boldsymbol{\theta}) \right)$$

For the term $\sum_{k=1}^{d} \boldsymbol{\Sigma}_{ik} \nabla_{x_k} p(\boldsymbol{x} \mid z; \boldsymbol{\theta})$, we have

$$\sum_{k=1}^{d} \boldsymbol{\Sigma}_{ik} \nabla_{x_k} p_1(\boldsymbol{x} \mid z; \boldsymbol{\theta}) = \sum_{k=1}^{d} \boldsymbol{\Sigma}_{ik} \left( -\frac{p_1(\boldsymbol{x} \mid z; \boldsymbol{\theta})}{z} \sum_{l=1}^{d} \boldsymbol{\Sigma}_{kl}^{-1} (x_l - \mu_l - z\xi_l) \right)$$

$$= -\frac{p_1(\boldsymbol{x} \mid z; \boldsymbol{\theta})}{z} \sum_{l=1}^{d} \left( \sum_{k=1}^{d} \boldsymbol{\Sigma}_{ik} \boldsymbol{\Sigma}_{kl}^{-1} (x_l - \mu_l - z\xi_l) \right).$$

Since $\sum_{k=1}^{d} \boldsymbol{\Sigma}_{ik} \boldsymbol{\Sigma}_{kl}^{-1}$ is equivalent to the $(i,l)$-entry of the matrix $\boldsymbol{\Sigma}\boldsymbol{\Sigma}^{-1}$, we can obtain that

$$\sum_{k=1}^{d} \boldsymbol{\Sigma}_{ik} \boldsymbol{\Sigma}_{kl}^{-1} = \begin{cases} 0, & i \neq l, \\ 1, & i = l. \end{cases}$$

Hence, we have

$$-\frac{p_1(\boldsymbol{x} \mid z; \boldsymbol{\theta})}{z} \sum_{k=1}^{d} \left( \sum_{l=1}^{d} \boldsymbol{\Sigma}_{ik} \boldsymbol{\Sigma}_{kl}^{-1} (x_l - \mu_l - z\xi_l) \right) = -\frac{p_1(\boldsymbol{x} \mid z; \boldsymbol{\theta})}{z} (x_i - \mu_i - z\xi_i).$$

Therefore, we prove the equation (60). $\qquad\square$

Computing the integral of the left side of (60), we have

$$\int_{\mathcal{Q}^{-1}(\boldsymbol{y})} \int_0^{+\infty} \sum_{k=1}^{d} \boldsymbol{\Sigma}_{ik} \nabla_{x_k x_i}^2 p_1(\boldsymbol{x} \mid z; \boldsymbol{\theta}) p(z) \mathrm{d}z \mathrm{d}\boldsymbol{x}$$

$$= \int_0^{+\infty} \int_{\mathcal{Q}^{-1}(\boldsymbol{y})} \left( \sigma_i \nabla_{x_i}^2 p_1(\boldsymbol{x} \mid z; \boldsymbol{\theta}) + \sum_{k \neq i} \boldsymbol{\Sigma}_{ik} \nabla_{x_k x_i}^2 p_1(\boldsymbol{x} \mid z; \boldsymbol{\theta}) \right) p(z) \mathrm{d}z \mathrm{d}\boldsymbol{x} \quad (61)$$

$$= 2\mathsf{E}_z[z^{-1}]\sigma_i \nabla_{\sigma_i} p(\boldsymbol{y}; \boldsymbol{\theta}) + [\boldsymbol{\Sigma}\boldsymbol{H}_1]_{ii}.$$

Then the integral of the right side of (60) is given by

$$\int_{\mathcal{Q}^{-1}(\boldsymbol{y})} \int_0^{+\infty} \nabla_{x_i} \left( -\frac{p_1(\boldsymbol{x}; \boldsymbol{\theta})}{z} (x_i - \mu_i - z\xi_i) \right) \mathrm{d}z \mathrm{d}\boldsymbol{x}$$

$$= \int_{\mathcal{Q}^{-1}(\boldsymbol{y}_{\backslash i})} \int_{l_i}^{u_i} \int_0^{+\infty} \nabla_{x_i} \left( -\frac{p_1(\boldsymbol{x}; \boldsymbol{\theta})}{z} (x_i - \mu_i - z\xi_i) \right) \mathrm{d}z \mathrm{d}x_i \mathrm{d}\boldsymbol{x}_{\backslash i} \quad (62)$$

$$= \mathsf{E}_z(z^{-1}) \left( (l_i - \mu_i) p(x_i = l_i, \boldsymbol{y}_{\backslash i}; \boldsymbol{\theta}) - \xi_i p_1(x_i = l_i, \boldsymbol{y}_{\backslash i}; \boldsymbol{\theta}) \right.$$

$$\left. - (u_i - \mu_i) p(x_i = u_i, \boldsymbol{y}_{\backslash i}; \boldsymbol{\theta}) + \xi_i p_1(x_i = u_i, \boldsymbol{y}_{\backslash i}; \boldsymbol{\theta}) \right).$$

Since (61) and (62) are equivalent, we have

$$\nabla_{\sigma_i} p(\boldsymbol{y}; \boldsymbol{\theta}) = \frac{1}{2\sigma_i} \left( (l_i - \mu_i) p(x_i = l_i, \boldsymbol{y}_{\backslash i}; \boldsymbol{\theta}) - \xi_i p_1(x_i = l_i, \boldsymbol{y}_{\backslash i}; \boldsymbol{\theta}) \right.$$

$$\left. - (u_i - \mu_i) p(x_i = u_i, \boldsymbol{y}_{\backslash i}; \boldsymbol{\theta}) + \xi_i p_1(x_i = u_i, \boldsymbol{y}_{\backslash i}; \boldsymbol{\theta}) - \mathsf{E}_z[z][\boldsymbol{\Sigma}\boldsymbol{H}_1]_{ii} \right) = \frac{1}{2} \boldsymbol{D}_{1,ii}.$$

Therefore, $\frac{1}{2}(\boldsymbol{H}_1 + \boldsymbol{D}_1) = \nabla_{\boldsymbol{\Sigma}} p(\boldsymbol{y}; \boldsymbol{\theta})$ is valid. $\qquad\square$

Based on Lemma 9, we have that

$$p^{-1}(\boldsymbol{y}; \boldsymbol{\theta}) \boldsymbol{\Sigma} (\boldsymbol{H}_1 + \boldsymbol{D}_1) \boldsymbol{\Sigma} = -2 \nabla_{\boldsymbol{\Sigma}^{-1}} \log p(\boldsymbol{y}; \boldsymbol{\theta}).$$

Hence, the distance (58) becomes

$$\|\mathbf{\Sigma} - \mathbf{\Sigma}^\star\|_{\mathsf{F}} = \left\| \underline{\mathbf{\Sigma}} - \mathbf{\Sigma}^\star - \frac{2}{n} \sum_{t=1}^n \nabla_{\underline{\mathbf{\Sigma}}^{-1}} \log p(\mathbf{y}; \underline{\boldsymbol{\theta}}) + \frac{2}{n} \sum_{t=1}^n \nabla_{\mathbf{\Sigma}^{\star-1}} \log p(\mathbf{y}; \boldsymbol{\theta}^\star) \right\|_{\mathsf{F}}$$

$$= \left\| \underline{\mathbf{\Sigma}} - \mathbf{\Sigma}^\star - \frac{2}{n} \sum_{t=1}^n \int_0^1 \nabla^2_{\tilde{\mathbf{\Sigma}}^{-1}, \tilde{\mathbf{\Sigma}}} \log p(\mathbf{y}_t; \tilde{\boldsymbol{\theta}}_\Sigma)(\tilde{\mathbf{\Sigma}} - \mathbf{\Sigma}^\star) \mathrm{d}\beta \right\|_{\mathsf{F}} \qquad (63)$$

$$\le \sup_{\beta \in (0,1]} \left\| \mathbf{I}_{d \times d} - \frac{2}{n} \sum_{t=1}^n \frac{\partial \mathrm{vec}\left( \nabla_{\tilde{\mathbf{\Sigma}}^{-1}} \log p(\mathbf{y}_t; \tilde{\boldsymbol{\theta}}_\Sigma) \right)}{\partial \mathrm{vec}\left( \tilde{\mathbf{\Sigma}} \right)} \right\|_2 \|\underline{\mathbf{\Sigma}} - \mathbf{\Sigma}^\star\|_{\mathsf{F}}.$$

where the second equation is given by the mean value theorem of integrals and $\tilde{\boldsymbol{\theta}}_\Sigma = \left\{ \underline{\boldsymbol{\mu}}, \tilde{\mathbf{\Sigma}}, \underline{\boldsymbol{\xi}} \right\}$ with $\tilde{\mathbf{\Sigma}} = \beta \underline{\mathbf{\Sigma}} + (1 - \beta) \mathbf{\Sigma}^\star$ and $\beta \in (0, 1]$.

To bound the term $\sup_{\beta \in (0,1]} \left\| \mathbf{I}_{d \times d} - \frac{2}{n} \sum_{t=1}^n \frac{\partial \mathrm{vec}\left( \nabla_{\tilde{\mathbf{\Sigma}}^{-1}} \log p(\mathbf{y}_t; \tilde{\boldsymbol{\theta}}_\Sigma) \right)}{\partial \mathrm{vec}(\tilde{\mathbf{\Sigma}})} \right\|_2$, we need some results for the term $\frac{\partial \mathrm{vec}\left( \nabla_{\mathbf{\Sigma}^{-1}} \log p(\mathbf{y}_t; \boldsymbol{\theta}) \right)}{\partial \mathrm{vec}(\mathbf{\Sigma})}$ at first.

**Lemma 11.** *The term $\frac{\partial \mathrm{vec}\left( \nabla_{\mathbf{\Sigma}^{-1}} \log p(\mathbf{y}; \boldsymbol{\theta}) \right)}{\partial \mathrm{vec}(\mathbf{\Sigma})}$ satisfy*

  *1. it is a positive definite matrix;*

  *2. $\mathbf{I}_{d \times d} - \frac{2}{n} \sum_{t=1}^n \frac{\partial \mathrm{vec}\left( \nabla_{\mathbf{\Sigma}^{-1}} \log p(\mathbf{y}_t; \boldsymbol{\theta}) \right)}{\partial \mathrm{vec}(\mathbf{\Sigma})}$ is also a positive definite matrix.*

*Proof.* Since $\log p(\mathbf{x} \mid z; \boldsymbol{\theta})$ is linear with respect to $\mathbf{\Sigma}^{-1}$, the second order derivative of $\log p(\mathbf{y}; \boldsymbol{\theta})$ with respect to $\mathbf{\Sigma}^{-1}$ is easier to be obtained. We first compute $\frac{\partial \mathrm{vec}\left( \nabla_{\mathbf{\Sigma}^{-1}} \log p(\mathbf{y}_t; \boldsymbol{\theta}) \right)}{\partial \mathrm{vec}(\mathbf{\Sigma}^{-1})}$ and transform it later.

The derivative term $\frac{\partial \mathrm{vec}\left( \nabla_{\mathbf{\Sigma}^{-1}} \log p(\mathbf{y}_t; \boldsymbol{\theta}) \right)}{\partial \mathrm{vec}(\mathbf{\Sigma}^{-1})}$ can be computed as

$$\frac{\partial \mathrm{vec}\left( \nabla_{\mathbf{\Sigma}^{-1}} \log p(\mathbf{y}; \boldsymbol{\theta}) \right)}{\partial \mathrm{vec}\left( \mathbf{\Sigma}^{-1} \right)} = \frac{\frac{\partial \mathrm{vec}\left( \nabla_{\mathbf{\Sigma}^{-1}} p(\mathbf{y}_t; \boldsymbol{\theta}) \right)}{\partial \mathrm{vec}(\mathbf{\Sigma}^{-1})} \cdot p(\mathbf{y}; \boldsymbol{\theta}) - \nabla_{\mathbf{\Sigma}^{-1}} p(\mathbf{y}; \boldsymbol{\theta}) \otimes \nabla_{\mathbf{\Sigma}^{-1}} p(\mathbf{y}; \boldsymbol{\theta})}{p^2(\mathbf{y}; \boldsymbol{\theta})}. \quad (64)$$

Since the first and second order derivatives of $p(\mathbf{y}; \boldsymbol{\theta})$ with respect to $\mathbf{\Sigma}^{-1}$ satisfy

$$\nabla_{\mathbf{\Sigma}^{-1}} p(\mathbf{y}; \boldsymbol{\theta}) = \int_{\mathcal{Q}^{-1}(\mathbf{y})} \int_0^{+\infty} \nabla_{\mathbf{\Sigma}^{-1}} p(\mathbf{x}, z; \boldsymbol{\theta}) \mathrm{d}z \mathrm{d}\mathbf{x},$$

and

$$\frac{\partial \mathrm{vec}\left( \nabla_{\mathbf{\Sigma}^{-1}} p(\mathbf{y}; \boldsymbol{\theta}) \right)}{\partial \mathrm{vec}\left( \mathbf{\Sigma}^{-1} \right)} = \int_{\mathcal{Q}^{-1}(\mathbf{y})} \int_0^{+\infty} \frac{\partial \mathrm{vec}\left( \nabla_{\mathbf{\Sigma}^{-1}} p(\mathbf{x}, z; \boldsymbol{\theta}) \right)}{\partial \mathrm{vec}\left( \mathbf{\Sigma}^{-1} \right)} \mathrm{d}z \mathrm{d}\mathbf{x},$$

based on Lemma 5, the expression (64) becomes

$$\frac{\frac{\partial \mathrm{vec}\left( \nabla_{\mathbf{\Sigma}^{-1}} p(\mathbf{y}; \boldsymbol{\theta}) \right)}{\partial \mathrm{vec}(\mathbf{\Sigma}^{-1})} \cdot p(\mathbf{y}; \boldsymbol{\theta}) - \nabla_{\mathbf{\Sigma}^{-1}} p(\mathbf{y}; \boldsymbol{\theta}) \otimes \nabla_{\mathbf{\Sigma}^{-1}} p(\mathbf{y}; \boldsymbol{\theta})}{p^2(\mathbf{y}; \boldsymbol{\theta})}$$

$$= \frac{\int_{\mathcal{Q}^{-1}(\mathbf{y})} \int_0^{+\infty} \frac{\partial \mathrm{vec}\left( \nabla_{\mathbf{\Sigma}^{-1}} p(\mathbf{x}, z; \boldsymbol{\theta}) \right)}{\partial \mathrm{vec}(\mathbf{\Sigma}^{-1})} \mathrm{d}z \mathrm{d}\mathbf{x}}{p(\mathbf{y}; \boldsymbol{\theta})}$$

$$- \frac{\int_{\mathcal{Q}^{-1}(\mathbf{y})} \int_0^{+\infty} \nabla_{\mathbf{\Sigma}^{-1}} p(\mathbf{x}, z; \boldsymbol{\theta}) \mathrm{d}z \mathrm{d}\mathbf{x}}{p(\mathbf{y}; \boldsymbol{\theta})} \otimes \frac{\int_{\mathcal{Q}^{-1}(\mathbf{y})} \int_0^{+\infty} \nabla_{\mathbf{\Sigma}^{-1}} p(\mathbf{x}, z; \boldsymbol{\theta}) \mathrm{d}z \mathrm{d}\mathbf{x}}{p(\mathbf{y}; \boldsymbol{\theta})}$$

$$= \mathsf{E}_{\mathbf{x}, z | \mathbf{y}; \boldsymbol{\theta}} \left[ \frac{\frac{\partial \mathrm{vec}\left( \nabla_{\mathbf{\Sigma}^{-1}} p(\mathbf{x}, z; \boldsymbol{\theta}) \right)}{\partial \mathrm{vec}(\mathbf{\Sigma}^{-1})}}{p(\mathbf{x}, z; \boldsymbol{\theta})} \right] - \mathsf{E}_{\mathbf{x}, z | \mathbf{y}; \boldsymbol{\theta}} \left[ \frac{\nabla_{\mathbf{\Sigma}^{-1}} p(\mathbf{x}, z; \boldsymbol{\theta})}{p(\mathbf{x}, z; \boldsymbol{\theta})} \right] \otimes \mathsf{E}_{\mathbf{x}, z | \mathbf{y}; \boldsymbol{\theta}} \left[ \frac{\nabla_{\mathbf{\Sigma}^{-1}} p(\mathbf{x}, z; \boldsymbol{\theta})}{p(\mathbf{x}, z; \boldsymbol{\theta})} \right].$$

$$(65)$$

Since we have

$$\nabla_{\boldsymbol{\Sigma}^{-1}} p(\boldsymbol{x}, z; \boldsymbol{\theta}) = \frac{p(\boldsymbol{x}, z; \boldsymbol{\theta})}{2z} \left( \boldsymbol{\Sigma} - (\boldsymbol{x} - \boldsymbol{\mu})(\boldsymbol{x} - \boldsymbol{\mu})^{\top} \right),$$

and

$$\frac{\partial \mathrm{vec}\left(\nabla_{\boldsymbol{\Sigma}^{-1}} p(\boldsymbol{x}, z; \boldsymbol{\theta})\right)}{\partial \mathrm{vec}\left(\boldsymbol{\Sigma}^{-1}\right)} = \frac{p(\boldsymbol{x}, z; \boldsymbol{\theta})}{4z^2} \left( \boldsymbol{\Sigma} - (\boldsymbol{x} - \boldsymbol{\mu})(\boldsymbol{x} - \boldsymbol{\mu})^{\top} \right) \otimes \left( \boldsymbol{\Sigma} - (\boldsymbol{x} - \boldsymbol{\mu})(\boldsymbol{x} - \boldsymbol{\mu})^{\top} \right)$$
$$- \frac{p(\boldsymbol{x}, z; \boldsymbol{\theta})}{2z} \boldsymbol{\Sigma} \otimes \boldsymbol{\Sigma},$$

the expression (65) can further be derived as

$$\mathsf{E}_{\boldsymbol{x}, z | \boldsymbol{y}; \boldsymbol{\theta}} \left[ \frac{\frac{\partial \mathrm{vec}\left(\nabla_{\boldsymbol{\Sigma}^{-1}} p(\boldsymbol{x}, z; \boldsymbol{\theta})\right)}{\partial \mathrm{vec}\left(\boldsymbol{\Sigma}^{-1}\right)}}{p(\boldsymbol{x}, z; \boldsymbol{\theta})} \right] - \mathsf{E}_{\boldsymbol{x}, z | \boldsymbol{y}; \boldsymbol{\theta}} \left[ \frac{\nabla_{\boldsymbol{\Sigma}^{-1}} p(\boldsymbol{x}, z; \boldsymbol{\theta})}{p(\boldsymbol{x}, z; \boldsymbol{\theta})} \right] \otimes \mathsf{E}_{\boldsymbol{x}, z | \boldsymbol{y}; \boldsymbol{\theta}} \left[ \frac{\nabla_{\boldsymbol{\Sigma}^{-1}} p(\boldsymbol{x}, z; \boldsymbol{\theta})}{p(\boldsymbol{x}, z; \boldsymbol{\theta})} \right]$$

$$= \frac{1}{4} \mathsf{E}_{\boldsymbol{x}, z | \boldsymbol{y}; \boldsymbol{\theta}} \left[ \frac{1}{z^2} \mathrm{vec}\left((\boldsymbol{x} - \boldsymbol{\mu})(\boldsymbol{x} - \boldsymbol{\mu})^{\top}\right) \mathrm{vec}\left((\boldsymbol{x} - \boldsymbol{\mu})(\boldsymbol{x} - \boldsymbol{\mu})^{\top}\right)^{\top} - \frac{1}{2z} \boldsymbol{\Sigma} \otimes \boldsymbol{\Sigma} \right]$$

$$- \frac{1}{4} \mathsf{E}_{\boldsymbol{x}, z | \boldsymbol{y}; \boldsymbol{\theta}} \left[ z^{-1} \mathrm{vec}\left((\boldsymbol{x} - \boldsymbol{\mu})(\boldsymbol{x} - \boldsymbol{\mu})^{\top}\right) \right] \mathsf{E}_{\boldsymbol{x}, z | \boldsymbol{y}; \boldsymbol{\theta}} \left[ z^{-1} \mathrm{vec}\left((\boldsymbol{x} - \boldsymbol{\mu})(\boldsymbol{x} - \boldsymbol{\mu})^{\top}\right)^{\top} \right]$$

$$= \frac{1}{4} \mathrm{Cov}_{\boldsymbol{x}, z | \boldsymbol{y}; \boldsymbol{\theta}} \left[ z^{-1} \mathrm{vec}\left(\boldsymbol{x} \boldsymbol{x}^{\top}\right) \right] - \frac{1}{2} \mathsf{E}_{z | \boldsymbol{y}; \boldsymbol{\theta}} \left[ z^{-1} \boldsymbol{\Sigma} \otimes \boldsymbol{\Sigma} \right].$$

(66)

Setting $\tilde{\boldsymbol{x}} = \boldsymbol{x}, z \mid \boldsymbol{y}$ and $f(\mathrm{vec}\left(\boldsymbol{x}\boldsymbol{x}^{\top}\right)) = z^{-1}\mathrm{vec}\left(\boldsymbol{x}\boldsymbol{x}^{\top}\right)$, since $\nabla^2_{\mathrm{vec}(\boldsymbol{x}\boldsymbol{x}^{\top})} \log p(\boldsymbol{x}, z \mid \boldsymbol{y}; \boldsymbol{\theta}) = -\frac{\boldsymbol{\Sigma}^{-1} \otimes \boldsymbol{\Sigma}^{-1}}{z}$, $p(\boldsymbol{x}, z \mid \boldsymbol{y}; \boldsymbol{\theta})$ is a log-concave function with respect to $\mathrm{vec}\left(\boldsymbol{x}\boldsymbol{x}^{\top}\right)$. Based on the Brascamp–Lieb inequality in Lemma 6, we have

$$\mathrm{Cov}_{\boldsymbol{x}, z | \boldsymbol{y}; \boldsymbol{\theta}} \left( z^{-1} \mathrm{vec}\left(\boldsymbol{x}\boldsymbol{x}^{\top}\right) \right) \prec 2 \mathsf{E}_{z | \boldsymbol{y}; \boldsymbol{\theta}} \left[ z^{-1} \right] \boldsymbol{\Sigma} \otimes \boldsymbol{\Sigma}.$$

Since $\boldsymbol{\Sigma} \otimes \boldsymbol{\Sigma}$ is positive definite, $\frac{\partial \mathrm{vec}\left(\nabla_{\boldsymbol{\Sigma}^{-1}} p(\boldsymbol{y}; \boldsymbol{\theta})\right)}{\partial \mathrm{vec}(\boldsymbol{\Sigma}^{-1})}$ is negative definite.

Based on

$$\frac{\partial \mathrm{vec}\left(\nabla_{\boldsymbol{\Sigma}^{-1}} p(\boldsymbol{x}, z; \boldsymbol{\theta})\right)}{\partial \mathrm{vec}\left(\boldsymbol{\Sigma}\right)} = -\frac{\partial \mathrm{vec}\left(\nabla_{\boldsymbol{\Sigma}^{-1}} p(\boldsymbol{x}, z; \boldsymbol{\theta})\right)}{\partial \mathrm{vec}\left(\boldsymbol{\Sigma}^{-1}\right)} \left( \boldsymbol{\Sigma}^{-1} \otimes \boldsymbol{\Sigma}^{-1} \right),$$

we can obtain that $\frac{\partial \mathrm{vec}\left(\nabla_{\boldsymbol{\Sigma}^{-1}} p(\boldsymbol{x}, z; \boldsymbol{\theta})\right)}{\partial \mathrm{vec}(\boldsymbol{\Sigma})}$ is positive definite and

$$\boldsymbol{I}_{d \times d} - \frac{2}{n} \sum_{t=1}^{n} \frac{\partial \mathrm{vec}\left(\nabla_{\boldsymbol{\Sigma}^{-1}} p(\boldsymbol{y}_t; \boldsymbol{\theta})\right)}{\partial \mathrm{vec}\left(\boldsymbol{\Sigma}\right)}$$

$$= \frac{1}{2n} \sum_{t=1}^{n} \left( \boldsymbol{\Sigma}^{-1} \otimes \boldsymbol{\Sigma}^{-1} \right) \mathrm{Cov}_{\boldsymbol{x}, z | \boldsymbol{y}_t; \boldsymbol{\theta}} \left( \mathrm{vec}\left(\boldsymbol{x}\boldsymbol{x}^{\top}\right) \right) \succ \boldsymbol{0}.$$

$\square$

Based on Lemma 11, we have

$$\sup_{\beta \in (0,1]} \left\| \boldsymbol{I}_{d \times d} - \sum_{t=1}^{n} \frac{\partial \mathrm{vec}\left(\nabla_{\tilde{\boldsymbol{\Sigma}}^{-1}} \log p(\boldsymbol{y}_t; \tilde{\boldsymbol{\theta}}_{\Sigma})\right)}{\partial \mathrm{vec}\left(\tilde{\boldsymbol{\Sigma}}\right)} \right\|_2$$

$$= \sup_{\beta \in (0,1]} \left\| \frac{1}{2n} \sum_{t=1}^{n} \left( \tilde{\boldsymbol{\Sigma}}^{-1} \otimes \tilde{\boldsymbol{\Sigma}}^{-1} \right) \mathrm{Cov}_{\boldsymbol{x}, z | \boldsymbol{y}_t; \tilde{\boldsymbol{\theta}}_{\Sigma}} \left[ \mathrm{vec}\left(\boldsymbol{x}\boldsymbol{x}^{\top}\right) \right] \right\|_2$$

$$= \max_{\boldsymbol{\theta}} \left\| \frac{1}{2n} \sum_{t=1}^{n} \left( \boldsymbol{\Sigma}^{-1} \otimes \boldsymbol{\Sigma}^{-1} \right) \mathrm{Cov}_{\boldsymbol{x}, z | \boldsymbol{y}_t; \boldsymbol{\theta}} \left( \mathrm{vec}\left(\boldsymbol{x}\boldsymbol{x}^{\top}\right) \right) \right\|_2$$

$$\triangleq c_{\Sigma} \in (0, 1).$$

## C   PROOF OF THEOREM 2

Based on Proposition 1, we have proved the linear convergence rates of the three parameters in $\boldsymbol{\theta}$. To connect the global convergence rates and the separable convergence rates, we introduce a result.

**Lemma 12** (Meng & Rubin (1994)). *The global convergence rate of an ECM algorithm is the maximum of the componentwise rates of convergence.*

Based on Lemma 12, we obtain that

$$\|\boldsymbol{\theta}^{(k)} - \boldsymbol{\theta}^\star\|_2 \le c^k \|\boldsymbol{\theta}^{(0)} - \boldsymbol{\theta}^\star\|_2,$$

where $c = \max\{c_\mu, c_\xi, c_\Sigma\} \in (0, 1)$.

## D   DERIVATIONS OF SURROGATE FUNCTIONS IN QUANTIZED MATRIX COMPLETION AND COMPRESSIVE SENSING

### D.1   SURROGATE FUNCTION DERIVATION IN QUANTIZED MATRIX COMPLETION

The goal of the low-rank matrix completion problem is to recover an unknown low-rank matrix $\boldsymbol{M} \in \mathbb{R}^{d_1 \times d_2}$ from an observed, yet incomplete, matrix. Let $\boldsymbol{X}$ denote a matrix whose entries are drawn from a normal mean–variance mixture distribution. We use $\mu_{ij}$ and $x_{ij}$ to represent the $(i, j)$-th entries of $\boldsymbol{M}$ and $\boldsymbol{X}$, respectively. Based on the normal mean-variance model, the relationship between $\mu_{ij}$ and $x_{ij}$ is given by

$$x_{ij} = \mu_{ij} + z\xi + z^{1/2}\sigma\epsilon,$$

where $\mu_{ij}$ serves as the location parameter of $x_{ij}$ and both $\xi$ and $\sigma$ are given constants. In the quantization scenario, we have $\boldsymbol{Y} = \mathcal{Q}(\boldsymbol{X})$. The entire matrix $\boldsymbol{Y}$ is not available for observation. Let $\boldsymbol{\Omega}$ denote the index set of the observed entries. Our goal is to recover $\boldsymbol{M}$ from incomplete $\boldsymbol{Y}$, which is equivalent to estimating all the $\mu_{ij}$ in the matrix $\boldsymbol{M}$.

Since $x_{ij}$ follows a univariate normal mean-variance model and $\boldsymbol{Y} = \mathcal{Q}(\boldsymbol{X})$, the density function of $y_{ij}$ is identical with (5) under the univariate case. Hence, the negative log-likelihood function for $y_{ij}$ with $ij \in \boldsymbol{\Omega}$ is

$$\sum_{(i,j) \in \boldsymbol{\Omega}} \log p(y_{ij} \mid \boldsymbol{M}),$$

which is the objective function of (16). Applying Jensen's inequality as in (9), we have the surrogate function

$$S(\boldsymbol{M}; \underline{\boldsymbol{M}}) = \sum_{(i,j) \in \boldsymbol{\Omega}} \left[ \mathsf{E}_{z_t | \boldsymbol{y}_t; \boldsymbol{\theta}} [\log p(z_t)] - \frac{1}{2} \log \det \sigma^2 \right.$$
$$\left. - \frac{1}{2\sigma^2} \left( (u_{ij} - 2v_{ij}\mu + \iota_{ij}\mu^2) - 2(w_{ij} - \mu)\xi + \zeta_{ij}\xi^2 \right) \right] + \text{const.},$$

where $u_{ij}$, $v_{ij}$, and $w_{ij}$ are the univariate versions of $\boldsymbol{U}_{ij}$, $\boldsymbol{v}_{ij}$, and $\boldsymbol{w}_{ij}$, respectively. Ignoring the terms which are independent with $\mu_{ij}$, the surrogate function becomes

$$S(\boldsymbol{M}; \underline{\boldsymbol{M}}) = \frac{1}{2\sigma^2} \sum_{(i,j) \in \boldsymbol{\Omega}} \left( 2v_t\mu - \iota_t\mu^2 - 2\mu\xi \right) + \text{const.}$$

Making a low-rank factorization to the matrix $\boldsymbol{M}$ as $\boldsymbol{M} = \boldsymbol{A}\boldsymbol{B}^\top$, we have $\mu_{ij} = \boldsymbol{a}_i \boldsymbol{b}_j^\top$. Setting $e_{ij} = \frac{v_{ij} - \xi}{\iota_{ij}}$, we have that maximizing $S(\boldsymbol{M}; \underline{\boldsymbol{M}})$ is equivalent to maximize

$$\sum_{(i,j) \in \boldsymbol{\Omega}} \left( \boldsymbol{a}_i \boldsymbol{b}_j^\top - e_{ij} \right)^2,$$

which is identical with (17).

### D.2 SURROGATE FUNCTION DERIVATION IN QUANTIZED COMPRESSIVE SENSING

In quantized compressive sensing, the base model with normal mean-variance mixture noise is given by

$$\boldsymbol{y} = \mathcal{Q}(\boldsymbol{x}), \quad \boldsymbol{x} = \boldsymbol{\Phi}\boldsymbol{\vartheta} + z\boldsymbol{\xi} + z^{1/2}\sigma\boldsymbol{\epsilon}.$$

In the above model, the term $\boldsymbol{\Phi}\boldsymbol{\vartheta}$ can be regarded as the location parameter of $\boldsymbol{x}$. We can use the ML estimation method to recover the sparse signal $\boldsymbol{\vartheta}$, which has the optimization problem

$$\max_{\boldsymbol{\vartheta}} \quad \log p\left(\boldsymbol{y} \mid \boldsymbol{\vartheta}\right).$$

Following the suggestions from Zymnis et al. (2009), we add a $\ell_1$-regularization term to force the solution of $\boldsymbol{\vartheta}$ to be sparse. Hence, we obtain the optimization problem

$$\max_{\boldsymbol{\vartheta}} \quad \log p\left(\boldsymbol{y} \mid \boldsymbol{\vartheta}\right) + \eta\|\boldsymbol{\vartheta}\|_1.$$

To solve the above optimization problem through ECM algorithm, we can apply the E-step as in (9) to obtain the surrogate function

$$S(\boldsymbol{\vartheta}; \underline{\boldsymbol{\vartheta}}) = \mathsf{E}_{z_t|\boldsymbol{y}_t;\underline{\boldsymbol{\theta}}}\left[\log p(z_t)\right] - \frac{1}{2}\log\det\sigma^2$$
$$- \frac{1}{2\sigma^2}\left(\left(\boldsymbol{u} - 2\boldsymbol{v}^\top\boldsymbol{A}\boldsymbol{\vartheta} + \iota\boldsymbol{\vartheta}^\top\boldsymbol{A}^\top\boldsymbol{A}\boldsymbol{\vartheta}\right) - 2(\boldsymbol{w} - \boldsymbol{A}\boldsymbol{\vartheta})^\top\boldsymbol{\xi} + \zeta\boldsymbol{\xi}^\top\boldsymbol{\xi}\right) + \eta\|\boldsymbol{\vartheta}\|_1 + \mathrm{const.}$$

Ignoring the terms which are independent with $\boldsymbol{\vartheta}$, the surrogate function becomes

$$S(\boldsymbol{\vartheta}; \underline{\boldsymbol{\vartheta}}) = \frac{1}{2\sigma^2}\left(2\boldsymbol{v}^\top\boldsymbol{A}\boldsymbol{\vartheta} - \iota\boldsymbol{\vartheta}^\top\boldsymbol{A}^\top\boldsymbol{A}\boldsymbol{\vartheta} - 2\boldsymbol{\vartheta}^\top\boldsymbol{A}^\top\boldsymbol{\xi}\right) + \eta\|\boldsymbol{\vartheta}\|_1 + \mathrm{const.}$$

Setting $\boldsymbol{e} = \frac{\boldsymbol{v} - \boldsymbol{\xi}}{\iota}$, we have that maximizing $S(\boldsymbol{\vartheta}; \underline{\boldsymbol{\vartheta}})$ is equivalent to maximize

$$\|\boldsymbol{A}\boldsymbol{\vartheta} - \boldsymbol{e}\|_2^2 + \eta\|\boldsymbol{\vartheta}\|_1,$$

which is identical with the surrogate function in quantized compressive sensing.

## E EXPERIMENT DETAILS

### E.1 BENCHMARK SETTINGS

**Quantized matrix completion:** For all algorithms presented in Table 2, the complete matrix $\boldsymbol{M}$ is reconstructed from the training data. Denote by $\boldsymbol{\Omega}_{\text{test}}$ the set of indices corresponding to entries in the test data. The definitions of the two benchmarks are given as follows.

1. Accuracy: $\sum_{(i,j)\in\boldsymbol{\Omega}_{\text{test}}} I(\mathcal{Q}(\mu_{ij}) = y_{ij})$;

2. RMSE: $\sqrt{\frac{1}{n_{\text{test}}}\sum_{(i,j)\in\boldsymbol{\Omega}_{\text{test}}}(\mu_{ij} - y_{ij})^2}$,

where $I(\cdot)$ is the indicator function.

**Quantized compressive sensing:** The signal-to-noise ratio (SNR) is defined as the ratio of the expectation of the original measurements to the variance of the noise term, which is

$$\mathrm{SNR} = \frac{\mathrm{E}[\boldsymbol{A}\boldsymbol{\vartheta}]}{\mathrm{Var}[\boldsymbol{x} - \boldsymbol{A}\boldsymbol{\vartheta}]}.$$

Denote the ground truth sparse signal as $\boldsymbol{\vartheta}_{\text{gd}}$, the estimated sparse signal as $\boldsymbol{\vartheta}_{\text{est}}$. The cosine similarity is defined as

$$\mathrm{Cos\ Sim} = \frac{\boldsymbol{\vartheta}_{\text{gd}}^\top\boldsymbol{\vartheta}_{\text{est}}}{\|\boldsymbol{\vartheta}_{\text{gd}}\|_2\|\boldsymbol{\vartheta}_{\text{est}}\|_2}.$$

## E.2   EXPERIMENTAL SETTINGS

In Figure 4, we set $\mu_i \sim \mathrm{Uniform}(0,1)$, $\xi_i \sim \mathrm{Uniform}(0,2)$, and $\mathbf{v}_i \sim \mathcal{N}(\mathbf{0}, \boldsymbol{I})$ for $i = 1, \ldots, d$, and construct the covariance matrix as $\boldsymbol{\Sigma} = \sum_{i=1}^{d} \mathbf{v}_i \mathbf{v}_i^{\top}$. For the GH distribution, we use $\lambda = -0.5$, $\delta = 2$, and $\gamma = 2$; for GHST, we set $\nu = 5$, $\lambda = -\mu/2$, $\delta = \nu$, and $\gamma = 0$. The number of samples is $n = 1 \times 10^4$ and the threshold is fixed at $\tau = 0.3$.

The specific parameter settings for the algorithms in Table 2 are given by

1. Standard Gaussian: $r = 2$ and $\sigma^2 = 0.8$;
2. one-bit Gaussian: $r = 2$ and $\sigma^2 = 0.8$;
3. multi-bit Gaussian: $r = 2$ and $\sigma^2 = 0.8$;
4. multi-bit Student's $t$: $r = 2$, $\sigma^2 = 0.8$, and $\nu = 6$;
5. multi-bit GHST: $r = 2$, $\sigma^2 = 0.8$, $\xi = -0.1$, and $\nu = 8$;
6. multi-bit GH: $r = 2$, $\sigma^2 = 0.8$, $\xi = -0.1$, $\lambda = -4$, $\delta = 4$, and $\gamma = 0.3$.

