# OpenReview forum: "ML Estimation from Bits"
_ICLR.cc/2026/Conference — ICLR 2026 Conference Withdrawn Submission_

### Official Review · Reviewer_Z3VM · 2025-10-29

**Soundness:** 2
**Presentation:** 2
**Contribution:** 2
**Rating:** 2
**Confidence:** 3

**Summary:**

In this paper, the authors consider maximum likelihood estimation of the parameters $(\mu,\xi,\Sigma)$ of a random vector $x \in \mathbb{R}^d$ following a normal variance-mean mixture model from quantized observations $y_i = Q(x_i)$ where $Q$ is a scalar quantizer and $x_1,…,x_n$ are iid samples of $x$. According to this model

$$x = \mu + z\xi + (z \Sigma)^{1	/2} \epsilon,$$

where $\mu$ is the location parameter, $\xi$ is the skewness parameter, $\Sigma$ is the scattering parameter, $\varepsilon$ is a standard normal random vector, and $z$ is a non-negative random variable with density $p(z)$.

Since evaluating the likelihood function $L$ involves double integration and is not possible analytically, the authors use Jensen’s inequality to construct a surrogate functional that lower bounds $L$ and can be maximized more efficiently by alternating steps of taking expectations and maximizing the current surrogate.

In Theorem 2 they show global convergence of the alternating method to stationary points of L at a linear rate.

**Strengths:**

+ Paper is easy to read and tackles a relevant problem

**Weaknesses:**

- Convergence result is not clearly stated
- Numerical evaluation is hard to interpret
- Efficiency of the method for covariance estimation is not analyzed (neither theoretically nor experimentally)

**Questions:**

Since the paper does not convince me that the ML approach outperforms existing methods for quantized covariance/correlation estimation, I do not recommend it to be accepted yet. In my opinion, the numerical comparison with existing approaches must be clarified to clearly prove gains in computational or sample efficiency. Moreover, the convergence result needs to be revised.

Here a list of questions/issues that should be addressed:

- l. 61: Q is quite loosely defined at the moment. In particular, the statement that Q becomes sign for e=2 is not correct since the present definition would also allow Q taking values -a and b for values in (-\infty,c) and [c,\infty), respectively, where $a,b > 0$ and $c$ can be chosen arbitrarily.

- Some relevant more recent literature has not been mentioned:

[1] Chen, Junren, and Michael K. Ng. "A parameter-free two-bit covariance estimator with improved operator norm error rate." Applied and Computational Harmonic Analysis (2025): 101774.

[2] Dirksen, Sjoerd, and Johannes Maly. "Tuning-free one-bit covariance estimation using data-driven dithering." IEEE Transactions on Information Theory 70.7 (2024): 5228-5247

- l. 109: The works of Chen et al. and of Dirksen et al. are not restricted to Gaussian distributions

- ll. 169+177: irrelevant -> independent ?

- l. 188: independent with -> independent of

- Proposition 1: For which stationary point does the result hold? It cannot hold for all stationary points simultaneously. From the proof, I cannot see that a particular stationary point is picked, so I doubt the correctness of the argument. I did not have time to check the proof in detail though.

- Section 5 is suddenly discussing matrix completion and compressed sensing as applications of the work without specifying how the results can be applied. For matrix completion this is then done in Section 6, for compressed sensing it’s completely missing. I find this structure strongly confusing.

- Section 6: I find the labelling of the plots highly unclear and I cannot really connect the lines to the methods discussed in the text. To make this comparison rigorous and interpretable, please clearly name the methods you compare and use the same labelling in text and plots. The present presentation does not convince me of the value of the method. Furthermore, efficiency in approximation is something that should definitely be evaluated and compared in the covariance estimation setting since existing approaches are quite cheap to compute.

- Finally, is there any hope to analyze the estimation error of the ML approach in dependence of the number of samples?  Chen et al. and of Dirksen et al. provide rigorous and non-asymptotic error guarantees for their respective estimators, which are essential for reliability of the method.

---

> ### Author Response · Authors · 2025-11-22
>
> **Comment 1:** Convergence result is not clearly stated. Proposition 1: For which stationary point does the result hold? It cannot hold for all stationary points simultaneously. From the proof, I cannot see that a particular stationary point is picked, so I doubt the correctness of the argument. I did not have time to check the proof in detail though.
>
> **Response to Comment 1:** Since our algorithm is under the ECM algorithmic framework, it is guaranteed to converge to a stationary point of the original objective function, where the specific stationary point reached depends on the chosen initial point. The above result with its proof is given in [A]. In the following, we show the proof.
>
> We first briefly introduce the ECM algorithm. The core of the ECM algorithm is that this algorithm iteratively optimizes a surrogate function $S$ instead of directly optimizing the objective function $L\left(\boldsymbol{\theta}\right)$. In the E-step, given the current parameters, the algorithm forms a surrogate function $S(\boldsymbol{\theta};\underline{\boldsymbol{\theta}})$. This function is constructed to be a tight lower bound of the log-likelihood function $L\left(\boldsymbol{\theta}\right)$; specifically, it is guaranteed $S(\boldsymbol{\theta};\underline{\boldsymbol{\theta}})\leq L\left(\boldsymbol{\theta}\right)$ everywhere. Besides, at the point of the current parameters, we have $S(\underline{\boldsymbol{\theta}};\underline{\boldsymbol{\theta}})= L\left(\underline{\boldsymbol{\theta}}\right)$ and $S^\prime(\underline{\boldsymbol{\theta}};\underline{\boldsymbol{\theta}})= L^\prime\left(\underline{\boldsymbol{\theta}}\right)$.
>
> After the E-step constructs the surrogate lower bound, the CM-steps sequentially update the parameters that maximize this surrogate function. Because the surrogate was a tight lower bound at the previous parameters and we have increased its value, the value of the true log-likelihood at the new parameters $\boldsymbol{\theta}^{+}$ is guaranteed to be greater than or equal to its previous value, which is
> \[
> L\left(\boldsymbol{\theta}^{+}\right)\geq S(\boldsymbol{\theta}^{+};\underline{\boldsymbol{\theta}})\geq S(\underline{\boldsymbol{\theta}};\underline{\boldsymbol{\theta}})= L\left(\underline{\boldsymbol{\theta}}\right).
> \]
> The updated $\boldsymbol{\theta}^{+}$ is a unique maximizer of $S(\boldsymbol{\theta};\underline{\boldsymbol{\theta}})$ which satisfies $S^\prime(\boldsymbol{\theta}^{+};\underline{\boldsymbol{\theta}})=0$. By alternating between E and CM-steps, the ECM algorithm guarantees a monotonic increase in the original objective function.
>
> Starting from the initial point $\boldsymbol{\theta}^{(0)}$, iteratively applying the E-step and CM-step to update $\boldsymbol{\theta}$ yields a sequence $\left\\{L\left(\boldsymbol{\theta}^{(0)}\right), L\left(\boldsymbol{\theta}^{(1)}\right), \dots\right\\}$. Under the normal mean–variance model assumption, the log-likelihood function $L\left(\boldsymbol{\theta}\right)$ is bounded, and as established in the preceding derivation, this sequence is monotonically increasing. Therefore, the sequence converges to a limit. At the limit point $L\left(\boldsymbol{\theta}^{(k)}\right)$ and $\boldsymbol{\theta}^{(k)}$, we have
> \[
> L\left(\boldsymbol{\theta}^{(k)}\right) = L\left(\boldsymbol{\theta}^{(k+1)}\right),
> \]
> and
> $
> L^\prime\left(\boldsymbol{\theta}^{(k+1)}\right) = S^\prime\left(\boldsymbol{\theta}^{(k+1)} \mid \boldsymbol{\theta}^{(k+1)}\right) = S^\prime\left(\boldsymbol{\theta}^{(k+1)} \mid \boldsymbol{\theta}^{(k)}\right) = 0.
> $
> Hence, the sequence $\left\\{\boldsymbol{\theta}^{(0)}, \boldsymbol{\theta}^{(1)}, \ldots, \boldsymbol{\theta}^{(k)}\right\\}$ converges to a stationary point of $L\left(\boldsymbol{\theta}\right)$. As a result, any algorithm that adheres to the ECM framework is guaranteed to converge to a local optimum (or a saddle point) of the original maximum likelihood problem [A].
>
> We will clarify this dependency in the final manuscript to avoid the impression that our result holds for all stationary points simultaneously.
>
> [A] McLachlan, G. J., and Krishnan, T. (2008). \textit{The EM algorithm and extensions}. John Wiley \& Sons.
>
> **Comment 2:** Numerical evaluation is hard to interpret. Section 6: I find the labelling of the plots highly unclear and I cannot really connect the lines to the methods discussed in the text. To make this comparison rigorous and interpretable, please clearly name the methods you compare and use the same labelling in text and plots.
>
> **Response to Comment 2:**
> To address the issue of insufficiently clear labels in the plots, the labels are updated, and their meanings are explained in the revised paper.

---

> ### Author Response · Authors · 2025-11-22
>
> **Comment 3:** The present presentation does not convince me of the value of the method. Furthermore, efficiency in approximation is something that should definitely be evaluated and compared in the covariance estimation setting since existing approaches are quite cheap to compute. Efficiency of the method for covariance estimation is not analyzed (neither theoretically nor experimentally).
>
> **Response to Comment 3:**
>
> Since the proposed method is based on ML estimation, it estimates the parameters from quantized data under specific distributional assumptions. Within this modeling framework, it is difficult to conduct a theoretical analysis of the efficiency of the parameter estimates, and existing theoretical results for ML estimation in such settings are also lacking. For empirical analysis, curves of the mean squared error (MSE) of the parameter estimates versus the number of samples (Figure 1) and the number of bits (Figure 2) are included in the experimental section. These results show that the method achieves better MSE performance because it directly solves the ML estimation problem, rather than an approximate or surrogate problem. Moreover, in both the quantized matrix completion (Table 1) and quantized compressive sensing (Table 2) scenarios, the proposed method also demonstrates superior performance.
>
>
> **Comment 4:** \textbf{Line 61:} $Q$ is loosely defined at the moment. The statement that $Q$ becomes `sign` for $e=2$ is not correct; the present definition would also allow $Q$ taking values $-a$ and $b$ for inputs in $(-\infty,c)$ and $[c,\infty)$, respectively, where $a, b > 0$ and $c$ is arbitrary.
>
> **Response to Comment 4:** We appreciate the reviewer's careful reading and agree that our original statement was inaccurate. The correct description is: when $e = 2$, $k_1 = -1$, $k_2 = 1$, and $\tau_1 = 0$, the quantization function $\mathcal{Q}$ reduces to the element-wise signum function. We will update the manuscript accordingly to ensure precision in the definition of signum function.

---

> ### Author Response · Authors · 2025-11-22
>
> **Comment 5:** Some relevant recent literature has not been mentioned:
> 1. Chen, Junren, and Michael K. Ng. ``A parameter-free two-bit covariance estimator with improved operator norm error rate.'' *Applied and Computational Harmonic Analysis* (2025): 101774.
> 2. Dirksen, Sjoerd, and Johannes Maly. ``Tuning-free one-bit covariance estimation using data-driven dithering.'' *IEEE Transactions on Information Theory* 70.7 (2024): 5228--5247.
>
> l. 109: The works of Chen et al. and of Dirksen et al. are not restricted to Gaussian distributions
>
> **Response to Comment 5:** Regarding the first paper by [A], we were not aware of this work during the preparation of our manuscript, as it appeared very recently. Nevertheless, we have cited their earlier related work [B]. The quantization function considered in [A], [B] differs substantially from that in our study:
> (1) Chen *et al.* adopt a uniform quantizer, which uses equal-width intervals, whereas our quantization function can employ unequal-width intervals;
> (2) Their quantizer typically operates over a bounded input range with saturation at the boundaries, while our quantization function can be defined over the entire real axis.
> Due to these differences, we believe the models studied are fundamentally distinct, and therefore, a direct comparison was not performed in the current manuscript.
>
> Regarding the second paper by [C], we have cited and considered their earlier related work [D]. The covariance estimation algorithm in [C] is similar to their previous method. We have compared that earlier method with our own in the 1-bit Gaussian setting, where it did not show competitive performance against our approach or other existing methods. Since the algorithm applies the same methodology to both Gaussian and non-Gaussian data, and its Gaussian-case results were not favorable, we did not include comparisons in the non-Gaussian scenario in the original submission.
>
> In the revised manuscript, we will add citations to both [A] and [D], and we will incorporate experiments comparing our non-Gaussian estimator with [D] in the non-Gaussian setting. The updated figures now provide a comparative analysis of the following methods: the constant dithering method, the adaptive dithering method proposed by Dirksen et al. [C] [D], and the two-bit covariance estimator alongside the parameter-free covariance estimator proposed by Chen et al. [A], [B]. Please refer to the numerical results in Figure 1 and Figure 2 for details.
>
> [A] Chen, J., and Ng, M. K. (2025). A parameter-free two-bit covariance estimator with improved operator norm error rate. *Applied and Computational Harmonic Analysis*, 101774.
>
> [B] Chen, J., Ng, M. K., and Wang, D. (2024). Quantizing heavy-tailed data in statistical estimation: (Near) minimax rates, covariate quantization, and uniform recovery. *IEEE Transactions on Information Theory*, 70(3), 2003–2038.
>
> [C] Dirksen, S., and Maly, J. (2024). Tuning-free one-bit covariance estimation using data-driven dithering. *IEEE Transactions on Information Theory*, 70(7), 5228–5247.
>
> [D] Dirksen, S., Maly, J., and Rauhut, H. (2022). Covariance estimation under one-bit quantization. *The Annals of Statistics*, 50(6), 3538–3562.
>
> **Comment 6:** ll. 169+177: irrelevant → independent ?
>
> l. 188: independent with → independent of
>
> **Response to Comment 6:** Thank you for carefully checking the manuscript and pointing out these typographical issues. We have corrected “irrelevant” to “independent” in lines 169 and 177, and replaced “independent with” by “independent of” in line 188 in the revised version.
>
> **Comment 7:** Section 5 is suddenly discussing matrix completion and compressed sensing as applications of the work without specifying how the results can be applied. For matrix completion this is then done in Section 6, for compressed sensing it’s completely missing. I find this structure strongly confusing.
>
> **Response to Comment 7:** In the revised version, we have clarified the descriptions of both models and made it more explicit how each of them relates to our proposed algorithm. In particular, we now add the corresponding discussion for the compressed sensing setting in Section 6, which was omitted in the initial submission due to page limitations. This revision makes the applicability of our results to both matrix completion and compressed sensing more transparent.

---

> ### Author Response · Authors · 2025-11-22
>
> **Comment 8:** Finally, is there any hope to analyze the estimation error of the ML approach in dependence of the number of samples? Chen et al. and of Dirksen et al. provide rigorous and non-asymptotic error guarantees for their respective estimators, which are essential for reliability of the method.
>
> **Response to Comment 8:** We acknowledge the importance of the non-asymptotic error guarantees provided in works such as Chen *et al.* [A], [B] and Dirksen *et al.* [C], [D]. Our current paper focuses primarily on proposing a new algorithm and detailing its optimization principles and computational procedures. Consequently, we have centered our theoretical analysis on convergence properties, without conducting a comprehensive study of the statistical aspects, such as estimation error or its dependence on sample size. We agree that such an analysis would be valuable for establishing stronger reliability guarantees and plan to investigate these directions in our future work.
> [A] Chen, J., and Ng, M. K. (2025). A parameter-free two-bit covariance estimator with improved operator norm error rate. *Applied and Computational Harmonic Analysis*, 101774.
>
> [B] Chen, J., Ng, M. K., and Wang, D. (2024). Quantizing heavy-tailed data in statistical estimation: (Near) minimax rates, covariate quantization, and uniform recovery. *IEEE Transactions on Information Theory*, 70(3), 2003–2038.
>
> [C] Dirksen, S., and Maly, J. (2024). Tuning-free one-bit covariance estimation using data-driven dithering. *IEEE Transactions on Information Theory*, 70(7), 5228–5247.
>
> [D] Dirksen, S., Maly, J., and Rauhut, H. (2022). Covariance estimation under one-bit quantization. *The Annals of Statistics*, 50(6), 3538–3562.

---

> ### Comment · Reviewer_Z3VM · 2025-11-26
>
> I thank the authors for their clarifications. I have some further questions/comments below. To summarize, I am still not convinced that the paper is ripe for publication. Since it only provides an empirical analysis of the algorithm’s estimation performance, the numerical benchmark should be more rigorous and should show clearer benefits in accuracy given the high computational costs of the proposed method (see detailed comments below).
>
>
> Regarding Comment 1:
>
> If I understand it correctly, you use McLachlan et al. (2008) to argue that the sequence $(\mu_k,\xi_k,\Sigma_k)$ converges to some stationary point $(\mu_*,\xi_*,\Sigma_*)$ and then, based on this, prove linear convergence in Proposition 1/Theorem 2. Is this correct?
>
> I looked a bit deeper into the proof now, but find it hard to follow all steps. Consequently, I cannot confirm correctness of the result. Considering Figure 3 in the revised version, I agree that locally a linear rate can be observed. Whether this is also holds globally, as claimed in Proposition 1, seems less clear to me.
>
> Regarding Comment 2:
>
> The plots are now much more readable!
>
> Regarding Comment 3:
>
> To me the current experiments suggest that the proposed ML estimation approach is able to slightly outperform existing methods, but pays heavily in terms of efficiency (see, e.g., Figure 1, and Table 2 Multi-bit Gaussian vs proposed methods). It might make sense to find a setting where the normal mean-variance mixture model allows to outperform existing approaches more clearly. Otherwise it is of limited use in practice. For instance, how does Figure 1 change if one considers heavier-tailed distributions?
>
> Table 3 is not clear to me. What are RSS and AOP? To understand how good your algorithm performs in 1-bit compressive sensing, one would have to compare it with state-of-the-art schemes such as [1] (Gaussian observations, no dither, no noise), [2] (Gaussian observations, no dither, bit-flips), [3] (heavy-tailed observations, random dither, noise+bit-flips), and [4] (Subgaussian observations, random dither, noise+bit-flips). Here it would also be interesting to understand how robust the ML estimator is in view of adversarial bit-flips (see [2-4]).
>
> [1] Y. Plan and R. Vershynin, “One-bit compressed sensing by linear programming,” Communications on Pure and Applied Mathematics, vol. 66, no. 8, pp. 1275–1297, 2013.
> [2] Y. Plan and R. Vershynin, “Robust 1-bit compressed sensing and sparse logistic regression: a convex programming approach,” IEEE Transactions on Information Theory, vol. 59, no. 1, pp. 482–494, 2013.
> [3] Sjoerd Dirksen and Shahar Mendelson. Non-Gaussian hyperplane tessellations and robust one-bit com- pressed sensing. Journal of the European Mathematical Society, 23(9):2913–2947, 2021.
> [4] Hans Christian Jung, Johannes Maly, Lars Palzer, and Alexander Stollenwerk. Quantized compressed
> sensing by rectified linear units. IEEE Transactions on Information Theory, 67(6):4125–4149, 2021.
>
> Regarding Comments 4, 6, 7, and 8:
>
> Thanks for the clarifications/corrections!
>
> Regarding Comment 5:
>
> I think it’s important to include these approaches in the numerical comparison, as now done in Figure 1. The statement „We have compared that earlier method [referring to the adaptive dithering approaches] with our own in the 1-bit Gaussian setting, where it did not show competitive performance against our approach or other existing methods.“ seems debatable to me. In fact, Figure 1 shows that adaptive dithering and parameter-free estimator perform comparable with the other schemes for small sample sizes and less than one order of magnitude worse for more samples, while being more efficient than the proposed method by almost six orders of magnitude.

---

> > ### Author Response · Authors · 2025-12-01
> > **Reply to follow-up comment. (2/3)**
> >
> > **Comment 3:** Table 3 is not clear to me. What are RSS and AOP? To understand how good your algorithm performs in 1-bit compressive sensing, one would have to compare it with state-of-the-art schemes such as [1] (Gaussian observations, no dither, no noise), [2] (Gaussian observations, no dither, bit-flips), [3] (heavy-tailed observations, random dither, noise+bit-flips), and [4] (Subgaussian observations, random dither, noise+bit-flips). Here it would also be interesting to understand how robust the ML estimator is in view of adversarial bit-flips (see [2-4]).
> >
> > [1] Y. Plan and R. Vershynin, “One-bit compressed sensing by linear programming,” Communications on Pure and Applied Mathematics, vol. 66, no. 8, pp. 1275–1297, 2013. [2] Y. Plan and R. Vershynin, “Robust 1-bit compressed sensing and sparse logistic regression: a convex programming approach,” IEEE Transactions on Information Theory, vol. 59, no. 1, pp. 482–494, 2013. [3] Sjoerd Dirksen and Shahar Mendelson. Non-Gaussian hyperplane tessellations and robust one-bit compressed sensing. Journal of the European Mathematical Society, 23(9):2913–2947, 2021. [4] Hans Christian Jung, Johannes Maly, Lars Palzer, and Alexander Stollenwerk. Quantized compressed sensing by rectified linear units. IEEE Transactions on Information Theory, 67(6):4125–4149, 2021.
> >
> > **Response to Comment 3:** We thank the reviewer for bringing these four articles to our attention. We supplement the experiments regarding quantized compressive sensing within Table 3. The existing methods included in our comparison consist of restricted step shrinkage (RSS) [A], adaptive outlier pursuit (AOP) [B], and linear programming (LP) [C], all of which are based on a noise-free assumption. Additionally, we compare methods that utilize logistic regression (LR) [D, E], Rectified Linear Units (ReLU) [F], and 1-bit Gaussian MLE [G] as loss functions to achieve robustness in modeling heavy-tailed raw data. To our understanding, paper [F] does not propose a new loss function; rather, its primary contribution lies in broadening the distributional assumptions of the measurement matrix from Gaussian to sub-Gaussian and heavy-tailed distributions. This focus falls outside the scope of our experimental comparison, and consequently, no comparison with this work is included.
> >
> > To demonstrate the robustness of the ML estimator against adversarial bit-flips, the number of sign flips occurring between the generated noise-free data and the noisy data is calculated, and the average bit-flip rate is subsequently recorded. The result is shown in the following table.
> >
> > | Method                     | Cos Sim (SNR = 0 dB, bit-flips rate = 22.3\%) | Time (s) | Cos Sim (SNR = -5 dB, bit-flips rate = 30.5\%) | Time (s) | Cos Sim (SNR = -10 dB, bit-flips rate = 37.3\%) | Time (s) |
> > |----------------------------|-----------------------------------------------|----------|-------------------------------------------------|----------|--------------------------------------------------|----------|
> > | RSS                        | 0.7872                                        | 0.2639   | 0.6598                                          | 0.2651   | 0.3027                                           | 0.2546   |
> > | AOP                        | 0.7052                                        | 0.2689   | 0.4041                                          | 0.2951   | 0.2209                                           | 0.2738   |
> > | LP                         | 0.5911                                        | 111.5    | 0.3785                                          | 168.6    | 0.1962                                           | 140.9    |
> > | Logi                       | 0.7515                                        | 0.0083   | 0.4648                                          | 0.0415   | 0.3312                                           | 0.0376   |
> > | ReLU                       | 0.6183                                        | 6.9214   | 0.3699                                          | 7.4717   | 0.2550                                           | 7.5803   |
> > | 1-bit Gaussian             | 0.8220                                        | 2.7704   | 0.5487                                          | 3.0875   | 0.2896                                           | 2.7202   |
> > | 1-bit Student's $t$ ECM (prop.) | 0.8396                                   | 2.8573   | 0.5678                                          | 3.1643   | 0.3031                                           | 3.3174   |
> > | 1-bit GHST ECM (prop.)     | 0.8893                                        | 2.7435   | 0.7701                                          | 3.4204   | 0.5237                                           | 4.0281   |
> > | 1-bit GH ECM (prop.)       | **0.8915**                                    | 2.7982   | **0.7717**                                      | 3.3562   | **0.5407**                                       | 3.9535   |

---

> > ### Author Response · Authors · 2025-12-01
> > **Reply to follow-up comment. (3/3)**
> >
> > [A] Laska, J. N., Wen, Z., Yin, W., and Baraniuk, R. G. (2011). Trust, but verify: Fast and accurate signal recovery from 1-bit compressive measurements. *IEEE Transactions on Signal Processing*, 59(11), 5289–5301.
> >
> > [B] Yan, M., Yang, Y., and Osher, S. (2012). Robust 1-bit compressive sensing using adaptive outlier pursuit. *IEEE Transactions on Signal Processing*, 60(7), 3868–3875.
> >
> > [C] Y. Plan and R. Vershynin. (2013). One-bit compressed sensing by linear programming. *Communications on Pure and Applied Mathematics*, 66(8), 1275–1297.
> >
> > [D] Y. Plan and R. Vershynin. (2012). Robust 1-bit compressed sensing and sparse logistic regression: a convex programming approach. *IEEE Transactions on Information Theory*, 59(1), 482–494.
> >
> > [E] Sjoerd Dirksen and Shahar Mendelson. (2021). Non-Gaussian hyperplane tessellations and robust one-bit compressed sensing. *Journal of the European Mathematical Society*, 23(9), 2913–2947.
> >
> > [F] Hans Christian Jung, Johannes Maly, Lars Palzer, and Alexander Stollenwerk. (2021). Quantized compressed sensing by rectified linear units. *IEEE Transactions on Information Theory*, 67(6), 4125–4149.
> >
> > [G] Zymnis, A., Boyd, S., and Candès, E. (2009). Compressed sensing with quantized measurements. *IEEE Signal Processing Letters*, 17(2), 149–152.

---

> ### Author Response · Authors · 2025-12-01
> **Reply to follow-up comment. (1/3)**
>
> **Comment 1:** If I understand it correctly, you use McLachlan et al. (2008) to argue that the sequence $\left(\boldsymbol{\mu}\_k,\boldsymbol{\xi}\_k,\boldsymbol{\varSigma}\_k\right)$ converges to some stationary point $\left(\boldsymbol{\mu}^\star,\boldsymbol{\xi}^\star,\boldsymbol{\varSigma}^\star\right)$ and then, based on this, prove linear convergence in Proposition 1/Theorem 2. Is this correct?
>
> I looked a bit deeper into the proof now, but find it hard to follow all steps. Consequently, I cannot confirm correctness of the result. Considering Figure 3 in the revised version, I agree that locally a linear rate can be observed. Whether this is also holds globally, as claimed in Proposition 1, seems less clear to me.
>
> **Response to Comment 1:** Indeed, based on the conclusions of McLachlan et al. [A], the proposed method guarantees convergence from an arbitrary initial point $\left(\boldsymbol{\mu}^{(0)},\boldsymbol{\xi}^{(0)},\boldsymbol{\varSigma}^{(0)}\right)$ to a stationary point $\left(\boldsymbol{\mu}^\star,\boldsymbol{\xi}^\star,\boldsymbol{\varSigma}^\star\right)$. Regarding the proof of the convergence rate, we first consider a single variable; without loss of generality, $\boldsymbol{\mu}$ is selected while the other two variables are held fixed. The relationship between the distance to the stationary point $\boldsymbol{\mu}^\star$ at iteration $k$, denoted as $\|\boldsymbol{\mu}^{(k)}-\boldsymbol{\mu}^\star\|$, and the distance at iteration $k+1$, $\|\boldsymbol{\mu}^{(k+1)}-\boldsymbol{\mu}^\star\|$, is given by:
>
> $$
> \|\boldsymbol{\mu}^{(k+1)}-\boldsymbol{\mu}^\star\|\_2\leq c\_{\mu}\|\boldsymbol{\mu}^{(k)}-\boldsymbol{\mu}^\star\|\_2,
> $$
>
> where $c\_{\mu}$ is a function of $\left(\boldsymbol{\mu},\boldsymbol{\xi},\boldsymbol{\varSigma}\right)$. We demonstrate that $c\_{\mu}\in(0,1)$ regardless of the values taken by $\left(\boldsymbol{\mu},\boldsymbol{\xi},\boldsymbol{\varSigma}\right)$. Since the proof imposes no assumptions on the proximity of $\boldsymbol{\mu}^{(k)}$ and $\boldsymbol{\mu}^{(k+1)}$ to the stationary point $\boldsymbol{\mu}^\star$ (i.e., it is not restricted to local convergence), this relationship can be generalized recursively from the initial point to the $k$-th iteration:
>
> $$
> \|\boldsymbol{\mu}^{(k)}-\boldsymbol{\mu}^\star\|\_2\leq \left(\max \{c\_{\mu}\}\right)^k\|\boldsymbol{\mu}^{(0)}-\boldsymbol{\mu}^\star\|\_2.
> $$
>
> This establishes the global convergence property for the variable $\boldsymbol{\mu}$. Analogous conclusions are subsequently derived for $\boldsymbol{\varSigma}$ and $\boldsymbol{\xi}$, thereby verifying that each of the three variables individually exhibits global linear convergence. Finally, invoking the theoretical results from [B], we demonstrate that if the individual components possess global linear convergence, the iterative update of these three variables within the ECM framework also exhibits global linear convergence.
>
> [A] McLachlan, G. J., and Krishnan, T. (2008). *The EM algorithm and extensions*. John Wiley & Sons.
>
> [B] Meng, X.-L., and Rubin, D. B. (1994). On the global and componentwise rates of convergence of the EM algorithm. *Linear Algebra and its Applications*, 199, 413–425.
>
> **Comment 2:** Since it only provides an empirical analysis of the algorithm’s estimation performance, the numerical benchmark should be more rigorous and should show clearer benefits in accuracy given the high computational costs of the proposed method (see detailed comments below). To me the current experiments suggest that the proposed ML estimation approach is able to slightly outperform existing methods, but pays heavily in terms of efficiency (see, e.g., Figure 1, and Table 2 Multi-bit Gaussian vs proposed methods). It might make sense to find a setting where the normal mean-variance mixture model allows to outperform existing approaches more clearly. Otherwise it is of limited use in practice. For instance, how does Figure 1 change if one considers heavier-tailed distributions?
>
> **Response to Comment 2:** Originally, the objective of Figure 1 was to establish the validity of our proposed algorithm by demonstrating that it achieves performance comparable to state-of-the-art methods in the classic 1-bit Gaussian scenario. However, we acknowledge that evaluating the algorithm on heavy-tailed data provides a more compelling illustration of its robust modeling capabilities and distinct advantages. Consequently, we have modified the experimental setup for Figure 1 by replacing the synthetic data with data sampled from a Student's $t$-distribution. The updated results demonstrate that our algorithm significantly outperforms methods predicated on Gaussian assumptions. Furthermore, even when compared against other methods specifically designed for robustness to heavy-tailed data, our proposed approach maintains a clear performance lead.

---

### Official Review · Reviewer_D8DK · 2025-10-30

**Soundness:** 3
**Presentation:** 3
**Contribution:** 3
**Rating:** 8
**Confidence:** 4

**Summary:**

The paper considers parameter estimation with quantized samples. This builds theoretically using on an ECM algorithm approach developed in a line of other works, now generalizing to the normal mean-variance mixture case.  An approximate ML function is used to approach ML estimation. An ECM algorithm is developed and convergence proofs are derived.  Numerical examples compare to baselines.

**Strengths:**

The theory is clearly developed and the ECM is shown to linearly converge to a stationary point, similar to prior analysis for the less general cases.  The method adds some generalization over past Gaussian, now with normal mean-variance mixture, adding some additional distribution parameters. The work is shown to reduce to previously developed Gaussian methods.  The authors present examples with matrix completion for a recommendation system which is an interesting real-life application.

The method might be useful for robustness against Gaussian mismatch, and the paper makes statements about heavy tailed distributions.

**Weaknesses:**

Overall the paper is strong on theory, and less so on the examples and testing. The examples are relatively benign and limited.  It isn’t clear if a local minimum is possible in the optimization, and what is lost between max-likelihood and using the bounded surrogate function approach. The overall complexity is not well characterized.  The quantization loss is also not clearly considered in the examples.

The robustness question is interesting, e.g., heavy tailed cases.  It would be useful to explore this.

**Questions:**

What happens with more than one bit quantization?  Complexity, for example.

How to characterize complexity and compare against other algorithms?

Show also a "full" quantized case to see what is lost?

---

> ### Author Response · Authors · 2025-11-22
>
> **Comment 1:** Overall the paper is strong on theory, and less so on the examples and testing. The examples are relatively benign and limited.
>
> **Response to Comment 1:** The primary objective of this work is to propose a general and robust model, together with an estimation algorithm that is equipped with convergence guarantees. Consequently, the focus of the manuscript is on analyzing the convergence properties and demonstrating the effectiveness of the proposed algorithm. Additional baseline algorithms for both quantized parameter estimation (Figure 1 and 2, Table 1 in the paper) and quantized matrix completion (Table 2 in the paper) have been incorporated, and the scale of the quantized matrix completion experiments has been increased.
> We have also conducted another experiment on 1-bit compressive sensing (Table 3 in the revised paper). The details of the 1-bit compressive sensing experiment are given in the follows:
>
> Existing methods for the 1-bit compressive sensing generally fall into three categories. The first two primarily focus on 1-bit quantization without incorporating additive noise into the original measurements [A]. The first category, termed regularizer-class algorithms, introduces additional regularization terms to the classical compressive sensing recovery problem to enforce consistency between the sparse signal and the measurements; the restricted step shrinkage (RSS) algorithm [B] is selected as a representative baseline. The second category, known as penalty-class algorithms, models sign flips between quantized and recovered measurements as penalty terms; the adaptive outlier pursuit (AOP) [C] algorithm is chosen for comparison. The third category assumes the presence of additive noise (typically Gaussian) in the original measurements [D]. Our proposed method builds upon this third category, with the model construction detailed in Section 5.2, extending the framework to the normal mean-variance mixture model.
>
> Performance comparisons are conducted using synthetic data, where the measurement dimension is denoted as $d_1=3000$ and the 30-sparse signal dimension as $d_2=1000$. To evaluate algorithmic robustness, an additive noise term exhibiting both biased and heavy-tailed characteristics is introduced to the original measurements. The signal-to-noise ratio (SNR) is defined as the ratio of the expectation of the original measurements to the variance of the noise term. Experiments are performed across various SNR levels using different algorithms. The evaluation metric is the cosine similarity between the original and the recovered sparse signals. For each SNR level, experiments are repeated 10 times, and the average cosine similarity (Cos Sim) and computational time are reported. The results are summarized in the Table 3. The GH distribution modeling achieve the largest average cosine similarity.
>
> **Table: Cosine similarity comparisons of 1-bit compressive sensing under different SNR levels.**
>
> | Method                      | SNR = 0 dB Cos Sim | SNR = 0 dB Time (s) | SNR = -5 dB Cos Sim | SNR = -5 dB Time (s) | SNR = -10 dB Cos Sim | SNR = -10 dB Time (s) |
> |-----------------------------|--------------------|---------------------|---------------------|----------------------|----------------------|-----------------------|
> | RSS                         | 0.7872             | 0.2639              | 0.6598              | 0.2651               | 0.3027               | 0.2546                |
> | AOP                         | 0.7052             | 0.2689              | 0.4041              | 0.2951               | 0.2209               | 0.2738                |
> | 1-bit Gaussian              | 0.8220             | 2.7704              | 0.5487              | 3.0875               | 0.2896               | 2.7202                |
> | 1-bit Student's $t$ ECM (prop.) | 0.8396         | 2.8573              | 0.5678              | 3.1643               | 0.3031               | 3.3174                |
> | 1-bit GHST ECM (prop.)      | 0.8893             | 2.7435              | 0.7701              | 3.4204               | 0.5237               | 4.0281                |
> | 1-bit GH ECM (prop.)        | **0.8915**         | 2.7982              | **0.7717**          | 3.3562               | **0.5407**           | 3.9535                |
>
> [A] Li, Z., Xu, W., Zhang, X., and Lin, J. (2018). A survey on one-bit compressed sensing: Theory and applications. *Frontiers of Computer Science*, 12(2), 217–230
>
> [B] Laska, J. N., Wen, Z., Yin, W., and Baraniuk, R. G. (2011). Trust, but verify: Fast and accurate signal recovery from 1-bit compressive measurements. *IEEE Transactions on Signal Processing*, 59(11), 5289–5301
>
> [C] Yan, M., Yang, Y., and Osher, S. (2012). Robust 1-bit compressive sensing using adaptive outlier pursuit. *IEEE Transactions on Signal Processing*, 60(7), 3868–3875
>
> [D] Zymnis, A., Boyd, S., and Candès, E. (2009). Compressed sensing with quantized measurements. *IEEE Signal Processing Letters*, 17(2), 149–152

---

> > ### Author Response · Authors · 2025-11-22
> >
> > **Comment 2:** It isn’t clear if a local minimum is possible in the optimization, and what is lost between max-likelihood and using the bounded surrogate function approach.
> >
> > **Response to Comment 2:** ECM is a classical algorithmic framework that provides convergence guarantees and is designed to solve the original problem rather than an approximate one. The core of the ECM algorithm is that this algorithm iteratively optimizes a surrogate function $S$ instead of directly optimizing the objective function $L\left(\boldsymbol{\theta}\right)$. In the E-step, given the current parameters, the algorithm forms a surrogate function $S(\boldsymbol{\theta};\underline{\boldsymbol{\theta}})$. This function is constructed to be a tight lower bound of the log-likelihood function $L\left(\boldsymbol{\theta}\right)$; specifically, it is guaranteed $S(\boldsymbol{\theta};\underline{\boldsymbol{\theta}})\leq L\left(\boldsymbol{\theta}\right)$ everywhere. Besides, at the point of the current parameters, we have $S(\underline{\boldsymbol{\theta}};\underline{\boldsymbol{\theta}})= L\left(\underline{\boldsymbol{\theta}}\right)$ and $S^\prime(\underline{\boldsymbol{\theta}};\underline{\boldsymbol{\theta}})= L^\prime\left(\underline{\boldsymbol{\theta}}\right)$.
> >
> > After the E-step constructs the surrogate lower bound, the CM-steps sequentially update the parameters that maximize this surrogate function. Because the surrogate was a tight lower bound at the previous parameters and we have increased its value, the value of the true log-likelihood at the new parameters $\boldsymbol{\theta}^{+}$ is guaranteed to be greater than or equal to its previous value, which is
> > \[
> > L\left(\boldsymbol{\theta}^{+}\right)\geq S(\boldsymbol{\theta}^{+};\underline{\boldsymbol{\theta}})\geq S(\underline{\boldsymbol{\theta}};\underline{\boldsymbol{\theta}})= L\left(\underline{\boldsymbol{\theta}}\right)
> > \]
> > By alternating between E and M-steps, the ECM algorithm guarantees a monotonic increase in the original objective function. This makes the EM algorithm an iterative solver for the original optimization problem, not an approximation of it. Each iteration is guaranteed to improve (or maintain) the solution with respect to the true objective. As a result, any algorithm that adheres to the ECM framework is guaranteed to converge to a local optimum (or a saddle point) of the original maximum likelihood problem [A].
> >
> > [A] McLachlan, G. J., and Krishnan, T. (2008). *The EM algorithm and extensions*. John Wiley & Sons.
> >
> > **Comment 3:** The overall complexity is not well characterized. What happens with more than one bit quantization? Complexity, for example.
> >
> > **Response to Comment 3:** Since the proposed method is an iterative algorithm, we introduce the time complexity of each iteration in the following. In each iteration, the computational complexity of the proposed algorithm can be analyzed as follows. Since the range of $\boldsymbol{y}$ is limited to discrete values, it is possible that some of the $n$ samples are identical. Therefore, when computing the statistics (i.e., the expectations, $\boldsymbol{q}$, $\boldsymbol{H}$, and $\boldsymbol{D}$ defined in Appendix A) required for parameter updates, it is sufficient to consider only distinct samples. Suppose that among the $n$ samples, there are $n_0$ distinct ones. The value of $n_0$ depends on the dimensionality $d$ of the original model and the number of discrete values that $\boldsymbol{y}$ can take (i.e., the number of bits), yielding $n_0 = \min\{(e+1)^d,\, n\}$. Then we give the time complexity in each iteration for updating parameters $\boldsymbol{\mu}$, $\boldsymbol{\varSigma}$, and $\boldsymbol{\xi}$.
> >
> > The remaining part of the Response to Comment 3 will be provided in the next comment.

---

> ### Author Response · Authors · 2025-11-22
>
> The update of $\boldsymbol{\mu}$ in (43) involves several steps. Computing the $p(\boldsymbol{y};\underline{\boldsymbol{\theta}})$ for $n_0$ distinct samples, as given in equation (5), requires evaluating the integral of the cumulative density function (CDF) of a Gaussian distribution over $z$ under the normal mean–variance mixture model. Using the Gauss–Legendre method with $K$ nodes, determining the positions and weights incurs $\mathcal{O}(K^2)$ complexity, while the Matlab built-in `mvncdf` function for multivariate normals costs $\mathcal{O}(d^3)$. Therefore, the total cost for $n_0$ $p(\boldsymbol{y};\underline{\boldsymbol{\theta}})$ evaluations is $\mathcal{O}(n_0 K d^3 + K^2 + d^2)$. Computing $p_{-1}(\boldsymbol{y};\underline{\boldsymbol{\theta}})$ requires an identical computation cost to $p(\boldsymbol{y};\underline{\boldsymbol{\theta}})$, adding another $\mathcal{O}(n_0 K d^3 + K^2 + d^2)$. The computation of $\boldsymbol{q}$ involves computing $p(x_i,\boldsymbol{y}\_{\backslash i},\underline{\boldsymbol{\theta}})$, with the $p(x_i;\underline{\boldsymbol{\theta}})$ with the determinant calculation $\mathcal{O}(d^3)$ and $p(\boldsymbol{y}\_{\backslash i}\mid x_i;\underline{\boldsymbol{\theta}})$ with the determinant calculation $\mathcal{O}(n_0 K d^3 + K^2 + d^2)$, leading to $\mathcal{O}(n_0 (K d^3 + d^3) + K^2 + d^2)$. Finally, the update of $\boldsymbol{\mu}^{+}$ has complexity $\mathcal{O}(d^2)$.
>
> The update of $\boldsymbol{\xi}$ in (50) follows a similar structure to that of $\boldsymbol{\mu}$. It also requires $p(\boldsymbol{y};\underline{\boldsymbol{\theta}})$ for $n_0$ with complexity $\mathcal{O}(n_0 K d^3 + K^2 + d^2)$, and $p_1(\boldsymbol{y};\underline{\boldsymbol{\theta}})$ for $n_0$ with the same complexity. Evaluating $\boldsymbol{q}\_1$ also takes the identical computation cost to $\boldsymbol{q}$, which is $\mathcal{O}(n_0 (K d^3 + d^3) + K^2 + d^2)$. Finally, updating $\boldsymbol{\xi}^{+}$ takes $\mathcal{O}(d^2)$.
>
> The update of $\boldsymbol{\varSigma}$ in (57) involves computing $p(\boldsymbol{y};\underline{\boldsymbol{\theta}})$ for $n_0$ distinct samples in the third time with $\mathcal{O}(n_0 K d^3 + K^2 + d^2)$. Then we construct $n_0$ matrices $\boldsymbol{H}\_1$ and $\boldsymbol{D}\_1$. Each entry of $\boldsymbol{H}\_1$ requires evaluations of $p\_{1}(x_i,x_j;\underline{\boldsymbol{\theta}})$ and $p\_{1}(\boldsymbol{y}\_{\backslash i,j}\mid x_i,x_j;\underline{\boldsymbol{\theta}})$ with complexity $\mathcal{O}(K d^3 + d^3 + K^2 + d^2)$. For the total $d^2$ entries across $n_0$ matrices, this yields $\mathcal{O}(n_0 (K d^5 + d^5) + K^2 d^2 + d^4)$. However, in practical computation, the time complexity of $d^5$ is prohibitively large. According to Lemma 9 in Appendix B.3 in the paper, $\boldsymbol{H}\_1 + \boldsymbol{D}\_1$ can also be computed via numerical differentiation. In this case, the time complexity is reduced to $\mathcal{O}(n_0d^3)$. The construction of $\boldsymbol{D}$ requires the computation of both $\boldsymbol{q}$ and $\boldsymbol{q}\_1$, costing $\mathcal{O}(n_0 (K d^3 + d^3) + K^2 + d^2)$. The computation of $\frac{1}{n}\sum_{t=1}^n p^{-1}(\boldsymbol{y}_{t};\underline{\boldsymbol{\theta}})\underline{\boldsymbol{\varSigma}}(\boldsymbol{H}\_{1,t}+\boldsymbol{D}\_{1,t})\underline{\boldsymbol{\varSigma}}$ has complexity $\mathcal{O}(n_0 d^3)$, and the update of $\boldsymbol{\varSigma}^{+}$ costs $\mathcal{O}(n_0 d^2)$.
> From the above derivation, it can be seen that the complexity is linearly related to $n_0$, where $n_0 = \min\{(e+1)^d,\, n\}$. In particular, when $(e+1)^d < n$, the parameters $e$, which control the number of bits, and $n_0$ exhibit a power-law relationship.
>
> **Comment 4:** The quantization loss is also not clearly considered in the examples.
>
> **Response to Comment 4:** Our work primarily focuses on designing an algorithm that estimates certain statistical properties of the original (pre-quantization) data from quantized observations. We intentionally did not provide a theoretical derivation of the quantization loss in the paper, as this was outside the scope of our main contributions. Nevertheless, the quantization loss can be observed empirically in our experiments by comparing the mean squared error between parameters estimated from quantized data and the corresponding true parameters from the synthetic unquantized data.

---

> > ### Author Response · Authors · 2025-11-22
> >
> > **Comment 5:** Robustness to heavy-tailed cases is mentioned but not explored experimentally.
> >
> > **Response to Comment 5:** In the experiments, we have incorporated experiments that explicitly examine robustness to heavy-tailed distributions. For quantized correlation estimation and quantized compressive sensing, we now generate synthetic data from heavy-tailed distributions and perform estimation using heavy-tailed models, including Student's $t$, GHST, and GH distributions. For quantized matrix completion, we additionally evaluate on a dataset with heavy-tailed characteristics, again using Student's $t$, GHST, and GH distributions for estimation. In addition, in the newly included quantized compressive sensing experiments (Table 3), heavy-tailed and skewed noise is injected into the synthetic data, under which conditions the proposed algorithm exhibits superior performance.
> >
> > **Comment 6:** How to characterize complexity and compare it against other algorithms?
> >
> > **Response to Comment 6:** Because existing algorithms employ different underlying ideas for estimation, a direct comparison of their time complexities is difficult. Instead, we provide a comparison in terms of empirical time cost, which more accurately reflects the actual computational effort required in practice. We have included these time cost measurements in quantized parameter estimation (Figures 1(b) and 2(b)), quantized matrix completion (Tables 2), and quantized compressive sensing (Table 3) of the revised manuscript to facilitate a clear comparison with other algorithms.
> >
> > Since the ECM algorithm is an iterative procedure, it does not have an advantage in terms of computational time compared with approximate methods or those designed for specific cases (such as 1-bit or Gaussian settings). However, the experimental results show that our method consistently achieves better performance, because it can more effectively model the latent heavy-tailed behavior and skewness in the original data.
> >
> > **Comment 7:** Can you also show a “full” quantized case to evaluate the loss compared to unquantized performance?
> >
> > **Response to Comment 7:**
> > As the number of quantization bits increases, our algorithm achieves progressively lower MSE, approaching the performance of the unquantized case. In the revised manuscript, Figure 2(a) presents the numerical experimental results across different bit settings, including the unquantized MLE results for direct comparison. We compared the results obtained by using the EM algorithm to perform MLE on the unquantized data that follows a Student's $t$ distribution (Unquantized MLE). In Figure 2(a), we can observe that as the number of bits increases, the MSE of the correlation matrix estimated by our algorithm decreases and gradually approaches the result of Unquantized MLE. Figure 2(b) further illustrates that the time cost of our algorithm increases with the number of bits, which is consistent with our theoretical analysis. These additions allow the performance and computational trade-offs of the fully quantized cases to be clearly evaluated alongside the unquantized baseline.

---

### Official Review · Reviewer_F6qK · 2025-10-30

**Soundness:** 3
**Presentation:** 2
**Contribution:** 2
**Rating:** 4
**Confidence:** 3

**Summary:**

The paper proposes an expectation/conditonal maximization (ECM) algorithm for estimation of signals following a mean variance mixture model from quantized measurements. Following the common premise for EM algorithms, the E step estimates latent variables based on observations and existing model parameter estimates and the CM step updates the parameters to best match the latent variables and observations. The paper includes convergence results for the model parameters and shows examples in covariance estimation, matrix completion, and compressed sensing.

**Strengths:**

The paper generalizes existing frameworks by letting the quantization scheme be arbitrary rather than just binary (e.g., sign preservation) and using a broader model for the signals (vs. the usual multivariate Gaussian) and shows agreement between new and existing results in these cases.

**Weaknesses:**

Since the paper is focused on broadening the model applied to solve existing problems currently addressed with narrow models, it would be good to get a discussion of the benefits afforded by this contribution. The experiments do not seem to need to leverage the broader model, and a comparison with existing approaches is lacking as detailed below.

The presentation is not always fully clear; there are multiple instances of notation used without introduction (detailed at the end of this response).

Noise does not appear to be considered in the acquisition process.

The numerical comparisons are for the most part focusing on the role of mismatch in the modeling on the performance of estimation, but there is a lack of comparison with previously proposed approaches. Although some competitors are mentioned in page 8, it is not clear if the comparison is against an ECM approach that uses a particular narrower data model or against the approaches of the cited references, which are not EM-based. There is no consideration of the computational cost of the EM approach to contrast against accuracy metrics (e.g., Table 2).

It would be good to have a discussion of applications for the quantized covariance estimation problem.

Line 115: Parameters mu, xi, Sigma have not been defined
Line 131: the Sigma^{-1} norm has not been defined (but one can guess)
Line 168: "theta underlined" is not previously defined
Line 169: “irrelevant of” does not seem to be appropriate wording. Perhaps independent or "does not depend"?
Title: the PDF title does not match the title in the database record.

**Questions:**

Is there a source for the statement about Netflix movie recommendations in line 275? It may hark back to "The Netflix Prize" from the decade of the 2000's, but it's not clear this is still true; a more generic statement could be made about recommendation systems though. See for example https://dl.acm.org/doi/10.1145/2891406

Can you elaborate on "the constraints in certain prior works" mentioned in line 312? It would better inform the choices made in Section 6.1, particularly because it's not clear that a comparison to existing approaches is being provided.

Given the closeness of the lines in Figure 1, would a wider zoom or a table be more appropriate to convey the results? Or alternatively, is it important to show which algorithm is better when their performances are very similar to one another?

Can you provide motivation or a citation for the use of the surrogate (16)? It is not clear how it connects to the mean variance mixture model being assumed. Perhaps it would be clearer to explicitly write an instantiation of the ECM algorithm of page 4 for each application?

---

> ### Author Response · Authors · 2025-11-22
>
> **Comment 1:** Since the paper is focused on broadening the model applied to solve existing problems currently addressed with narrow models, it would be good to get a discussion of the benefits afforded by this contribution. The experiments do not seem to need to leverage the broader model, and a comparison with existing approaches is lacking, as detailed below.
>
> **Response to Comment 1:** Our paper extends the traditional ML estimation algorithm, which has been limited to the 1-bit Gaussian setting, to an approach that can handle the multi-bit normal mean–variance model. Introducing the multi-bit framework can significantly improve the accuracy of estimating the statistical properties of the original data, as demonstrated in {Figure 2}. Furthermore, when the original data exhibit pronounced heavy-tailed or skewed characteristics, the conventional Gaussian model fails to capture these properties adequately. In such cases, the normal mean–variance model, which includes Student’s $t$, generalized hyperbolic skew $t$ (GHST), and generalized hyperbolic (GH) distributions, can provide effective modeling. This advantage is reflected in quantized parameter estimation (Figures 1, 2, 3, and 4 and Table 1), quantized matrix completion (Table 2), and quantized compressive sensing (Table 3).
>
> **Comment 2:** The presentation is not always fully clear; there are multiple instances of notation used without introduction.
>
> Line 115: Parameters mu, xi, Sigma have not been defined Line 131: the $\Sigma^{-1}$ norm has not been defined (but one can guess).
>
> Line 168: "theta underlined" is not previously defined.
>
> Line 169: “irrelevant of” does not seem to be appropriate wording. Perhaps independent or "does not depend"?
>
> Title: the PDF title does not match the title in the database record.
>
> **Response to Comment 2:** We thank the reviewer for their careful and thorough review.
>
> Regarding Line 115, we have corrected the issue to ensure that all variables are properly defined before use.
> For Line 131, we would like to clarify that the definition of the matrix norm was already provided in the submitted draft as
> $$\Vert\boldsymbol{x}\Vert_{\boldsymbol{A}} = \boldsymbol{x}^\top \boldsymbol{A} \boldsymbol{x}.$$
>
> For Line 168, the meaning of $\underline{\boldsymbol{\theta}}$ was introduced in a footnote on the same page in the initial draft. It denotes that, throughout this paper, underlined variables correspond to values that are given as constants.
>
> In Line 169, our intention was that the value of the constant term does not change when the parameter $\boldsymbol{\theta}$ changes. We agree that the phrasing ''independent of'' is more appropriate than ''irrelevant of'' in this context, and we will change it to ''independent of'' in the revised version to ensure clarity and correctness.
>
> We thank the reviewer for noting the mismatch between the PDF title and the database record. This discrepancy was unintended. The PDF title will be corrected to ensure consistency.
>
> **Comment 3:** Noise does not appear to be considered in the acquisition process.
>
> **Response to Comment 3:** In our work, we model the raw data as random variables following the normal mean-variance mixture model. Under this modeling assumption, the noise present in the signal acquisition stage is inherently taken into account. Regarding the noise introduced during the quantization process, to the best of our knowledge, we have found that the existing literature has scarcely provided substantial investigation into this direction. Consequently, our current manuscript does not address the quantization noise.
>
> **Comment 4:** The numerical comparisons are for the most part focusing on the role of mismatch in the modeling on the performance of estimation, but there is a lack of comparison with previously proposed approaches. Although some competitors are mentioned in page 8, it is not clear if the comparison is against an ECM approach that uses a particular narrower data model or against the approaches of the cited references, which are not EM-based.
>
> **Response to Comment 4:** In the 1-bit Gaussian setting, many existing methods for estimating the correlation matrix have already been compared (Figure 1). In the multi-bit and non-Gaussian settings, several existing methods are added for comparison in the revised manuscript (Figure 2). Figures 3 and 4 illustrate the convergence behavior of the proposed algorithm and therefore do not include comparisons with existing methods. In Tables 2 and 3, extensive comparisons are conducted with many effective existing methods for quantized matrix completion and quantized compressive sensing.
>
> (The remaining part of the Response to Comment 4 will be provided in the next comment.)

---

> > ### Author Response · Authors · 2025-11-22
> >
> > In order to avoid potential confusion regarding which methods are proposed in this work and which methods employ the ECM algorithm, in all figures and tables that include other methods, our proposed methods are explicitly labeled with ``(prop.)'' without the situations that we only compare with our proposed algorithms. Each method included in the comparison is explicitly annotated with the specific algorithm it employs, including the ECM algorithm.
> >
> > **Comment 5:** There is no consideration of the computational cost of the EM approach to contrast against accuracy metrics (e.g., Table 2).
> >
> > **Response to Comment 5:** We have now incorporated an analysis of computational time into the experimental figures and tables. In the quantized parameter estimation, we update Figures 1 and 2 to show the time cost of both our proposed method and the existing methods. We also have updated Tables 2 and 3 to include the time cost for each method, providing a direct comparison of the computational cost alongside the accuracy metrics.
> >
> > Since the ECM algorithm is an iterative procedure, it does not have an advantage in terms of computational time compared with approximate methods or those designed for specific cases (such as 1-bit or Gaussian settings). However, the experimental results show that our method consistently achieves better performance because it can more effectively model the latent heavy-tailed behavior and skewness in the original data.
> >
> >
> > **Comment 6:** It would be good to have a discussion of applications for the quantized covariance estimation problem.
> >
> > **Response to Comment 6:** We appreciate the reviewer's suggestion. Quantized covariance estimation is central to many signal processing tasks where analog-to-digital converters impose quantization on observed signals. From such quantized data, one can infer the autocorrelation function of the underlying unquantized random process, which is critical in applications such as direction-of-arrival estimation [A] and power estimation in wireless sensor networks [B]. Beyond array signal processing, quantized covariance estimation also plays an important role in spectrum sensing [C] and networked sensing [D] scenarios. We will incorporate a brief discussion summarizing these application domains in the revised manuscript.
> >
> > [A] Bar-Shalom, O., and Weiss, A. J. (2002). DOA estimation using one-bit quantized measurements. *IEEE Transactions on Aerospace and Electronic Systems*, 38(3), 868–884.
> >
> > [B] Mo, J., Schniter, P., and Heath, R. W. (2017). Channel estimation in broadband millimeter wave MIMO systems with few-bit ADCs. *IEEE Transactions on Signal Processing*, 66(5), 1141–1154.
> >
> > [C] Yang, J., Song, Z., Zhang, H., and Gao, Y. (2025). Compressive spectrum sensing with 1-bit ADCs. *IEEE Transactions on Vehicular Technology*.
> >
> > [D] Chi, Y., and Fu, H. (2017). Subspace learning from bits. *IEEE Transactions on Signal Processing*, 65(17), 4429–4442.
> >
> > **Comment 7:** Is there a source for the statement about Netflix movie recommendations in line 275? It may hark back to "The Netflix Prize" from the decade of the 2000's, but it's not clear this is still true; a more generic statement could be made about recommendation systems though. See for example https://dl.acm.org/doi/10.1145/2891406
> >
> > **Response to Comment 7:** We thank the reviewer for pointing this out. Our original statement was based on the introduction to the matrix completion problem in [A]. After reviewing the reference provided by the reviewer, we have revised the wording to make it more accurate:
> >
> > The study of matrix completion was popularized following the Netflix Million Dollar Challenge, which posed the task: accurately predicting the values of those entries with a user–item matrix in which entries represent item ratings.
> >
> > We will incorporate this revised statement into the manuscript to ensure precision and clarity.
> >
> > [A] Bhaskar, S. A. (2016). Probabilistic low-rank matrix completion from quantized measurements. *Journal of Machine Learning Research*, 17(60), 1–34.

---

> > > ### Author Response · Authors · 2025-11-22
> > >
> > > **Comment 8:** Can you elaborate on "the constraints in certain prior works" mentioned in line 312? It would better inform the choices made in Section 6.1, particularly because it's not clear that a comparison to existing approaches is being provided.
> > >
> > > **Response to Comment 8:** The phrase “the constraints in certain prior works” refers specifically to the limitations discussed in Question 2. In particular, works [A]-[G] are able to estimate the correlation matrix only in the case of one-bit quantization. Furthermore, in works [A], [B], [C], [D], and [G], an additional assumption is made that the underlying data follow a Gaussian distribution. These constraints motivated the choices made in Section 6.1 and the design of our comparative evaluation. The above explanation and discussion have already been incorporated into the revised manuscript.
> > > [A] J.H. Van Vleck and D. Middleton. The spectrum of clipped noise. *Proceedings of the IEEE*, 54(1):2–19, 1966.
> > > [B] Chun-Lin Liu and Zi-Min Lin. One-bit autocorrelation estimation with non-zero thresholds. In *2021 IEEE International Conference on Acoustics, Speech and Signal Processing (ICASSP)*, pp. 4520–4524. IEEE, 2021.
> > > [C] Yu-Hang Xiao, Lei Huang, David Ramírez, Cheng Qian, and Hing Cheung So. Covariance matrix recovery from one-bit data with non-zero quantization thresholds: Algorithm and performance analysis. *IEEE Transactions on Signal Processing*, 71:4060–4076, 2023.
> > > [D] Chun-Lin Liu and Yi-Hung Chou. Approximation and analysis of the one-bit Hermite law. In *2025 IEEE International Conference on Acoustics, Speech and Signal Processing (ICASSP)*, pp. 1–5. IEEE, 2025.
> > > [E] Sjoerd Dirksen, Johannes Maly, and Holger Rauhut. Covariance estimation under one-bit quantization. *The Annals of Statistics*, 50(6):3538–3562, 2022.
> > > [F] Sjoerd Dirksen and Johannes Maly. Tuning-Free One-Bit Covariance Estimation Using Data-Driven Dithering. *IEEE Transactions on Information Theory*, 70(7):5228–5247, July 2024.
> > > [G] Arian Eamaz, Farhang Yeganegi, and Mojtaba Soltanalian. Covariance recovery for one-bit sampled non-stationary signals with time-varying sampling thresholds. *IEEE Transactions on Signal Processing*, 70:5222–5236, 2022.
> > >
> > > **Comment 9:** Given the closeness of the lines in Figure 1, would a wider zoom or a table be more appropriate to convey the results? Or alternatively, is it important to show which algorithm is better when their performances are very similar to one another?
> > >
> > > **Response to Comment 9:**
> > > In the revised manuscript, we have clarified that the numerical experiments demonstrate the superiority of our proposed method in terms of the mean squared error (MSE) of the correlation matrix estimation in multi-bit scenarios. Since works [A], [B], and [C] all adopt the similar underlying model, the differences among the proposed algorithms lie primarily in estimation accuracy and computational efficiency. As a result, the performance curves in Figure 1 are very close to each other, leading to the small gaps observed.
> > >
> > > [A] Chun-Lin Liu and Zi-Min Lin. One-bit autocorrelation estimation with non-zero thresholds. In *2021 IEEE International Conference on Acoustics, Speech and Signal Processing (ICASSP)*, pp. 4520–4524. IEEE, 2021.
> > > [B] Yu-Hang Xiao, Lei Huang, David Ramírez, Cheng Qian, and Hing Cheung So. Covariance matrix recovery from one-bit data with non-zero quantization thresholds: Algorithm and performance analysis. *IEEE Transactions on Signal Processing*, 71:4060–4076, 2023.
> > > [C] Chun-Lin Liu and Yi-Hung Chou. Approximation and analysis of the one-bit Hermite law. *In 2025 IEEE International Conference on Acoustics, Speech and Signal Processing (ICASSP)*, pp. 1–5. IEEE, 2025.
> > >
> > > **Comment 10:** Can you provide motivation or a citation for the use of the surrogate (16)? It is not clear how it connects to the nor mean-variance mixture model being assumed. Perhaps it would be clearer to explicitly write an instantiation of the ECM algorithm of page 4 for each application?
> > >
> > > **Response to Comment 10:** In the manuscript, the surrogate function for quantized matrix completion is obtained from the original MLE problem via the E-step of the proposed ECM method. However, some intermediate steps are omitted, which may lead to confusion. In the following, we will provide the detailed derivation steps and clarify the connection between the surrogate function and the assumed normal mean-variance mixture model.
> > >
> > > Consider a general normal mean-variance mixture model as
> > > \begin{equation}
> > >     \boldsymbol{x} = \boldsymbol{\mu}+z\boldsymbol{\xi}+\left(z\boldsymbol{\varSigma}\right)^\frac{1}{2}\boldsymbol{\epsilon}.\nonumber
> > > \end{equation}
> > >
> > > (The remaining part of the Response to Comment 10 will be provided in the next comment.)

---

> ### Author Response · Authors · 2025-11-22
>
> Given that $\boldsymbol{y}=\mathcal{Q}\left(\boldsymbol{x}\right)$, the density function of $\boldsymbol{y}$ is given as follows:
> \begin{equation}
>     p(\boldsymbol{y};\boldsymbol{\theta})=\int_{\mathcal{Q}^{-1}(\boldsymbol{y})}p(\boldsymbol{x};\boldsymbol{\theta})\mathrm{d}\boldsymbol{x}=\int_{\mathcal{Q}^{-1}(\boldsymbol{y})}\int_0^{\infty}p(\boldsymbol{x}\mid z;\boldsymbol{\theta})p(z)\mathrm{d}z\mathrm{d}\boldsymbol{x},
> \end{equation}
> where $\mathcal{Q}^{-1}(\boldsymbol{y})$ maps the quantized data $\boldsymbol{y}$ into a hyper-rectangle whose projection on each dimension $i$ is an interval corresponding to $[\mathcal{Q}(\boldsymbol{y})]_{i}$.
>
> Given $n$ independent and identically distributed samples $\boldsymbol{y}_{1},\ldots,\boldsymbol{y}_{n}$, the ML estimation problem is given by
> \begin{equation}
>     \max_{\boldsymbol{\theta}}\quad L(\boldsymbol{\theta})= \sum_{t=1}^n\log p(\boldsymbol{y}_t; \boldsymbol{\theta}).
> \end{equation}
>
> Regarding $z$ as a hidden variable, applying Jensen's inequality leads to
>
> $L(\boldsymbol{\theta}) \geq\sum_{t=1}^{n}\mathsf{E}_{\boldsymbol{x}_t\mid \boldsymbol{y}_t;\underline{\boldsymbol{\theta}}}\left[\mathsf{E} [\log p(z_t\mid\boldsymbol{x}_t;\boldsymbol{\theta})]\right]+\mathrm{const}.\triangleq S(\boldsymbol{\theta};\underline{\boldsymbol{\theta}}).$
>
> Hence, the surrogate function $S(\boldsymbol{\theta};\underline{\boldsymbol{\theta}})$ can be further expressed as follows:
> \begin{align}
> S(\boldsymbol{\theta};\underline{\boldsymbol{\theta}})
> &=\sum_{t=1}^{n}\left[\mathsf{E}_{z_t\mid \boldsymbol{y}_t;\underline{\boldsymbol{\theta}}}\left[\log p(z_t)\right]-\frac{1}{2}\log\det\boldsymbol{\varSigma}\right. \left.-\frac{1}{2}\mathrm{Tr}\left(\left(\boldsymbol{U}_t-2\boldsymbol{v}_t\boldsymbol{\mu}^\top+\iota_t\boldsymbol{\mu}\boldsymbol{\mu}^\top\right)-2(\boldsymbol{w}_t-\boldsymbol{\mu})\boldsymbol{\xi}^\top+\zeta_t\boldsymbol{\xi}\boldsymbol{\xi}^\top\right)\boldsymbol{\varSigma}^{-1}\vphantom{\frac{1}{2}\log\det\boldsymbol{\varSigma}}\right]+\mathrm{const}.,\nonumber
> \end{align}
>
> The derivation provided above corresponds to the general case. We will subsequently present how this derivation applies specifically to the quantized matrix completion setting. The goal of the low-rank matrix completion problem is to recover an unknown low-rank matrix $\boldsymbol{M} \in \mathbb{R}^{d_1 \times d_2}$ from an observed, yet incomplete, matrix. Let $\boldsymbol{X}$ denote a matrix whose entries are drawn from a normal mean–variance mixture distribution. We use $\mu_{ij}$ and $x_{ij}$ to represent the $(i,j)$-th entries of $\boldsymbol{M}$ and $\boldsymbol{X}$, respectively. Based on the normal mean-variance model, the relationship between $\mu_{ij}$ and $x_{ij}$ is given by
> \begin{equation}
>     x_{ij} = \mu_{ij} + z\xi + z^{1/2} \sigma \epsilon,
>     \nonumber
> \end{equation}
> where $\mu_{ij}$ serves as the location parameter of $x_{ij}$ and both $\xi$ and $\sigma$ are given constants.  In the quantization scenario, we have  $\boldsymbol{Y} = \mathcal{Q}(\boldsymbol{X})$. The entire matrix $\boldsymbol{Y}$ is not available for observation. Let $\boldsymbol{\Omega}$ denote the index set of the observed entries. Our goal is to recover $\boldsymbol{M}$ from incomplete $\boldsymbol{Y}$, which is equivalent to estimating all the $\mu_{ij}$ in the matrix $\boldsymbol{M}$.
>
> Since $x_{ij}$ follows a univariate normal mean-variance model and $\boldsymbol{Y}=\mathcal{Q}\left(\boldsymbol{X}\right)$, the density function of $y_{ij}$ is equivalent to the density function above under the univariate case. Hence, the negative log-likelihood function for $y_{ij}$ with ${ij}\in\boldsymbol{\Omega}$ is
> \[
> \sum_{(i,j)\in \boldsymbol{\Omega}}\log p\left(y_{ij}\mid\boldsymbol{M}\right),
> \]
> which is the objective function of (15). Applying Jensen's inequality, we have the surrogate function
> \begin{align}
> S(\boldsymbol{M};\underline{\boldsymbol{M}})
> =\sum_{(i,j)\in \boldsymbol{\Omega}}\left[\mathsf{E}\_{z\_{t}\mid \boldsymbol{y}\_{t};\underline{\boldsymbol{\theta}}}\left[\log p(z_t)\right]-\frac{1}{2}\log\det\sigma^2-\frac{1}{2\sigma^2}\left(\left(u_{ij}-2v_{ij}\mu+\iota_{ij}\mu^2\right)-2(w_{ij}-\mu)\xi+\zeta_{ij}\xi^2\right)\right]+\mathrm{const}.,\nonumber
> \end{align}
>
> where $u\_{ij}$, $v\_{ij}$, and $w\_{ij}$ are the univariate versions of $\boldsymbol{U}\_{ij}$, $\boldsymbol{v}\_{ij}$, and $\boldsymbol{w}\_{ij}$, respectively.
> Ignoring the terms which are independent with $\mu\_{ij}$, the surrogate function becomes
>
> The remaining part of the Response to Comment 10 will be provided in the next comment.

---

> > ### Author Response · Authors · 2025-11-22
> >
> > \begin{align}
> > S(\boldsymbol{M};\underline{\boldsymbol{M}})
> > &=\frac{1}{2\sigma^2}\sum_{(i,j)\in \boldsymbol{\Omega}}\left(2v_t\mu-\iota_t\mu^2-2\mu\xi\right)+\mathrm{const}.\nonumber
> > \end{align}
> > Making a low-rank factorization to the matrix $\boldsymbol{M}$ as $\boldsymbol{M} = \boldsymbol{A} \boldsymbol{B}^\top$, we have $\mu\_{ij}=\boldsymbol{a}\_{i} \boldsymbol{b}\_{j}^\top$. Setting $e\_{ij}=\frac{v\_{ij}-\xi}{\iota\_{ij}}$, we have that maximizing $S(\boldsymbol{M};\underline{\boldsymbol{M}})$ is equivalent to maximize
> > $
> > \sum\_{(i,j) \in \boldsymbol{\Omega}} \left( \boldsymbol{a}\_{i} \boldsymbol{b}\_{j}^\top - e\_{ij} \right)^{2},
> > $
> > which is (16) in the manuscript.
> > Besides, we also revised the manuscript to make the introduction of the surrogate (16) clear.

---

### Official Review · Reviewer_Cium · 2025-11-07

**Soundness:** 3
**Presentation:** 2
**Contribution:** 2
**Rating:** 4
**Confidence:** 3

**Summary:**

This paper proposes the ECM (expectation conditional maximization) algorithm for quantized maximum-likelihood estimation. The authors prove convergence guarantees for the algorithm and test the algorithm on covariance estimation and matrix completion tasks.

**Strengths:**

1. This paper extends previous works on quantized statistical estimation to multiple bit and a more general normal mean-varaince mixture model.
2. The proposed method demonstrates generality to some extent, which can be applied to covariance estimation and matrix completion, the latter task is very important in recommendation systems.

**Weaknesses:**

1. It is not clear whether to what extent the normal mean-variance mixture model is useful in machine learning applications. The experiment is only performed on MovieLens 100k data, and the performance is not compared with other machine learning approaches to demonstrate the effectiveness of the maximmum-likelihood estimator.
2. The scalability of the approach is not clear. The MovieLens 1M and 20M benchmark may be more relevant for modern machine learning applications.

**Questions:**

What are other potential applications of the normal mean-variance mixture model in other machine learning tasks?

---

> ### Author Response · Authors · 2025-11-22
>
> **Comment 1:** It is not clear whether to what extent the normal mean-variance mixture model is useful in machine learning applications. What are other potential applications of the normal mean-variance mixture model in other machine learning tasks?
>
> **Response to Comment 1:**
> The normal mean-variance mixture model, which encompasses many classical distributions (Gaussian, Student's \(t\), and generalized hyperbolic), is a flexible and power statistical model. In fact, it has been applied in various machine learning domains, including pattern recognition [A], variational inference [B], factor analysis [C], sparse representation [D], and spectral analysis [E]. In the revised manuscript, we will incorporate a brief review of these representative works to clarify the relevance of the normal mean-variance mixture model to the machine learning field.
>
> [A] Chatzis, S. P. (2010). Hidden Markov models with nonelliptically contoured state densities. *IEEE Transactions on Pattern Analysis and Machine Intelligence*, 32(12), 2297–2304.
>
> [B] Lukashchuk, M., Trésor, R., Nuijten, W. W. L., Senoz, I., and de Vries, B. (2025). The quotient Bayesian learning rule. *2025 The Thirty-Ninth Annual Conference on Neural Information Processing Systems (NIPS)*.
>
> [C] Wei, Y., Tang, Y., and McNicholas, P. D. (2018). Flexible high-dimensional unsupervised learning with missing data. *IEEE Transactions on Pattern Analysis and Machine Intelligence*, 42(3), 610–621.
>
> [D] Archambeau, C., and Bach, F. (2008). Sparse probabilistic projections. *Advances in Neural Information Processing Systems*, 21.
>
> [E] Srivastava, A., Liu, X., and Grenander, U. (2002). Universal analytical forms for modeling image probabilities. *IEEE Transactions on Pattern Analysis and Machine Intelligence*, 24(9), 1200–1214.
>
> **Comment 2:**  The experiment is only performed on MovieLens 100k data, and the performance is not compared with other machine learning approaches to demonstrate the effectiveness of the maximum-likelihood estimator. The scalability of the approach is not clear. The MovieLens 1M and 20M benchmark may be more relevant for modern machine learning applications.
>
> **Response to Comment 2:** The reason for previously choosing the 100k dataset was that we want to verify the effectiveness of the algorithm and demonstrate its superior performance. However, your suggestion is correct in pointing out that scalability is indeed an important aspect that also needs to be considered.
> To address these concerns, we have expanded our experimental evaluation from MovieLens 100k to the larger MovieLens 1M dataset. We also add the classic machine learning approaches for matrix completion, including the singular value decomposition (SVD [A]) model, the \(\ell_2\)-regularized matrix factorization (\(\ell_2\)-regularized  [B]) model, and the nuclear norm minimization (nuclear norm [C]) model. Furthermore, we also compare with some approaches are established upon random variable model assumptions. These methods include: directly modeling \(x_{ij}\) in (14) under a Gaussian assumption without quantization (Gaussian [D]); performing 1-bit quantization with a threshold of 3.5 under the premise that \(x_{ij}\) follows a Gaussian distribution (1-bit Gaussian [E]); and interpreting the observed ratings of 1--5 as the outcome of multi-bit quantization applied to \(x_{ij}\) under a Gaussian assumption (multi-bit Gaussian [F]). Our proposed method is regarded as an extension of the multi-bit Gaussian approach by extending the Gaussian assumption to a normal mean-variance mixture model, comprising multi-bit Student's \(t\), multi-bit GHST, and multi-bit GH distributions.
>
> The updated results on the MovieLens 1M dataset are presented in the following table. As shown, our proposed method (Multi-bit GH ECM) achieves the highest training accuracy and the lowest test RMSE, outperforming both standard baselines and alternative probabilistic models. These results demonstrate that our approach scales effectively to larger datasets while maintaining superior performance.
>
> | **Method**                         | **Time (s)** | **Accuracy** | **RMSE**   |
> |------------------------------------|--------------|--------------|------------|
> | SVD                                | 364.36       | 0.4261       | 0.9148     |
> | L2-regularization                  | 165.08       | 0.4388       | 0.9355     |
> | Nuclear Norm                       | 867.39       | 0.3802       | 1.1662     |
> | Gaussian                           | 25.26        | 0.4363       | 0.9486     |
> | 1-bit Gaussian                     | 110.44       | 0.4217       | 0.9659     |
> | Multi-bit Gaussian ECM             | 90.84        | 0.4453       | 0.9151     |
> | Multi-bit Student's \(t\) ECM (prop.) | 232.16    | 0.4464       | 0.9091     |
> | Multi-bit GHST ECM (prop.)         | 252.23       | 0.4495       | 0.9076     |
> | Multi-bit GH ECM (prop.)           | 312.32       | **0.4510**   | **0.8892** |

---

> ### Author Response · Authors · 2025-11-22
>
> The experiments demonstrate that, after the scale is increased, our algorithm still maintains better performance.
>
> [A] Sarwar, B. M., Karypis, G., Konstan, J. A., and Riedl, J. T. (2000). Application of dimensionality reduction in recommender systems: A case study. *WebKDD Workshop at the ACM SIGKKD*, ACM.
>
> [B] Paterek, A. (2007). Improving regularized singular value decomposition for collaborative filtering. *Proceedings of the KDD Cup and Workshop*, 394–401, ACM.
>
> [C] Cai, J.-F., Candès, E. J., and Shen, Z. (2010). A singular value thresholding algorithm for matrix completion. *SIAM Journal on Optimization*, 20(4), 1956–1982.
>
> [D] Candès, E., and Plan, Y. (2010). Matrix completion with noise. *Proceedings of the IEEE*, 98(6), 925–936.
>
> [E] Davenport, M. A., Plan, Y., Van Den Berg, E., and Wootters, M. (2014). 1-bit matrix completion. *Information and Inference*, 3(3), 189–223.
>
> [F] Bhaskar, S. A. (2016). Probabilistic low-rank matrix completion from quantized measurements. *Journal of Machine Learning Research*, 17(60), 1–34.
>
> [G] Li, Z., Xu, W., Zhang, X., and Lin, J. (2018). A survey on one-bit compressed sensing: Theory and applications. *Frontiers of Computer Science*, 12(2), 217–230.
>
> [H] Laska, J. N., Wen, Z., Yin, W., and Baraniuk, R. G. (2011). Trust, but verify: Fast and accurate signal recovery from 1-bit compressive measurements. *IEEE Transactions on Signal Processing*, 59(11), 5289–5301.
>
> [I] Yan, M., Yang, Y., and Osher, S. (2012). Robust 1-bit compressive sensing using adaptive outlier pursuit. *IEEE Transactions on Signal Processing*, 60(7), 3868–3875.
>
> [J] Zymnis, A., Boyd, S., and Candès, E. (2009). Compressed sensing with quantized measurements. *IEEE Signal Processing Letters*, 17(2), 149–152.

---

### Note · Authors · 2026-01-28

I have read and agree with the venue's withdrawal policy on behalf of myself and my co-authors.

---

### Meta-Review · Area_Chair_khrV · 2025-12-18

**Summary:**

This paper studies an iterative approach to maximum likelihood estimation (MLE). Convergence to stationary point was established. Authors also tested the algorithm on movie recommendations. Reviewers have divergent ratings. On the positive side, reviewers believe that the technical contribution is strong. On the other side, reviewers found that the data assumption needs more justification, and more baseline methods and applications need to be added.

The AC read through the paper. The AC cannot fully agree with the strength of technical contribution: using MLE to estimate signals is not new; in the context of 1-bit matrix completion, this approach was developed around 10 years ago. While the AC understands that the main claim is the convergence to stationary point, it should be noted that this is much weaker than convergence to global optimum.

**Reviewer Concerns:**

Reviewers' concern on empirical study might be fixed, but not quickly since the amount of experiments will be nontrivial.

**Reviewer Scores:**

Likely unchanged.

---

### Decision · Program_Chairs · 2026-01-26

Reject